# Efficient Stochastic Optimisation via Sequential Monte Carlo

James Cuin [1]   Davide Carbone [2]   Yanbo Tang [1]   O. Deniz Akyildiz [1]

## Abstract

The problem of optimising functions with intractable gradients frequently arises in machine learning and statistics, ranging from maximum marginal likelihood estimation procedures to fine-tuning of generative models. Stochastic approximation methods for this class of problems typically require inner sampling loops to obtain (biased) stochastic gradient estimates, which rapidly becomes computationally expensive. In this work, we develop sequential Monte Carlo (SMC) samplers for optimisation of functions with intractable gradients. Our approach replaces expensive inner sampling methods with efficient SMC approximations, which can result in significant computational gains. We establish convergence results for the basic recursions defined by our methodology which SMC samplers approximate. We demonstrate the effectiveness of our approach on the reward-tuning of energy-based models within various settings.

## 1. Introduction

In machine learning and statistics, we often encounter minimisation problems for a loss function $\ell(\theta) : \mathbb{R}^{d_\theta} \to \mathbb{R}$, where the gradient can be expressed as an expectation over a parameter-dependent (intractable) distribution $\pi_\theta$, i.e., $\nabla_\theta \ell(\theta) = \mathbb{E}_{X \sim \pi_\theta}[H_\theta(X)]$, for some measurable function $H_\theta : \mathbb{R}^{d_x} \to \mathbb{R}^{d_\theta}$. Developing efficient gradient estimators in this setting is critical as it arises in many applications, including maximum marginal likelihood estimation (Atchadé et al., 2017; De Bortoli et al., 2021; Akyildiz et al., 2025; Gruffaz et al., 2024), training energy-based (Song & Kingma, 2021; Carbone et al., 2023; Oliva et al., 2025) and latent energy-based models (Pang et al., 2020; Marks et al.,

2025), and fine-tuning of generative models (Marion et al., 2025). The main challenge is the dependence of the distribution $\pi_\theta$ on the parameter $\theta$, which is often intractable. Consequently, obtaining estimates of the gradient $\nabla_\theta \ell(\theta)$ requires inner sampling loops using Markov chain Monte Carlo (MCMC) methods (Atchadé et al., 2017; De Bortoli et al., 2021), which can be expensive and slow to converge.

In this work, we propose a novel approach to this class of problems using sequential Monte Carlo (SMC) (Del Moral et al., 2006). We replace the inner sampling loops with efficient SMC approximations, allowing us to develop a framework that is both computationally efficient and theoretically sound. Our main contributions are as follows:

**(C1)** We develop a general and flexible SMC-based framework for optimising functions with intractable gradients. We show that some existing algorithms can be seen as special cases of our framework. The resulting class of SMC algorithms are efficient and general, and can be adapted to various problem settings.

**(C2)** We propose a high-level theoretical analysis of the iterations which our SMC-based algorithms approximate. We provide convergence rates under standard assumptions for idealised versions of our algorithms, indicating their effectiveness in practice. We further provide a theoretical discussion on the behaviour of the effective sample size (ESS) under our setting.

**(C3)** We demonstrate the effectiveness of our approach through extensive experiments on fine-tuning energy-based models, showcasing significant improvements in both computational efficiency and optimisation performance.

### 1.1. Notation

Fix a measurable space $(E, \mathcal{E})$. Given a measurable function $f : E \to \mathbb{R}^d$ and a measure $\mu$ on $(E, \mathcal{E})$, we write $\mu(f) := \int_E f(x)\mu(\mathrm{d}x) \in \mathbb{R}^d$. We use $\| \cdot \|_2$ to denote the Euclidean norm, and we use $B_x(y)$ to denote an $L^2$ ball centred at $x$ with radius $y$. A Markov kernel $\mathsf{K}$ from a measurable space $(E, \mathcal{E})$ to $(E', \mathcal{E}')$ is a mapping $\mathsf{K} : E \times \mathcal{E}' \to [0, 1]$ such that for each $x \in E$, $\mathsf{K}(x, \cdot)$ is a probability measure on $\mathcal{E}'$, and for each $A \in \mathcal{E}'$, the map $x \mapsto \mathsf{K}(x, A)$ is $\mathcal{E}$-measurable. We denote the density of a Markov kernel $\mathsf{K}$ with respect to the Lebesgue measure by the same symbol.

[1]Department of Mathematics, Imperial College London, London, United Kingdom [2]Laboratoire de Physique de l'Ecole Normale Supérieure, ENS Université PSL, CNRS, Sorbonne Université, Université de Paris, Paris, France. Correspondence to: James Cuin <jamie.cuin23@imperial.ac.uk>.

*Proceedings of the 43rd International Conference on Machine Learning*, Seoul, South Korea. PMLR 306, 2026. Copyright 2026 by the author(s).

## 2. Background

### 2.1. Stochastic Optimisation with Intractable Gradients

We are interested in gradient-based optimisation for a loss $\ell : \mathbb{R}^{d_\theta} \to \mathbb{R}$ where the gradient is given by an expectation over a parameter-dependent distribution $\pi_\theta$, for $\theta \in \mathbb{R}^{d_\theta}$. We assume that the distribution $\pi_\theta$ has a density with respect to the Lebesgue measure on $\mathbb{R}^{d_x}$ given by

$$\pi_\theta(x) = e^{-U_\theta(x)}/Z_\theta, \quad Z_\theta = \int_{\mathbb{R}^{d_x}} e^{-U_\theta(x)} \mathrm{d}x, \quad (1)$$

where $U_\theta : \mathbb{R}^{d_x} \to \mathbb{R}$ is the potential function and $Z_\theta$ is the intractable normalising constant. We further assume that there exists a measurable function $H_\theta : \mathbb{R}^{d_x} \to \mathbb{R}^{d_\theta}$ such that the gradient of the loss $\ell$ can be expressed as

$$\nabla_\theta \ell(\theta) = \mathbb{E}_{X \sim \pi_\theta}\left[H_\theta(X)\right], \quad (2)$$

where $\pi_\theta$ is defined in (1). This setting is of significant interest in the literature and includes a wide range of applications, some of which are summarised in the next section.

### 2.2. Related Literature

Below we summarize some of the key applications that fit the setting of (2) and highlight the key existing work from a methodological perspective. We omit application-oriented works as there are simply too many works using the frameworks developed for estimating the gradient (2).

**Maximum marginal likelihood estimation (MMLE)**: Consider a latent variable model with observed data $Y$ and latent variables $X$, where the joint distribution is given by $p_\theta(x, y)$. The marginal likelihood is $p_\theta(y) = \int p_\theta(x, y) dx$. The gradient of the negative log-marginal likelihood can be expressed as $\nabla_\theta \ell(\theta) = -\mathbb{E}_{X \sim p_\theta(x|y)}[\nabla_\theta \log p_\theta(x, y)]$. This fits the form of (2) with $\pi_\theta = p_\theta(\cdot|y)$ and $H_\theta(\cdot) = -\nabla_\theta \log p_\theta(\cdot, y)$. This setting is related to the Expectation-Maximisation (EM) algorithm (Dempster et al., 1977) and its stochastic variants (Fort & Moulines, 2003; Gruffaz et al., 2024). A number of methods that use (approximate) MCMC methods appeared recently in the literature, including Atchadé et al. (2017); De Bortoli et al. (2021); Gruffaz et al. (2024). An SMC-based approach was proposed by Cuin et al. (2025), which is a special case of our framework, see Remark 1. We also note that a number of other interacting particle-based approaches were also proposed for this problem (Kuntz et al., 2023; Akyildiz et al., 2025).

**Training energy-based models (EBMs)**: For a data distribution $p_{\text{data}}$, the training objective for training EBMs is to minimise the negative log-likelihood $\ell(\theta) = -\mathbb{E}_{Y \sim p_{\text{data}}}[\log p_\theta(Y)]$, for $\pi_\theta(\cdot) \propto \exp(-E_\theta(\cdot))$. The gradient of this loss can be expressed as $\nabla_\theta \ell(\theta) = \mathbb{E}_{X \sim \pi_\theta}[\nabla_\theta E_\theta(X) - \mathbb{E}_{Y \sim p_{\text{data}}}[\nabla_\theta E_\theta(Y)]$, so

that, $H_\theta(\cdot) = \nabla_\theta E_\theta(\cdot) - \mathbb{E}_{Y \sim p_{\text{data}}}[\nabla_\theta E_\theta(Y)]$. Standard contrastive divergence (CD) (Hinton, 2002; Song & Kingma, 2021) uses (approximate) MCMC steps to sample from $\pi_\theta$. Instead of fresh MCMC runs, persistent contrastive divergence (PCD) algorithms (Tieleman, 2008) maintain a set of unweighted particles that approximate the gradient (see Oliva et al. (2025) for an analysis of this method). Improving on this particle-based approach, Carbone et al. (2023) develops an SMC-based approach to estimate the intractable gradient (which will be also shown to be a special case of our general method, see Remark 1).

**Reward tuning**: The alignment of a (pre-trained) parameterised generative model $\pi_0$ with a downstream objective, encoded through the potentially non-differentiable reward function $R$, is defined by the KL-regularised objective, $\ell(\theta) = -\mathbb{E}_{\pi_\theta}[R(X)] + \beta_{\text{KL}} D_{\text{KL}}(\cdot \| \cdot)$, where the forward KL variant is natural when direct access to a reference distribution is available (see Appendix E.1), whilst the reverse KL variant is preferred when $\pi_0$ is specified implicitly (see Appendix E.2). Indeed, the respective gradient fits the form of (2) (see Appendix F.5). Optimising such an objective requires estimates under the evolving model $\pi_\theta$, whose $\theta$ dependence is typically implicit through a stochastic sampling procedure. Standard training approaches based on MCMC, mentioned above, are typically employed. As an alternative, Marion et al. (2025) develops a single-loop scheme that update parameters and particles in a joint manner.

**Sequential Monte Carlo methods for optimisation.** In the special case of MMLE, the most related works to ours are Johansen et al. (2008) and Crucinio (2025) which proposed SMC-based MMLE methods. Johansen et al. (2008) propose a method based on SMC which concentrates on the minimisers of the marginal likelihood as $N$ grows. Crucinio (2025) develops an approach based on mirror descent for updating parameters while SMC is used for posterior sampling. A similar method is also studied in Cuin et al. (2025) for MMLE where the approach is developed with unadjusted Langevin algorithm (ULA)-based proposals and general optimisers. While derived for different settings, we also mention a number of SMC methods for optimisation here. In Miguez et al. (2009; 2010), authors introduce an SMC method for cost functions that can be sequentially optimised in space (rather than time). Gradient-free stochastic optimisation methods have also been developed based off SMC methods, see, e.g. Akyildiz et al. (2020); Gerber & Douc (2022). A related line of work is the class of EM algorithms for state-space models (Kantas et al., 2015; Chopin et al., 2020; Yildirim et al., 2012), which use SMC methods to estimate gradients of the marginal log-likelihood. While the latter SMC-EM methods can be seen as special cases of our framework, the former optimisation methods operate in different settings.

# 3. Sequential Monte Carlo for Optimisation

Consider a generic first-order stochastic optimiser update:

$$\theta_{k+1} = \mathsf{OPT}(\theta_k, g_k, \gamma_k), \qquad (3)$$

where $g_k$ is the output of a (possibly stochastic) gradient estimator and $(\gamma_k)_{k \geq 1}$ is the step-size sequence. In our setting, $g_k \approx \nabla_\theta \ell(\theta_k) = \mathbb{E}_{X \sim \pi_{\theta_k}}[H_{\theta_k}(X)]$ is an intractable gradient of the loss function $\ell(\theta)$, and $\pi_{\theta_k}$ is a probability distribution that depends on the current parameter $\theta_k$. In this section, we are interested in developing SMC methods to sequentially sample from the sequence of distributions $(\pi_{\theta_k})_{k \geq 0}$, in order to construct an estimator for $\nabla_\theta \ell(\theta_k)$ at each iteration of a first order optimiser. As opposed to standard methods which use MCMC kernels to approximately sample from each $\pi_{\theta_k}$ independently, we propose to leverage the scalability of SMC methods to sequentially sample from the sequence $(\pi_{\theta_k})_{k \geq 0}$, reusing samples from previous iterations to improve the efficiency of the sampling procedure.

## 3.1. Feynman-Kac flows for gradient estimation

We follow the SMC samplers methodology (Del Moral et al., 2006). Consider the target distribution at time $k$, $\pi_{\theta_k}$, and let $\mathsf{L}_{n-1} : E \times \mathcal{E} \to [0,1]$ be a sequence of backward Markov kernels with densities $\mathsf{L}_{n-1}(x_n, x_{n-1})$ for $1 \leq n \leq k$. We then define the following sequence of unnormalised probability distributions on the product space $E^{k+1}$:

$$\Pi_{\theta_{0:k}}(x_{0:k}) := \Pi_{\theta_k}(x_k) \prod_{n=1}^{k} \mathsf{L}_{n-1}(x_n, x_{n-1}), \qquad (4)$$

where $\Pi_\theta(x) = e^{-U_\theta(x)}$ is the unnormalised density associated to $\pi_\theta$. Let $\pi_{\theta_{0:k}} \propto \Pi_{\theta_{0:k}}$ be the associated probability distribution on $E^{k+1}$. It can be shown that the $x_k$-marginal is indeed $\pi_{\theta_k}$. The idea of SMC samplers is then to construct a sequence of approximations to the distributions $(\pi_{0:k})_{k \geq 0}$ on the product spaces $(E^{k+1})_{k \geq 0}$, using a collection of weighted particles. Marginalising these approximations to the last coordinate $k$ then provides approximations to the sequence of target distributions $(\pi_{\theta_k})_{k \geq 0}$.

To define a sequential importance sampling scheme, we also need to define a proposal. Let $\mathsf{K}_n : E \times \mathcal{E} \to [0,1]$ be a sequence of forward Markov kernels with densities $\mathsf{K}_n(x_{n-1}, x_n)$ for $1 \leq n \leq k$. Define the following sequence of proposal distributions on the product space $E^{k+1}$:

$$q_{0:k}(x_{0:k}) := q_0(x_0) \prod_{n=1}^{k} \mathsf{K}_n(x_{n-1}, x_n), \qquad (5)$$

where $q_0$ is an initial distribution on $E$. The unnormalised importance weights associated to the target distribution (4)

and the proposal distribution (5) can then be computed as

$$W_k(x_{0:k}) := \frac{\Pi_{\theta_{0:k}}(x_{0:k})}{q_{0:k}(x_{0:k})} \qquad (6)$$

$$= W_{k-1}(x_{0:k-1}) G_k(x_{k-1}, x_k), \qquad (7)$$

where

$$G_k(x_{k-1}, x_k) := \frac{\Pi_{\theta_k}(x_k) \mathsf{L}_{k-1}(x_k, x_{k-1})}{\Pi_{\theta_{k-1}}(x_{k-1}) \mathsf{K}_k(x_{k-1}, x_k)}. \qquad (8)$$

Define a test function $\varphi : E \to \mathbb{R}^{d_\theta}$. Then, we have the following Feynman-Kac formulae.

**Theorem 1** (Del Moral 2004). *For any $k \geq 1$, define $G_0(x_0) := \Pi_{\theta_0}(x_0)/q_0(x_0)$. If $X_{0:k} \sim q_{0:k}$, then*

$$\Pi_{\theta_k}(\varphi) := \mathbb{E}\left[\varphi(X_k) \prod_{n=0}^{k} G_n(X_{n-1}, X_n)\right]. \qquad (9)$$

*and $\pi_{\theta_k}(\varphi) = \Pi_{\theta_k}(\varphi)/\Pi_{\theta_k}(1)$ where $G_0(X_{-1}, X_0) := G_0(X_0)$.*

*Proof.* Let $(X_0, \ldots, X_k) \sim q_{0:k}$. By definition,

$$\mathbb{E}\left[\varphi(X_k) \prod_{n=0}^{k} G_n(X_{n-1}, X_n)\right]$$

$$= \int \varphi(x_k) q_{0:k}(x_{0:k}) \prod_{n=0}^{k} G_n(x_{n-1}, x_n) \, \mathrm{d}x_{0:k}$$

$$= \int \varphi(x_k) \Pi_{\theta_k}(x_k) \prod_{n=1}^{k} \mathsf{L}_{n-1}(x_n, x_{n-1}) \, \mathrm{d}x_{0:k},$$

where we used (5)–(8) and by telescoping the $\Pi_{\theta_n}$ terms. Integrating out $x_{0:k-1}$ using that each $\mathsf{L}_{n-1}$ is a Markov kernel yields

$$\mathbb{E}\left[\varphi(X_k) \prod_{n=0}^{k} G_n(X_{n-1}, X_n)\right] = \int \varphi(x_k) \Pi_{\theta_k}(x_k) \, \mathrm{d}x_k.$$

$\square$

This is a well-known identity in SMC-literature. In our context, its importance becomes clear when we set $\varphi = H_{\theta_k}$ which results in a recursion for the intractable gradient $\nabla_\theta \ell(\theta_k) = \pi_{\theta_k}(H_{\theta_k})$. Using this, we construct a sequential sampling scheme to approximate $\nabla_\theta \ell(\theta_k)$ at each iteration of the first-order optimiser (3).

## 3.2. SMC Implementation

We now describe a standard SMC sampler that approximates the Feynman–Kac flow in Theorem 1 (Del Moral et al.,

2006). Let $\{(X_k^{(i)}, w_k^{(i)})\}_{i=1}^N$ denote weighted particles at time $k$. We first sample $X_0^{(i)} \sim q_0$ for $i = 1, \ldots, N$ and set

$$W_0^{(i)} = G_0(X_0^{(i)}), \qquad G_0(x_0) := \frac{\Pi_{\theta_0}(x_0)}{q_0(x_0)},$$

so that $W_0^{(i)} \equiv 1$ when $q_0 = \pi_{\theta_0}$. For $k \geq 1$, propagate

$$\bar{X}_k^{(i)} \sim \mathsf{K}_k\left(X_{k-1}^{(i)}, \cdot\right),$$

for every $i = 1, \ldots, N$ and update the unnormalised weights using (8),

$$W_k^{(i)} = W_{k-1}^{(i)} G_k\left(X_{k-1}^{(i)}, \bar{X}_k^{(i)}\right). \tag{10}$$

Normalise $w_k^{(i)} = W_k^{(i)} / \sum_{j=1}^N W_k^{(j)}$ and form the estimator for the test function $\varphi$ with

$$\pi_{\theta_k}^N(\varphi) := \sum_{i=1}^N w_k^{(i)} \varphi(\bar{X}_k^{(i)}). \tag{11}$$

In general, the product of incremental weights in (10) can lead to weight degeneracy, where only a few particles have significant weights. To mitigate this, resampling is typically employed. For this, we define the effective sample size

$$\mathrm{ESS}_k := \frac{\left(\sum_{i=1}^N W_k^{(i)}\right)^2}{\sum_{i=1}^N \left(W_k^{(i)}\right)^2} = \frac{1}{\sum_{i=1}^N \left(w_k^{(i)}\right)^2}. \tag{12}$$

If $\mathrm{ESS}_k$ falls below a threshold (e.g., $\tau N$ for $\tau \in (0,1)$), we resample the particles according to $\{W_k^{(i)}\}_{i=1}^N$ and reset weights to $1/N$.

## 3.3. Stochastic Optimisation via SMC samplers

We now describe our SMC-based stochastic optimisation algorithm, which we term stochastic optimisation via sequential Monte Carlo (SOSMC). At each iteration $k$ of the optimiser (3), we run one iteration of the SMC sampler described above to obtain weighted particles $\{(X_k^{(i)}, w_k^{(i)})\}_{i=1}^N$ approximating $\pi_{\theta_k}$. We then use the estimator (11) with $\varphi = H_{\theta_k}$ to approximate the intractable gradient $\nabla_\theta \ell(\theta_k)$:

$$g_k := \pi_{\theta_k}^N(H_{\theta_k}) = \sum_{i=1}^N w_k^{(i)} H_{\theta_k}(X_k^{(i)}). \tag{13}$$

This estimator is then plugged into the first-order optimiser update (3) to obtain the next iterate $\theta_{k+1}$. The complete SOSMC procedure is given in Algorithm 1.

**Remark 1.** When the forward kernel is chosen as the ULA with step-size $h > 0$:

$$\mathsf{K}_k(x_{k-1}, x_k) = \mathcal{N}(x_k; x_{k-1} - h\nabla U_{\theta_{k-1}}(x_{k-1}), 2hI),$$

---

**Algorithm 1** Stochastic Optimisation via SMC (SOSMC)

**Require:** Initial parameter $\theta_0$, initial distribution $q_0$, number of particles $N$, resampling threshold $\tau$, first-order optimiser $\mathsf{OPT}$, sequence of forward kernels $\{\mathsf{K}_k\}_{k\geq 1}$, sequence of backward kernels $\{\mathsf{L}_k\}_{k\geq 0}$.

1: **for** $k = 0, 1, 2, \ldots$ **do**
2:   **if** $k = 0$ **then**
3:     **for** $i = 1$ to $N$ **do**
4:       Sample $\bar{X}_0^{(i)} \sim q_0$
5:       Set $W_0^{(i)} = \Pi_{\theta_0}(\bar{X}_0^{(i)})/q_0(\bar{X}_0^{(i)})$
6:     **end for**
7:   **else**
8:     **for** $i = 1$ to $N$ **do**
9:       Sample $\bar{X}_k^{(i)} \sim \mathsf{K}_k(X_{k-1}^{(i)}, \cdot)$
10:       Update weights:

$$W_k^{(i)} = W_{k-1}^{(i)} \frac{\Pi_{\theta_k}(\bar{X}_k^{(i)})\mathsf{L}_{k-1}(\bar{X}_k^{(i)}, X_{k-1}^{(i)})}{\Pi_{\theta_{k-1}}(X_{k-1}^{(i)})\mathsf{K}_k(X_{k-1}^{(i)}, \bar{X}_k^{(i)})}$$

11:     **end for**
12:   **end if**
13:   Normalize weights: $w_k^{(i)} = W_k^{(i)} / \sum_{j=1}^N W_k^{(j)}$
14:   Compute gradient estimator:

$$g_k = \sum_{i=1}^N w_k^{(i)} H_{\theta_k}(\bar{X}_k^{(i)})$$

15:   Update parameter:

$$\theta_{k+1} = \mathsf{OPT}(\theta_k, g_k, \gamma_k)$$

16:   **if** $\mathrm{ESS}_k < \tau N$ **then**
17:     Sample $\{a_k^{(i)}\}_{i=1}^N \sim \mathrm{Categorical}(\{w_k^{(i)}\}_{i=1}^N)$
18:     Set: $X_k^{(i)} = \bar{X}_k^{(a_k^{(i)})}$ for all $i$
19:     Reset weights: $W_k^{(i)} = 1$ for all $i$
20:   **else**
21:     Set: $X_k^{(i)} = \bar{X}_k^{(i)}$ for all $i$
22:   **end if**
23: **end for**

---

and

$$\mathsf{L}_{k-1}(x_k, x_{k-1}) = \mathcal{N}(x_{k-1}; x_k - h\nabla U_{\theta_k}(x_k), 2hI),$$

we recover a number of existing stochastic optimisation algorithms as special cases of SOSMC, see Appendix A.1. When $\ell(\theta)$ is the negative log-marginal likelihood in a latent variable model (LVM), our choice of kernels results in the so-called Jarzynski adjusted Langevin algorithm for EM (JALA-EM) algorithm for MMLE, developed by Cuin et al. (2025). On the other hand, if $\ell(\theta)$ is the negative maximum-likelihood loss, also known as cross-entropy in that context, arising from a training objective in an EBM, our algorithm

takes the form of the SMC sampler proposed by Carbone et al. (2023). While these two papers develop their methods by fixing the choice of kernels, our SOSMC framework allows for more general choices.

**Remark 2.** Our framework offers flexible choices for forward and backward kernels in algorithm design. For example, $\mathsf{K}_k$ can be chosen to be $\pi_{\theta_k}$-invariant, such as a Metropolis adjusted Langevin algorithm (MALA) kernel or others discussed in Appendix A.2, whilst $\mathsf{L}_{k-1}$ may be the corresponding time reversal, which simplifies the weights significantly (Del Moral et al., 2006). It should also be noted that the discretized kernel admits several equivalent parametrizations: the forward and backward kernels can each be taken as $\mathsf{K}_k$ or $\mathsf{K}_{k-1}$, and all resulting combinations coincide in the continuous-time limit while remaining consistent with discrete-time SMC schemes. We refer to (Schönle et al., 2025b), and in particular to Figure 2 therein, for a detailed discussion. One could also adopt underdamped Langevin dynamics or deterministic dynamics and coherently estimate the corresponding importance weights, see for instance (Schönle et al., 2025a).

## 4. Analytical results

In this section, we provide a basic theoretical analysis for our method to understand its limiting behavior. We first analyze an idealized scheme that assumes the expectations in the Feynman–Kac identity (9) can be computed exactly. We then discuss how the variance of the weights behave given our choice of marginal distributions.

### 4.1. Gradient descent based on Feynman-Kac flows

We analyze an idealised version of the scheme where the expectations in (9) are computed exactly. This is not possible in practice, but it clarifies the limiting behaviour. We use the notation from Section 2, in particular (1) and (2). The next lemma shows that, under exact expectations, the Feynman–Kac weights recover the true gradient.

**Lemma 1** (Exact gradient recovery). *Assume* $Z_\theta = \int e^{-U_\theta(x)} \mathrm{d}x < \infty$ *for all* $\theta \in \mathbb{R}^{d_\theta}$. *Fix* $\theta_0$, *set* $X_0 \sim \pi_{\theta_0}$ *and* $W_0 = 1$, *and define* $(\theta_k, X_k, W_k)_{k \geq 0}$ *by*

$$\theta_k = \theta_{k-1} - \gamma \frac{\mathbb{E}[H_{\theta_{k-1}}(X_{k-1})W_{k-1}]}{\mathbb{E}[W_{k-1}]}, \quad (14)$$

$$X_k \sim \mathsf{K}_k(X_{k-1}, \cdot), \quad (15)$$

$$W_k = W_{k-1} \frac{\Pi_{\theta_k}(X_k)\mathsf{L}_{k-1}(X_k, X_{k-1})}{\Pi_{\theta_{k-1}}(X_{k-1})\mathsf{K}_k(X_{k-1}, X_k)}. \quad (16)$$

*Then, for all* $k \geq 1$,

$$\frac{\mathbb{E}[H_{\theta_{k-1}}(X_{k-1})W_{k-1}]}{\mathbb{E}[W_{k-1}]} = \pi_{\theta_{k-1}}(H_{\theta_{k-1}}) = \nabla\ell(\theta_{k-1}).$$

*Proof.* By construction, $W_{k-1} = \prod_{n=1}^{k-1} G_n(X_{n-1}, X_n)$ with incremental weights $G_n$ as in (8). Applying the Feynman–Kac identity (9) with $\varphi = H_{\theta_{k-1}}$ and $\varphi = 1$ yields

$$\mathbb{E}[H_{\theta_{k-1}}(X_{k-1})W_{k-1}] = \Pi_{\theta_{k-1}}(H_{\theta_{k-1}}),$$
$$\mathbb{E}[W_{k-1}] = \Pi_{\theta_{k-1}}(1).$$

Using $\pi_{\theta_{k-1}}(\cdot) = \Pi_{\theta_{k-1}}(\cdot)/\Pi_{\theta_{k-1}}(1)$ and (2) gives the claim. $\square$

In order to prove that gradient descent converges for gradients built using the iterations in Lemma (1), we need an assumption on our loss $\ell$.

**A 1.** *The function* $\ell : \mathbb{R}^{d_\theta} \to \mathbb{R}$ *is* $\mu$-*Polyak-Łojasiewicz (PŁ), i.e.*

$$\ell(\theta) - \inf_\theta \ell(\theta) \leq \frac{1}{2\mu}\|\nabla\ell(\theta)\|^2,$$

*for all* $\theta$ *and* $L$-*smooth, i.e.,*

$$\|\nabla\ell(\theta) - \nabla\ell(\theta')\| \leq L\|\theta - \theta'\|,$$

*for all* $\theta, \theta' \in \mathbb{R}^{d_\theta}$.

**Proposition 1** (Convergence of the idealised scheme.). *Assume* $Z_\theta < \infty$ *for all* $\theta \in \mathbb{R}^{d_\theta}$ *and let the iterates be defined by* (14) *with step-size* $\gamma > 0$ *and Markov kernels* $\mathsf{K}_k, \mathsf{L}_k$. *Under A1, for any* $\gamma \leq 1/L$ *we have*

$$\ell(\theta_k) - \inf_\theta \ell(\theta) \leq (1 - \gamma\mu)^k(\ell(\theta_0) - \inf_\theta \ell(\theta)). \quad (17)$$

*Proof.* By Lemma 1, the update for $\theta_k$ reduces to exact gradient descent $\theta_k = \theta_{k-1} - \gamma\nabla\ell(\theta_{k-1})$. The stated rate then follows from the standard convergence guarantee of gradient descent for $\mu$-PŁ and $L$-smooth objectives, see, e.g., Garrigos & Gower (2023, Theorem 3.9). $\square$

**Remark 3.** When Algorithm 1 is implemented with finitely many particles, the update is driven by a stochastic, biased, self-normalised SMC estimate of the gradient. For a fixed target sequence, standard SMC theory gives, under stability and regularity conditions, bias of order $\mathcal{O}(1/N)$ and mean-squared error (MSE) of order $\mathcal{O}(1/N)$ for bounded test functions (Del Moral, 2004; Crisan & Doucet, 2002). In our setting, however, the target sequence depends on previous particle approximations, so proving such bounds requires tools for mean-field/interacting particle models (Del Moral & Rio, 2011). These bounds could then be combined with biased stochastic gradient descent (SGD) analyses, see, e.g., Demidovich et al. (2023); Karimi et al. (2019); we leave the development of this theory to future work.

## 4.2. An Analysis for the Effective Sample Size

While standard SMC analysis can be applied to analyse the error of the particle approximations (which we leave to future work), we briefly discuss how our construction of marginal distributions $(\pi_{\theta_k})_{k=0}^K$ affects the performance of the method and related design choices implied by it.

For this, we analyse the following quantity

$$\rho_k(\gamma) := \frac{N}{1 + \chi^2(\pi_{\theta_k} \| \pi_{\theta_{k-1}})}, \qquad (18)$$

where $\chi^2(\pi_{\theta_k} \| \pi_{\theta_{k-1}})$ is the $\chi^2$-divergence between $\pi_{\theta_k}$ and $\pi_{\theta_{k-1}}$, and $\gamma > 0$ is the step-size used to go from $\theta_{k-1}$ to $\theta_k$. This quantity is closely related to the ESS of the particle system when the forward kernel $\mathsf{K}_k$ is $\pi_{\theta_k}$-invariant and the backward kernel $\mathsf{L}_{k-1}$ is the time reversal of $\mathsf{K}_k$. In this case, given samples from $\pi_{\theta_{k-1}}$, we have the relationship $\hat{\rho}_k^N(\gamma) = \mathrm{ESS}_k(\gamma)$ (Dai et al., 2022).

To gain insight into the behaviour of this quantity, we let $\pi_\theta = \mathcal{N}(\theta, \Sigma)$. Since $\theta_k = \theta_{k-1} - \gamma \nabla \ell(\theta_{k-1})$, using the exact form of $\chi^2$-divergence between Gaussians, we obtain

$$\rho_k(\gamma) = N \exp\left(-\gamma^2 \|\nabla \ell(\theta_{k-1})\|_{\Sigma^{-1}}^2\right), \qquad (19)$$

where $\|\cdot\|_{\Sigma^{-1}}$ denotes the Mahalanobis norm. This expression provides insights into the impact of step-size $\gamma$. In particular, larger step-sizes may lead to a sharp decrease in the ESS, indicating degeneracy. It is also apparent from (19) that larger gradients $\|\nabla \ell(\theta_{k-1})\|$ may also reduce the ESS. When the estimator for the gradient is replaced by its particle approximation, this can lead to high variance in the gradients, which can cause large norms for the gradients. This would also quickly degrade the ESS. It is clear that the change in the parameters $(\theta_k)_{k=0}^K$ should be controlled to maintain a reasonable ESS, which can be achieved by tuning the step-size $\gamma$ or using modern optimisers that clip the gradients.

The same intuition holds for more general models as summarised below.

**Proposition 2.** *Let $k \in \mathbb{N}$, $\|\nabla \ell(\theta_{k-1})\|_2 = L_k < \infty$ and let:*

$$J_k(\theta, x) = \left(\frac{\pi_\theta(x)}{\pi_{\theta_k}(x)} - 1 + (\theta - \theta_k)^\top \nabla_\theta U_\theta(x)\right)^2$$

*If*

$$\sup_{\theta \in B_{\theta_{k-1}}(\gamma L_k)} \mathbb{E}_{\pi_{\theta_{k-1}}}[J_k(\theta, X)] = o(\gamma^2),$$

*then the $\chi^2$-divergence between $\pi_{\theta_k}$ and $\pi_{\theta_{k-1}}$ satisfies*

$$\chi^2(\pi_{\theta_k}, \pi_{\theta_{k-1}}) = \gamma^2 \nabla \ell(\theta_{k-1})^\top I_{\theta_{k-1}} \nabla \ell(\theta_{k-1}) + o(\gamma^2),$$

*where $I_{\theta_{k-1}} = \mathbb{E}_{\theta_{k-1}}\left[\nabla_\theta U_{\theta_{k-1}}(X) \nabla_\theta U_{\theta_{k-1}}(X)^\top\right]$.*

The proof follows immediately from the definition of $\chi^2$-divergence. We show analogous results with more explicit sufficient conditions in Appendix B. For general models, the local geometry of the loss landscape plays a role in the selection of the step size, if the landscape is very flat then larger step sizes are appropriate and vice versa.

**Remark 4.** Adaptive SMC schemes based on ESS control are well established, particularly when intermediate targets $(\pi_k)_{k\geq 0}$ are specified a priori (Del Moral et al., 2012). Motivated by the exponential sensitivity outlined in (19), we adapt $\gamma_k$ using a simple multiplicative rule; with $c > 1$, we set $\gamma_k = c\gamma_{k-1}$ when $\mathrm{ESS} > \tau_{\mathrm{adapt}}N$, whilst set $\gamma_k = \gamma_{k-1}/c$ when $\mathrm{ESS} < \tau_{\mathrm{adapt}}N$, providing a simple, scale free mechanism for maintaining a stable discrepancy regime.

## 5. Experiments

We empirically investigate the performance of SOSMC below, with complete details provided in Appendix E. The code can be found in https://github.com/akyildiz-group/SOSMC and note experiments were run on a personal computer and a Google Colab T4 GPU.

### 5.1. Reward tuning of Langevin Processes

To begin, we revisit the Langevin reward tuning experiment of Marion et al. (2025), to serve as a controlled setting for comparing both single and nested loop sampling algorithms, that of implicit diffusion (IMPDIFF) and stochastic optimisation via unadjusted Langevin algorithm (SOUL) respectively, with not only the SOSMC-ULA variant of our method, but also those outlined in Appendix A.2. Indeed, this setup facilitates an analytically tractable stationary distribution that we can directly sample from, motivating the forward KL-regularised objective,

$$\ell(\theta) = -\mathbb{E}_{\pi_\theta}[R(X)] + \beta_{\mathrm{KL}}\mathrm{KL}(\pi_{\mathrm{ref}} \| \pi_\theta), \qquad (20)$$

where $\pi_\theta$ is the Gaussian mixture induced by the potential $V(\cdot, \theta)$ and $\pi_{\mathrm{ref}}$ is a Gaussian mixture with frozen parameters $\theta_{\mathrm{ref}}$, whilst the potentially non-differentiable reward functions considered, denoted by $R$, are designed to promote discontinuities and disconnected high-reward regions.

All methods considered maintain persistent particles that are propagated under the ULA, or respective variant kernel, treated as *stop-gradient* quantities, and used to form Monte Carlo estimates of $\nabla_\theta \ell$, as outlined in Appendix E.1. The methods differ, however, in how expectations under $\pi_\theta$ are approximated. Notably, IMPDIFF utilises unweighted particle averages after a single Langevin step, in contrast to SOSMC's weighted particle averages, where the former results in biased estimates when particles lag the evolving target distribution. Conversely, SOUL utilises time-averaging

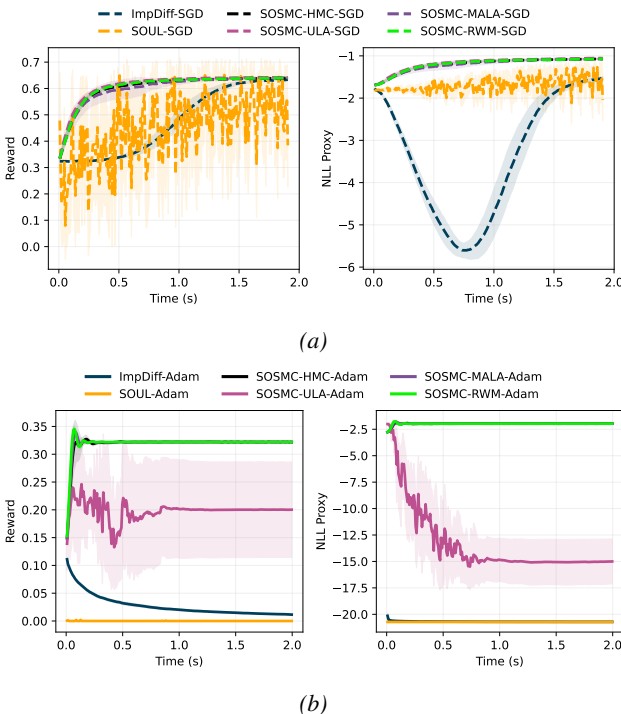

*(a)*

*(b)*

*Figure 1.* Wall-clock convergence of mean reward (left) and NLL (right), across 10 runs, for IMPDIFF, SOUL, and SOSMC variants under different OPT for *(a)* $V_{\text{dual}}$ & $R_{\text{smooth}}$; *(b)* $V_{\text{tight}}$ & $R_{\text{tight}}$.

along a single trajectory, in which the ULA kernel has been applied many times per outer iteration. In order to ensure fair comparison the number of kernel applications per outer iteration is equated across methods.

Performance is evaluated with respect to both outer iteration count and wall-clock runtime, where, for each iteration, the mean reward over the particle collection, as well as the mean energy is reported. Notably the latter quantity tracks the NLL in this setup (see Appendix E.1). Under a fixed wall-clock budget, enforced using device-synchronised timing, the methods are applied to various rewards and choices of $\pi_{\text{ref}}$, across multiple random seeds, in the cases that OPT is ADAM and SGD respectively. Notably, swift convergence, relative to IMPDIFF, is observed for SOSMC variants and SOUL, although the latter exhibits greater run-to-run variability (see Figure 1a) and also a failure mode in Figure 1b, where the single chain fails to transition between modes effectively, whilst the Metropolis-corrected kernel variants exhibit robust performance.

## 5.2. Reward Tuning of EBMs - 2D Datasets

Next, we consider reward tuning energy-based models (EBMs) on standard synthetic two-dimensional benchmarks, as visualised in Figure A.7. For each dataset, we first train a baseline EBM, $\pi_0(x) \propto \exp(-E_{\theta_0}(x))$, using PCD. We

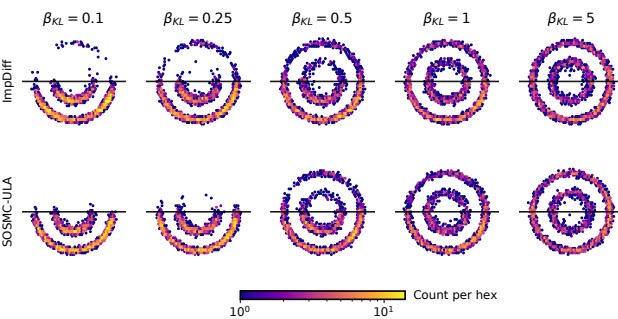

*Figure 2.* Terminal density snapshots of $\pi_{\theta_K}$, using identical initialisation and shared noise, across increasing $\beta_{\text{KL}}$, for $R_{\text{lower}}$.

note $E_\theta$ is parameterised by a MLP with smooth activations so that $\nabla_x E_\theta$ is well-defined for Langevin sampling, whilst the standard contrastive loss, $\mathbb{E}_{x^+ \sim \mathcal{D}}[E_\theta(x^+)] - \mathbb{E}_{x^- \sim \pi_\theta}[E_\theta(x^-)]$, is augmented with mild energy and gradient regularisation to prevent pathological energy levels and excessively sharp gradients (see Appendix E.2).

Given the frozen pre-trained model $\pi_0$, we tune a trainable energy $\pi_\theta(x) \propto \exp(-E_\theta(x))$ by optimising the reverse-KL regularised objective,

$$\ell(\theta) = -\mathbb{E}_{\pi_\theta}[R(X)] + \beta_{\text{KL}}\text{KL}(\pi_\theta \| \pi_0), \qquad (21)$$

initialising $\theta$ at $\theta_0$ and keeping $E_0 \equiv E_{\theta_0}$ fully detached. We focus on half-plane indicator rewards (e.g. $R_{\text{lower}} = \mathbf{1}_{\{x_2 < 0\}}$), which are non-differentiable, yet admit a closed-form optima, that is the exponentially tilted distribution $\pi^\star(x) \propto \pi_0(x) \exp(R(x)/\beta_{\text{KL}})$, providing an explicit target mass $\pi^\star(H)$ on the rewarded half-plane $H$ (see Appendix E.2), and a concrete reference when interpreting achieved reward levels.

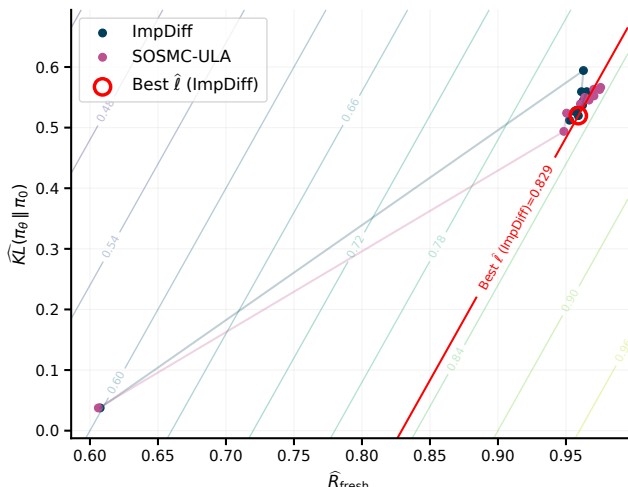

*Figure 3.* Tuning trajectories for IMPDIFF and SOSMC-ULA in the $(\widehat{R}_{\text{fresh}}, \widehat{\text{KL}})$ plane, for $\beta_{\text{KL}} = 0.25$. Contours denote level sets of $\ell$, with SOSMC-ULA to the right of (i.e. better than) the best IMPDIFF solution achieved during tuning (red).

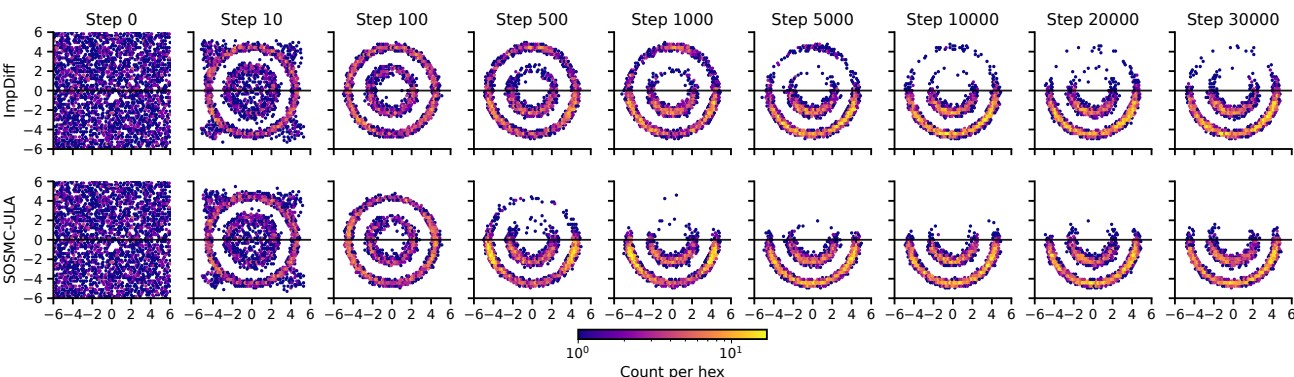

*Figure 4.* Density snapshots along the sampling evolution of $\pi_{\theta_K}$ with half-plane reward for both IMPDIFF (top) and SOSMC-ULA (bottom), for $\beta_{\text{KL}} = 0.25$, with identical initialisation and shared noise. Mass concentrates in the $R_{\text{lower}}$ region significantly faster for SOSMC-ULA.

A key distinction in this setting is between (i) the *particle reward* $\widehat{R}_{\text{particle}}(k)$, computed on the particles used for gradient estimation, and (ii) the *fresh reward* $\widehat{R}_{\text{fresh}}(k)$, estimated by running independent long Langevin chains targeting the *frozen* energy $E_{\theta_k}$ from a diffuse initialisation. The former reflects the particle population that drives $\theta$ updates, but can be notably biased when the persistent particle distribution lags $\pi_{\theta_k}$, whilst the latter is substantially more expensive, however tracks $\mathbb{E}_{\pi_{\theta_k}}[R(X)]$ more accurately and is thus the quantity we use to evaluate model performance. We also notably estimate $\text{KL}(\pi_\theta \| \pi_0)$ using numerical quadrature (see Appendix E.2), enabling optimisation trajectories in the $(\widehat{R}_{\text{fresh}}, \widehat{\text{KL}})$ plane to be reported against objective contours.

Here, IMPDIFF and SOSMC-ULA are compared across datasets, half-plane rewards, and regularisation strengths, as in Figure 2, with the latter method achieving better objective contours in cases where $\beta_{\text{KL}}$ is small (see, e.g., Figure 3), whilst achieving comparable objective values when $\beta_{\text{KL}}$ is large. Furthermore, a defining characteristic of IMPDIFF is that $\widehat{R}_{\text{particle}}$ can be a poor proxy for $\mathbb{E}_{\pi_\theta}[R(X)]$, particularly when $\theta$ changes rapidly, thus requiring long evaluation chains to be run. In contrast, weighted particle rewards obtained from SOSMC-ULA closely track $\mathbb{E}_{\pi_\theta}[R(X)]$ throughout tuning, with the method simultaneously needing significantly shorter evaluation chains to compute $\widehat{R}_{\text{fresh}}$. Indeed, this behaviour is consistent with the quicker stabilisation of Langevin trajectories into high reward regions under the tuned model, as illustrated in Figures 4 and A.10.

### 5.3. Reward Tuning of EBMs - MNIST

Finally, we consider the reward tuning of a convolutional EBM, pre-trained on MNIST using a sampling procedure distinct from the kernel utilised during tuning. Notably, this experiment serves to validate the robustness and applicability of SOSMC in a higher-dimensional setting, whilst also highlighting a practically relevant scenario in which the pre-

trained model is learnt under a specific sampling heuristic, while tuning proceeds under a ULA kernel.

Starting from the pre-trained reference model, $\pi_0(x) \propto \exp(-E_{\theta_0}(x))$, we again choose to optimise the reverse-KL regularised objective (21), initialising $\theta = \theta_0$, and keeping the reference energy model frozen throughout tuning. Three reward functions are considered, specifically a *brightness*, *darkness*, and *half-plane* reward as detailed in Appendix E.3. Since the sampling heuristic differs from the ULA kernel used within tuning, we evaluate performance using $\widehat{R}_{\text{fresh}}$, obtained through long chains under the original pre-training sampler targeting the *fixed* tuned energy $E_{\theta_k}$, giving an estimate of $\mathbb{E}_{\pi_{\theta_k}}[R(X)]$ reflecting the model distribution realised by the established sampling procedure. In contrast, $\widehat{R}_{\text{particle}}$ is interpreted solely as an optimisation diagnostic.

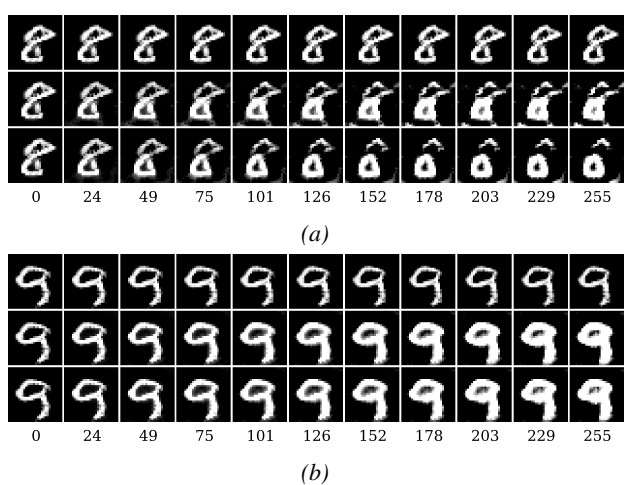

*Figure 5.* Example sampling trajectories, with step index, under the pre-training sampler kernel, for the pre-trained model (top), and $\pi_{\theta_K}$ for both IMPDIFF (middle) and SOSMC-ULA (bottom). In *(a)* the reward favours **mass in the lower half-plane**, whilst in *(b)* reward favours **brighter** images, as outlined in Appendix E.3.

Across all rewards and values of $\beta_{\mathrm{KL}}$, both IMPDIFF and SOSMC-ULA consistently obtain $\pi_{\theta_K}$ with increased $\widehat{R}_{\mathrm{fresh}}$ relative to the pre-trained baseline. Indeed, due to the explicit regularisation in (21), improvements in $\mathbb{E}_{\pi_\theta}[R(X)]$ alone are not sufficient to determine optimisation quality, nevertheless we note the resulting *fresh* samples are consistent with the respective value of $\beta_{\mathrm{KL}}$, whilst not exhibiting *reward hacking*, as illustrated in Figure 5a and 5b. Consequently, this experiment demonstrates SOSMC-ULA to be applicable to realistic image-based settings without degrading optimisation behaviour, supporting the generality of the proposed methodology beyond controlled low-dimensional experiments.

### 5.4. MMLE - Image Deblurring

To demonstrate an application beyond reward tuning, we now consider a Bayesian image deblurring problem, following the setup of Encinar et al. (2025), where the goal is to recover a latent clean image $x \in \mathbb{R}^{d_x}$ from a blurred noisy observation $y = Bx^\star + \varepsilon$, where $B$ is a known blurring operator, $x^\star$ is the ground-truth image, and $\varepsilon \sim \mathcal{N}(0, \sigma^2 I)$. Notably, $B$ is information destroying and thus the inverse problem is ill-conditioned, which we address through equipping the likelihood with an isotropic total-variation prior, $p_\theta = C(\theta) \exp(-e^\theta \mathrm{TV}(x))$, so that the resulting posterior takes the form

$$\pi_\theta(x) \propto \exp\left(-\frac{1}{2\sigma^2}\|y - Bx\|_2^2 - e^\theta \mathrm{TV}(x) + \log C(\theta)\right),$$

where $\mathrm{TV}(x) = \|\nabla_d x\|_1$ denotes the total variation, with $\nabla_d$ denoting the two-dimensional discrete gradient operator which is non-differentiable. Since TV is positively one-homogeneous, $C(\theta) \propto e^{d_x \theta}$, and, as detailed in Appendix E.4, serves as a MMLE problem satisfying (2), where

$$\nabla_\theta \ell(\theta) = \mathbb{E}_{X \sim \pi_\theta}\left[e^\theta \mathrm{TV}(X) - d_x\right].$$

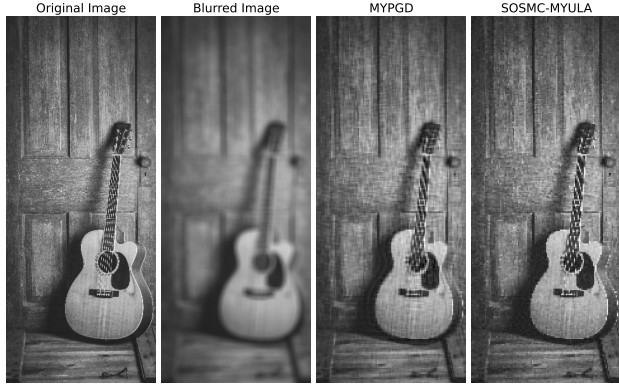

*Figure 6.* The ground truth image (left), blurred image (middle-left) and posterior mean reconstructions for MYPGD (middle-right) and SOSMC-MYULA (right) particle clouds, for image deblurring Setup A.

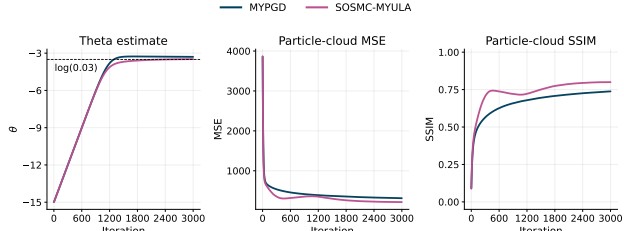

*Figure 7.* Evolution of parameter estimates (left), MSE (middle) and structural similarity index measure (right) for MYPGD and SOSMC-MYULA particle clouds, for the image deblurring Setup A, with $N = 20$ particles and $\theta_0 = -15$. The horizontal dashed line indicates the strength of the total variation prior manually set in Durmus et al. (2018) and Goldman et al. (2022).

Here, the SOSMC-Moreau-Yosida unadjusted Langevin algorithm (MYULA) variant is compared against the Moreau-Yosida particle gradient descent (MYPGD) baseline introduced in Encinar et al. (2025), so that both methods share the same proximal image dynamics (see Appendix E.4) yet differ in how the evolving posterior is approximated. Indeed, the former method maintains a weighted particle cloud to approximate $\pi_{\theta_k}$, in contrast to the persistent unweighted particle cloud maintained by the latter method.

Across different setups (see Figures 6, A.16 and A.19), where both the ground-truth and blurring operator applied is altered, superior performance for SOSMC-MYULA is observed. In particular, improvements in both the MSE and SSIM are observed across setups (see Figures 7, A.18 and A.20), as are sharper image reconstructions (see Figures A.21, A.17 and A.22). Full experimental details can be found in Appendix E.4.

## 6. Conclusion

We formulate an SMC framework for optimisation of general loss functions whose gradients are intractable. Our algorithm is based on general SMC samplers (Del Moral et al., 2006) which provides an efficient structure that avoids expensive MCMC loops for estimating gradients. The generality of our framework allows tailored methods to be developed. Future work will focus on two main directions: (i) theoretical guarantees using general theory of SMC and stochastic optimisation to obtain convergence rates for our particle approximations, and (ii) exploration of accelerated schemes that can improve the fast convergence behaviour of SOSMC, especially when coupled with adaptive schemes like in Kim et al. (2025).

## Acknowledgements

J. C. is supported by EPSRC through the Modern Statistics and Statistical Machine Learning (StatML) CDT pro-

gramme, grant no. EP/S023151/1. D.C. is supported by PR[AI]RIE-PSAI, a French government grant managed by the Agence Nationale de la Recherche under the France 2030 program. D.C. worked under the auspices of Italian National Group of Mathematical Physics (GNFM) of INdAM.

## Impact Statement

This work develops stochastic optimisation methods for objectives whose gradients are defined as expectations under parameter-dependent distributions with intractable normalising constants. Such problems are fundamental in probabilistic inference, scientific modeling, and control, where principled formulations are often limited by computational constraints. By enabling more reliable optimisation in these settings, the proposed methods can support advances in multiple scientific areas. The work is methodological in nature and does not introduce new application-specific objectives or data sources; consequently, potential societal impacts depend on downstream uses. As with other general-purpose optimization tools, responsible application requires careful modeling, validation, and transparent reporting of approximation errors and limitations.

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

## A. Kernel choices and incremental weight derivations

In this section, we outline the explicit form of the forward and backward kernels, for the SOSMC variants considered throughout the experiments, along with the corresponding incremental weight derivation.

### A.1. Derivation of incremental weights for the ULA kernel

When the forward kernel is chosen as the ULA kernel with step-size $h > 0$:

$$\mathsf{K}_k(x_{k-1}, x_k) = (4\pi h)^{-d/2} \exp\left(-\frac{1}{4h}\|x_k - x_{k-1} + h\nabla U_{\theta_{k-1}}(x_{k-1})\|^2\right),$$

and

$$\mathsf{L}_{k-1}(x_k, x_{k-1}) = (4\pi h)^{-d/2} \exp\left(-\frac{1}{4h}\|x_{k-1} - x_k + h\nabla U_{\theta_k}(x_k)\|^2\right),$$

the incremental weight in (8) becomes

$$
\begin{aligned}
G_k(x_{k-1}, x_k) &= \frac{\Pi_{\theta_k}(x_k)\,\mathsf{L}_{k-1}(x_k, x_{k-1})}{\Pi_{\theta_{k-1}}(x_{k-1})\,\mathsf{K}_k(x_{k-1}, x_k)} \\
&= \exp\left(-U_{\theta_k}(x_k) + U_{\theta_{k-1}}(x_{k-1})\right) \\
&\quad \times \exp\left(-\frac{1}{4h}\left\|x_{k-1} - x_k + h\nabla U_{\theta_k}(x_k)\right\|^2 + \frac{1}{4h}\left\|x_k - x_{k-1} + h\nabla U_{\theta_{k-1}}(x_{k-1})\right\|^2\right).
\end{aligned}
$$

Equivalently, the log-incremental weight is

$$
\begin{aligned}
\log G_k(x_{k-1}, x_k) = {} &-U_{\theta_k}(x_k) + U_{\theta_{k-1}}(x_{k-1}) \\
&- \frac{1}{4h}\left\|x_{k-1} - x_k + h\nabla U_{\theta_k}(x_k)\right\|^2 + \frac{1}{4h}\left\|x_k - x_{k-1} + h\nabla U_{\theta_{k-1}}(x_{k-1})\right\|^2.
\end{aligned}
$$

Expanding the squared norms, we obtain

$$
\begin{aligned}
\log G_k(x_{k-1}, x_k) = {} &-U_{\theta_k}(x_k) + U_{\theta_{k-1}}(x_{k-1}) \\
&- \frac{1}{4h}\left(\|x_{k-1} - x_k\|^2 + h^2\|\nabla U_{\theta_k}(x_k)\|^2 + 2h(x_{k-1} - x_k)^\top \nabla U_{\theta_k}(x_k)\right) \\
&+ \frac{1}{4h}\left(\|x_k - x_{k-1}\|^2 + h^2\|\nabla U_{\theta_{k-1}}(x_{k-1})\|^2 + 2h(x_k - x_{k-1})^\top \nabla U_{\theta_{k-1}}(x_{k-1})\right).
\end{aligned}
$$

To connect it to earlier work (Cuin et al., 2025; Carbone et al., 2023), we rewrite

$$
\begin{aligned}
\log G_k(x_{k-1}, x_k) = {} &-U_{\theta_k}(x_k) - \frac{1}{2}(x_{k-1} - x_k) \cdot \nabla U_{\theta_k}(x_k) - \frac{h}{4}\|\nabla U_{\theta_k}(x_k)\|^2 \\
&+ U_{\theta_{k-1}}(x_{k-1}) + \frac{1}{2}(x_k - x_{k-1}) \cdot \nabla U_{\theta_{k-1}}(x_{k-1}) + \frac{h}{4}\|\nabla U_{\theta_{k-1}}(x_{k-1})\|^2.
\end{aligned}
$$

Define

$$\alpha_{\theta_k}(x, x') = U_{\theta_k}(x) + \frac{1}{2}(x' - x) \cdot \nabla U_{\theta_k}(x) + \frac{h}{4}\|\nabla U_{\theta_k}(x)\|^2,$$

then we get the compact expression for full weights

$$\log W_k = \log W_{k-1} - \alpha_{\theta_k}(x_k, x_{k-1}) + \alpha_{\theta_{k-1}}(x_{k-1}, x_k),$$

which matches the expressions provided in Cuin et al. (2025, Proposition 1) and Carbone et al. (2023, Proposition 1).

## A.2. Derivation of incremental weights for reversible kernels

Next, we consider forward kernels that are invariant, and in fact reversible, with respect to the target $\pi_{\theta_k}$ for the MALA and random-walk Metropolis (RWM) variants, and with respect to the extended target $\widetilde{\pi}_{\theta_k}(x, p)$ for the Hamiltonian Monte Carlo (HMC) variant.

To this end, for the MALA and RWM variants, we let $\mathsf{K}_k$ denote a reversible Markov transition with invariant distribution $\pi_{\theta_k}$, and choose

$$\mathsf{L}_{k-1}(x_k, x_{k-1}) = \frac{\pi_{\theta_k}(x_{k-1})\mathsf{K}_k(x_{k-1}, x_k)}{\pi_{\theta_k}(x_k)} = \mathsf{K}_k(x_k, x_{k-1}),$$

that is, $\mathsf{L}_{k-1}$ is the time reversal of $\mathsf{K}_k$ under $\pi_{\theta_k}$, and, by reversibility, coincides with the same Markov transition run in the reverse direction. To be clear, for Metropolis–Hastings kernels, $\mathsf{K}_k$ denotes the full transition defined by the proposal and accept–reject rule, including the probability of remaining at the current state after rejection. As outlined below, for the HMC variant the same construction is applied on the extended state space.

Substituting this choice of kernel into (8) gives

$$\begin{aligned} G_k(x_{k-1}, x_k) &= \frac{\Pi_{\theta_k}(x_k)\mathsf{L}_{k-1}(x_k, x_{k-1})}{\Pi_{\theta_{k-1}}(x_{k-1})\mathsf{K}_k(x_{k-1}, x_k)} \\ &= \frac{\Pi_{\theta_k}(x_k)}{\Pi_{\theta_{k-1}}(x_{k-1})} \frac{\pi_{\theta_k}(x_{k-1})}{\pi_{\theta_k}(x_k)} \\ &= \frac{\Pi_{\theta_k}(x_{k-1})}{\Pi_{\theta_{k-1}}(x_{k-1})} \\ &= \exp\left(-U_{\theta_k}(x_{k-1}) + U_{\theta_{k-1}}(x_{k-1})\right), \end{aligned}$$

and so the compact expression for full weights is given by

$$\log W_k = \log W_{k-1} - U_{\theta_k}(x_{k-1}) + U_{\theta_{k-1}}(x_{k-1}).$$

Now, we outline the explicit reversible transitions used in the MALA, RWM, and HMC variants.

**MALA kernel**
When the forward kernel is chosen as the MALA transition with step-size $h > 0$, the proposal density is

$$q_{\theta_k}^{\text{MALA}}(x_{k-1}, y_k) = (4\pi h)^{-d/2} \exp\left(-\frac{1}{4h}\left\|y_k - x_{k-1} + h\nabla U_{\theta_k}(x_{k-1})\right\|^2\right),$$

and the proposal $y_k$ is accepted with probability

$$a_{\theta_k}^{\text{MALA}}(x_{k-1}, y_k) = 1 \wedge \frac{\Pi_{\theta_k}(y_k)q_{\theta_k}^{\text{MALA}}(y_k, x_{k-1})}{\Pi_{\theta_k}(x_{k-1})q_{\theta_k}^{\text{MALA}}(x_{k-1}, y_k)}.$$

Thus, the next state is

$$x_k = \begin{cases} y_k, & \text{with probability } a_{\theta_k}^{\text{MALA}}(x_{k-1}, y_k), \\ x_{k-1}, & \text{otherwise.} \end{cases}$$

This proposal–acceptance rule defines the reversible transition $\mathsf{K}_k^{\text{MALA}}$ targeting $\pi_{\theta_k}$.

**RWM kernel**
When the forward kernel is chosen as the RWM transition with step-size $h > 0$, the proposal density is

$$q^{\text{RWM}}(x_{k-1}, y_k) = (4\pi h)^{-d/2} \exp\left(-\frac{1}{4h}\|y_k - x_{k-1}\|^2\right),$$

and the proposal $y_k$ is accepted with probability

$$a_{\theta_k}^{\mathrm{RWM}}(x_{k-1}, y_k) = 1 \wedge \frac{\Pi_{\theta_k}(y_k) q^{\mathrm{RWM}}(y_k, x_{k-1})}{\Pi_{\theta_k}(x_{k-1}) q^{\mathrm{RWM}}(x_{k-1}, y_k)}$$

$$= 1 \wedge \exp\left(-U_{\theta_k}(y_k) + U_{\theta_k}(x_{k-1})\right),$$

where in the second equality we have utilised the symmetry of the proposal density.

Thus, the next state is

$$x_k = \begin{cases} y_k, & \text{with probability } a_{\theta_k}^{\mathrm{RWM}}(x_{k-1}, y_k), \\ x_{k-1}, & \text{otherwise.} \end{cases}$$

This proposal–acceptance rule defines the reversible transition $\mathsf{K}_k^{\mathrm{RWM}}$ targeting $\pi_{\theta_k}$.

**HMC kernel**

For the HMC variant, the invariant transition is defined on the extended state space $(x, p) \in \mathbb{R}^d \times \mathbb{R}^d$. Following the standard construction of Neal (2011), we let $m > 0$ denote the mass parameter and define the extended unnormalised and corresponding normalised target densities as

$$\widetilde{\Pi}_\theta(x, p) = \exp\left(-U_\theta(x) - \frac{1}{2m}\|p\|^2\right), \qquad \widetilde{\pi}_\theta(x, p) = \frac{\widetilde{\Pi}_\theta(x, p)}{\widetilde{Z}_\theta},$$

and the corresponding Hamiltonian as

$$H_\theta(x, p) = U_\theta(x) + \frac{1}{2m}\|p\|^2.$$

At iteration $k$, the HMC transition targets $\widetilde{\pi}_{\theta_k}$ and the momentum is refreshed at the start of the move, so that $p_{k-1} \sim \mathcal{N}(0, mI)$. Given a leapfrog step-size $\epsilon > 0$, one leapfrog step targeting $U_{\theta_k}$ is

$$p_{\ell+1/2} = p_\ell - \frac{\epsilon}{2}\nabla U_{\theta_k}(x_\ell),$$

$$x_{\ell+1} = x_\ell + \frac{\epsilon}{m}p_{\ell+1/2},$$

$$p_{\ell+1} = p_{\ell+1/2} - \frac{\epsilon}{2}\nabla U_{\theta_k}(x_{\ell+1}),$$

and the proposal, obtained after $L$ leapfrog steps, is denoted by

$$(y_k, \rho_k) = \Phi_{\theta_k}^{L, \epsilon}(x_{k-1}, p_{k-1}),$$

and is accepted with probability

$$a_{\theta_k}^{\mathrm{HMC}} = 1 \wedge \exp\left(-H_{\theta_k}(y_k, \rho_k) + H_{\theta_k}(x_{k-1}, p_{k-1})\right).$$

Thus, the next extended state is

$$(x_k, p_k) = \begin{cases} (y_k, -\rho_k), & \text{with probability } a_{\theta_k}^{\mathrm{HMC}}, \\ (x_{k-1}, -p_{k-1}), & \text{otherwise.} \end{cases}$$

Here, the sign change in the momentum is the standard HMC momentum flip, included to make the extended-state transition reversible. This proposal–acceptance rule defines the reversible transition $\mathsf{K}_k^{\mathrm{HMC}}$ targeting $\widetilde{\pi}_{\theta_k}$.

To be clear, the (extended-space) incremental weight is again given by

$$G_k^{\mathrm{HMC}}\left((x_{k-1}, p_{k-1}), (x_k, p_k)\right) = \frac{\widetilde{\Pi}_{\theta_k}(x_{k-1}, p_{k-1})}{\widetilde{\Pi}_{\theta_{k-1}}(x_{k-1}, p_{k-1})}$$

$$= \frac{\exp\left(-U_{\theta_k}(x_{k-1}) - \frac{1}{2m}\|p_{k-1}\|^2\right)}{\exp\left(-U_{\theta_{k-1}}(x_{k-1}) - \frac{1}{2m}\|p_{k-1}\|^2\right)}$$

$$= \exp\left(-U_{\theta_k}(x_{k-1}) + U_{\theta_{k-1}}(x_{k-1})\right).$$

# B. Proofs

We give some additional results on the $\chi^2$-divergence between $\pi_{\theta_k}$ and $\pi_{\theta_{k-1}}$ as a function of the step size $\gamma$ in this section. First we present a result with a result with local boundedness assumptions on the gradient and the Hessian. Recall that

$$\chi^2(\pi_{\theta_k}, \pi_{\theta_{k-1}}) = \int_{x \in \mathbb{R}^{d_x}} \left( \frac{\pi_{\theta_k}(x)}{\pi_{\theta_{k-1}}(x)} - 1 \right)^2 \pi_{\theta_{k-1}}(x) dx,$$

and for a matrix $A$ let $\|A\|_2$ denote its operator norm.

**Proposition 3.** *Let $k \in \mathbb{N}$, assume that for all $x \in E$, $U_\theta(x)$ is second order continuously differentiable in $\theta$ and $\|\nabla \ell(\theta_{k-1})\|_2 = L_k < \infty$. Further assume that $\sup_{x \in E} \sup_{\theta \in B_{\theta_{k-1}}(\gamma L_k)} \|\nabla_\theta U_\theta(x)\|_2 = C_{1,k} < \infty$ and $\sup_{x \in E} \sup_{\theta \in B_{\theta_{k-1}}(\gamma L_k)} \|\nabla_\theta \nabla_\theta^\top U_\theta(x)\|_2 = C_{2,k} < \infty$ then*

$$\chi^2(\pi_{\theta_k}, \pi_{\theta_{k-1}}) = \gamma^2 \nabla \ell(\theta_{k-1})^\top I_{\theta_{k-1}} \nabla \ell(\theta_{k-1}) + O(\gamma^3),$$

*where $I_{\theta_{k-1}} = \mathbb{E}_{\theta_{k-1}} \left[ \nabla_\theta U_{\theta_{k-1}}(X) \nabla_\theta U_{\theta_{k-1}}(X)^\top \right]$.*

*Proof.* First consider a second order Taylor expansion of $U_\theta(x)$ for a fixed value of $x$. let $t \in \mathbb{R}^{d_\theta}$ be an arbitrary vector, then for all $x \in \mathbb{R}^{d_x}$

$$U_{\theta+t}(x) - U_\theta(x) = t^\top \nabla_\theta U_\theta(x) + \frac{t^\top \nabla_\theta \nabla_\theta^\top U_{\tilde{\theta}(x)}(x) t}{2},$$

where the elements of $\tilde{\theta}(x)$ lies on a line segment between $\theta$ and $\theta + t$, but varies depending on the value of $x$ by Taylor's theorem. Then

$$\int_{x \in \mathbb{R}^{d_x}} \left( \frac{\pi_{\theta+t}(x)}{\pi_\theta(x)} - 1 \right)^2 \pi_\theta(x) dx = \int_{x \in \mathbb{R}^{d_x}} \left( \exp \left( -t^\top \nabla_\theta U_\theta(x) - \frac{t^\top \nabla_\theta \nabla_\theta^\top U_{\tilde{\theta}(x)}(x) t}{2} \right) - 1 \right)^2 \pi_\theta(x) dx$$

$$\leq \int_{x \in \mathbb{R}^{d_x}} \left( -t^\top \nabla_\theta U_\theta(x) - \frac{t^\top \nabla_\theta \nabla_\theta^\top U_{\tilde{\theta}(x)}(x) t}{2} \right)^2$$

$$\times \left( \exp \left( \max \left\{ 0, -t^\top \nabla_\theta U_\theta(x) - \frac{t^\top \nabla_\theta \nabla_\theta^\top U_{\tilde{\theta}(x)}(x) t}{2} \right\} \right) \right)^2 \pi_\theta(x) dx$$

where we have use the first order Taylor expansion and noting that $\exp(x) \leq 1 + x \exp(\tilde{x})$ for an $\tilde{x} \in (0, x)$ for $x \geq 0$, and $\tilde{x} \in (-x, 0)$. The above implies that

$$\int_{x \in \mathbb{R}^{d_x}} \left( \frac{\pi_{\theta+t}(x)}{\pi_\theta(x)} - 1 \right)^2 \pi_\theta(x) dx$$

$$\leq \left( \mathbb{E}_\theta \left[ t^\top \nabla_\theta U_\theta(X) \nabla_\theta U_\theta(X)^\top t \right] + \mathbb{E}_\theta \left[ t^\top \nabla_\theta U_\theta(X) t^\top \nabla_\theta \nabla_\theta^\top U_{\tilde{\theta}(X)}(X) t \right] \right.$$

$$\left. + \frac{1}{4} \mathbb{E}_\theta \left[ \left( t^\top \nabla_\theta \nabla_\theta^\top U_{\tilde{\theta}(x)}(X) t \right)^2 \right] \right) \cdot \sup_{x \in E} \exp \left( 2 \left| t^\top \nabla_\theta U_\theta(x) + t^\top \nabla_\theta \nabla_\theta^\top U_{\tilde{\theta}(x)}(x) t \right| \right), \quad (22)$$

where $\mathbb{E}_\theta$ denotes the expectation under $\pi_\theta$. Substitute $t = -\gamma \nabla \ell(\theta_{k-1})$ and $\theta = \theta_{k-1}$, noting that the update rule for $\theta_k$ is $\theta_k = \theta_{k-1} - \gamma \nabla \ell(\theta_{k-1})$, and therefore $\|\theta_k - \theta_{k-1}\|_2 \leq \gamma L_k$. Combining this bound with the boundedness assumptions of the proposition we have

$$(22) \leq \{ \gamma^2 \nabla \ell(\theta_{k-1})^\top I_{\theta_{k-1}} \nabla \ell(\theta_{k-1}) + O(\gamma^3) \} \cdot \exp(2\gamma L_k C_1 + \gamma^2 L_k^2 C_2)$$

$$= \{ \gamma^2 \nabla \ell(\theta_{k-1})^\top I_{\theta_{k-1}} \nabla \ell(\theta_{k-1}) + O(\gamma^3) \} \cdot (1 + O(\gamma))$$

$$= \gamma^2 \nabla \ell(\theta_{k-1})^\top I_{\theta_{k-1}} \nabla \ell(\theta_{k-1}) + O(\gamma^3),$$

where $I_{\theta_{k-1}}$ is the Fisher information at $\theta_{k-1}$. Finally the equality in the statement of the proposition is due to the fact that the gap in the upper bound is of order $O(\gamma^3)$. $\qquad \square$

The above proposition's boundedness assumption may be too strong for many models. We give the following proposition which shows a worse rate of decay for the $\chi^2$-divergence between $\pi_k$ and $\pi_{k-1}$ in $\gamma$, but with weaker assumptions. The assumptions in the following proposition are similar to the local dominating conditions used in classical asymptotic statistics (Van der Vaart, 1998, Chapter 5), the key difference being that here we require exponential moments to be finite.

**Proposition 4.** *Let $k \in \mathbb{N}$, assume for all $x \in E$ that $U_\theta(x)$ is second order continuously differentiable in $\theta$ and $\|\nabla \ell(\theta_{k-1})\|_2 = L_k < \infty$. Further assume that there exist vector and matrix functions $S_{1,k}(x)$ and $S_{2,k}(x)$ such that $\forall_{x \in E} \sup_{\theta \in B_{\theta_{k-1}}(\gamma L_k)} |\nabla_\theta U_\theta(x)| \leq S_{1,k}(x)$ and $\forall_{x \in E} \sup_{\theta \in B_{\theta_{k-1}}(\gamma L_k)} \nabla_\theta \nabla_\theta^\top U_\theta(x) \preceq S_{2,k}(x)$, where*

$$\mathbb{E}_{\theta_{k-1}}[\exp(\lambda^\top S_{1,k}(X))] < \infty \text{ and}$$
$$\mathbb{E}_{\theta_{k-1}}[\exp(\lambda^\top S_{2,k}(X))\lambda] < \infty,$$

*for $\lambda \in \mathbb{R}^{d_\theta}$ in a neighbourhood of $0$. Then*

$$\chi^2(\pi_{\theta_k}, \pi_{\theta_{k-1}}) = \sqrt{\gamma^2 \nabla \ell(\theta_{k-1})^\top I_{\theta_{k-1}} \nabla \ell(\theta_{k-1})} + O(\gamma^{3/2}),$$

*where $I_{\theta_{k-1}} = \mathbb{E}_{\theta_{k-1}} \left[ \nabla_\theta U_{\theta_{k-1}}(x) \nabla_\theta U_{\theta_{k-1}}(x)^\top \right].$*

*Proof.* Consider a second order Taylor expansion of $U_\theta(x)$, let $t \in \mathbb{R}^{d_\theta}$ be an arbitrary vector, then for all $x \in \mathbb{R}^{d_x}$

$$U_{\theta+t}(x) - U_\theta(x) = t^\top \nabla_\theta U_\theta(x) + \frac{t^\top \nabla_\theta \nabla_\theta^\top U_{\tilde{\theta}(x)}(x)t}{2},$$

where the elements of $\tilde{\theta}(x)$ lies on a line segment between $\theta$ and $\theta + t$, consider

$$\int_{x \in \mathbb{R}^{d_x}} \left( \frac{\pi_{\theta+t}(x)}{\pi_\theta(x)} - 1 \right)^2 \pi_\theta(x)dx \leq \int_{x \in \mathbb{R}^{d_x}} \left| \frac{\pi_{\theta+t}(x)}{\pi_\theta(x)} - 1 \right| \pi_\theta(x)dx$$

$$= \int_{x \in \mathbb{R}^{d_x}} \left| \exp\left( -t^\top \nabla_\theta U_\theta(x) - \frac{t^\top \nabla_\theta \nabla_\theta^\top U_{\tilde{\theta}(x)}(x)t}{2} \right) - 1 \right| \pi_\theta(x)dx$$

$$\leq \int_{x \in \mathbb{R}^{d_x}} \left| t^\top \nabla_\theta U_\theta(x) + \frac{t^\top \nabla_\theta \nabla_\theta^\top U_{\tilde{\theta}(x)}(x)t}{2} \right|$$

$$\times \exp\left( \max\left\{ 0, -t^\top \nabla_\theta U_\theta(x) - \frac{t^\top \nabla_\theta \nabla_\theta^\top U_{\tilde{\theta}(x)}(x)t}{2} \right\} \right) \pi_\theta(x)dx$$

$$\leq \sqrt{\int_{x \in \mathbb{R}^{d_x}} \left( t^\top \nabla_\theta U_\theta(x) + \frac{t^\top \nabla_\theta \nabla_\theta^\top U_{\tilde{\theta}(x)}(x)t}{2} \right)^2 \pi_\theta(x)dx}$$

$$\times \sqrt{\int_{x \in \mathbb{R}^{d_x}} \exp\left( \max\left\{ 0, -2t^\top \nabla_\theta U_\theta(x) - t^\top \nabla_\theta \nabla_\theta^\top U_{\tilde{\theta}(x)}(x)t \right\} \right) \pi_\theta(x)dx},$$

by Cauchy-Schwarz inequality. Now let $t = -\gamma \nabla \ell(\theta_{k-1})$ and $\theta = \theta_{k-1}$, for the first term, by direct computation we have that

$$\sqrt{\int_{x \in \mathbb{R}^{d_x}} \left( t^\top \nabla_\theta U_\theta(x) + \frac{t^\top \nabla_\theta \nabla_\theta^\top U_{\tilde{\theta}(x)}(x)t}{2} \right)^2 \pi_\theta(x)dx} = \sqrt{\gamma^2 \nabla \ell(\theta_{k-1})^\top I_{\theta_{k-1}} \nabla \ell(\theta_{k-1})} + O(\gamma^{3/2}),$$

as the score and the hessian are locally dominated by functions $S_{1,k}(x)$ and $S_{2,k}(x)$ for all $k \in \mathbb{N}$ such that $E_{\theta_{k-1}}[\nabla \ell^\top S_{1,k}(X)] < \infty$ and $E_{\theta_{k-1}}[\nabla \ell(\theta_{k-1})^\top S_{2,k}(X)\nabla \ell(\theta_{k-1})] < \infty$ (if exponential moments exist, then all moments exist and by assumption $\|\nabla \ell(\theta_{k-1})\| = L_k < \infty$). As for the other term by Cauchy-Swartz's inequality

$$\sqrt{\int_{x \in \mathbb{R}^{d_x}} \exp\left( \max\left\{ 0, -2\gamma \nabla \ell(\theta_{k-1})^\top \nabla_\theta U_\theta(x) - \gamma^2 \nabla \ell(\theta_{k-1})^\top \nabla_\theta \nabla_\theta^\top U_{\tilde{\theta}(x)}(x)\nabla \ell(\theta_{k-1}) \right\} \right) \pi_\theta(x)dx}$$

$$\leq \left( \int_{x \in \mathbb{R}^{d_x}} \exp\left( 4\gamma \nabla \ell(\theta_{k-1})^\top S_1(x) \right) \pi_\theta(x)dx \right)^{1/4} \cdot \left( \int_{x \in \mathbb{R}^{d_x}} \exp\left( 2\gamma^2 \nabla \ell(\theta_{k-1})^\top S_2(x) \nabla \ell(\theta_{k-1}) \right) \pi_\theta(x)dx \right)^{1/4},$$

as $\gamma \to 0$, $2\gamma\nabla\ell(\theta_{k-1})$ will eventually lie in a neighbourhood of 0, therefore by the absolute convergence of the power series of $\exp(x)$ and the existence of exponential moments assumption

$$\int_{x\in\mathbb{R}^{d_x}} \exp\left(4\gamma\nabla\ell(\theta_{k-1})^\top S_{1,k}(x)\right) \pi_\theta(x)dx = 1 + O(\gamma)$$

$$\int_{x\in\mathbb{R}^{d_x}} \exp\left(2\gamma^2\nabla\ell(\theta_{k-1})^\top S_{2,k}(x)\nabla\ell(\theta_{k-1})\right) \pi_\theta(x)dx = 1 + O(\gamma),$$

combining all of these estimates shows the desired result. $\qquad\square$

## C. Gradient Estimation for Reward Tuning

Let $\pi_\theta$ denote the parametric probability distribution

$$\pi_\theta(x) = \frac{\exp(-U_\theta(x))}{Z_\theta}, \qquad Z_\theta = \int_{\mathbb{R}^{d_x}} \exp(-U_\theta(x))\, dx,$$

as previously introduced in (1). Additionally, let $R : \mathbb{R}^{d_x} \to \mathbb{R}$ denote a (potentially non-differentiable) reward function, whilst $\pi_0$ denotes a fixed reference distribution with energy $U_0$.

Throughout reward tuning, we consider loss objectives of the form

$$\ell(\theta) = -\mathbb{E}_{\pi_\theta}[R(X)] + \beta_{\mathrm{KL}}\mathrm{D}_{\mathrm{KL}}(\cdot\|\cdot), \tag{23}$$

where $\mathrm{D}_{\mathrm{KL}}(\cdot\|\cdot)$ may denote either the forward or reverse KL respectively. Indeed, the forward variant is natural in cases when direct access to $\pi_0$ is available, whilst the reverse variant is natural when $\pi_0$ is defined implicitly.

In either case, the respective gradient is compatible with (2), where, through following the derivations outlined in Appendix F.1, we arrive at the following form for the forward KL variant,

$$\nabla_\theta \overrightarrow{\ell}(\theta) = (\mathbb{E}_{\pi_\theta}[R(x)\nabla_\theta U_\theta(x)] - \mathbb{E}_{\pi_\theta}[R(x)]\mathbb{E}_{\pi_\theta}[\nabla_\theta U_\theta(x)]) + \beta_{\mathrm{KL}}(\mathbb{E}_{\pi_0}[\nabla_\theta U_\theta(x)] - \mathbb{E}_{\pi_\theta}[\nabla_\theta U_\theta(x)]).$$

whilst for the reverse KL variant we follow Appendix F.2, to obtain the form

$$\nabla_\theta \overleftarrow{\ell}(\theta) = \mathbb{E}_{\pi_\theta}[A_\theta(x)\nabla_\theta U_\theta(x)] - \mathbb{E}_{\pi_\theta}[A_\theta(x)]\,\mathbb{E}_{\pi_\theta}[\nabla_\theta U_\theta(x)],$$

where $A_\theta(x) = R(x) + \beta_{\mathrm{KL}}(U_\theta(x) - U_0(x))$.

As outlined in Section 3, at a high level, SOSMC maintains a weighted particle population $\left\{X_k^{(i)}, w_k^{(i)}\right\}$, with $w_k^{(i)} \geq 0$ and $\sum_{i=1}^N w_k^{(i)} = 1$, where $k$ denotes the index of the outer iteration. The gradient is then approximated via this weighted particle population,

$$g_k = \sum_{i=1}^N w_k^{(i)} H_{\theta_k}(X_k^{(i)}),$$

where in the case of the forward KL variant we specifically have

$$\overrightarrow{g_k} = \sum_{i=1}^N w_k^{(i)} R(X_k^{(i)})\nabla_\theta U_{\theta_k}(X_k^{(i)}) - \left(\sum_{i=1}^N w_k^{(i)} R(X_k^{(i)})\right)\left(\sum_{i=1}^N w_k^{(i)}\nabla_\theta U_{\theta_k}(X_k^{(i)})\right)$$

$$+ \beta_{\mathrm{KL}}\left(\frac{1}{M}\sum_{j=1}^M \nabla_\theta U_{\theta_k}(X_{0,k}^{(j)}) - \sum_{i=1}^N w_k^{(i)}\nabla_\theta U_{\theta_k}(X_k^{(i)})\right), \tag{24}$$

where $\{X_{0,k}^{(j)}\}_{j=1}^M$ denotes an independent reference batch, drawn from $\pi_0$, at outer iteration index $k$.

In contrast, for the reverse KL variant we have

$$
\begin{aligned}
\overleftarrow{g_k} =& \sum_{i=1}^{N} w_k^{(i)} R(X_k^{(i)}) \nabla_\theta U_{\theta_k}(X_k^{(i)}) - \left( \sum_{i=1}^{N} w_k^{(i)} R(X_k^{(i)}) \right) \left( \sum_{i=1}^{N} w_k^{(i)} \nabla_\theta U_{\theta_k}(X_k^{(i)}) \right) \\
&+ \beta_{\mathrm{KL}} \left( \sum_{i=1}^{N} w_k^{(i)} \left( U_{\theta_k}(X_k^{(i)}) - U_0(X_k^{(i)}) \right) \nabla_\theta U_{\theta_k}(X_k^{(i)}) \right) \\
&- \beta_{\mathrm{KL}} \left( \sum_{i=1}^{N} w_k^{(i)} \left( U_{\theta_k}(X_k^{(i)}) - U_0(X_k^{(i)}) \right) \right) \left( \sum_{i=1}^{N} w_k^{(i)} \nabla_\theta U_{\theta_k}(X_k^{(i)}) \right).
\end{aligned}
\tag{25}
$$

Notably, in the case of IMPDIFF, gradient estimates correspond to the above case with $w_k^{(i)} = 1/N$. Although SOUL also utilises uniform weights, the manner in which expectations are constructed differs, where a time average of a single chain is instead utilised, so that

$$
\widehat{\mathbb{E}}_k^{\mathrm{SOUL}}[\varphi(X)] = \frac{1}{N_{\mathrm{eff}}} \sum_{j=1}^{N_{\mathrm{eff}}} \varphi(X_{k,j}),
$$

whilst the reference batch, in the case of the forward KL variant, is handled identically.

As emphasised in Appendix F.3, particles and weights are treated as *stop-gradient* quantities throughout. Furthermore, *surrogate* losses are in practice utilised due to specifics related to current automatic differentiation frameworks, for which we again refer to Appendix F.3.

## D. Algorithms

---

**Algorithm 2** SOSMC-ULA for reward tuning a pretrained EBM (forward or reverse KL)

---

**Require:** Frozen pre-trained $E_{\theta_0}$, reward $R(\cdot)$, number of particles $N$, outer iterations $K$, step size $\gamma$, noise scale $\sigma$, regularisation coefficient $\beta_{\mathrm{KL}}$, ESS threshold $\tau$, optimiser OPT.

Initialise trainable model $E_\theta$ at $\theta_0$, particles $\{X_0^{(i)}\}_{i=1}^{N} \sim \pi_0$, and weights $W_0^{(i)} = 1$.

*Define:* $\alpha_\theta(x \to x'; \sigma) = E_\theta(x) + \frac{1}{2\sigma^2}(x' - x)^\top \nabla_x E_\theta(x) + \frac{\gamma}{4\sigma^2} \|\nabla_x E_\theta(x)\|^2$     *// treated as constant in $\nabla_\theta$*

**for** $k = 1, \ldots, K - 1$ **do**

    $w_{k-1}^{(i)} = W_{k-1}^{(i)} / \sum_{j=1}^{N} W_{k-1}^{(j)}$

    Compute $g_{k-1}(w_{k-1})$ as in (24) for forward-KL, or as in (25) for reverse-KL.

    $\theta_k = \mathrm{OPT}(\theta_{k-1}, g_{k-1}(w_{k-1}))$

    **for** $i = 1, \ldots, N$ **do**

        $X_k^{(i)} = X_{k-1}^{(i)} - \gamma \nabla_x E_{\theta_{k-1}}(X_{k-1}^{(i)}) + \sqrt{2\gamma}\, \sigma\, \varepsilon_{k-1}^{(i)},$     where $\varepsilon_{k-1}^{(i)} \sim \mathcal{N}(0, I)$.

        $\alpha_{k-1}^{(i)} = \alpha_{\theta_{k-1}}(X_{k-1}^{(i)} \to X_k^{(i)}; \sigma)$

        $\alpha_k^{(i)} = \alpha_{\theta_k}(X_k^{(i)} \to X_{k-1}^{(i)}; \sigma)$

        $\log W_k^{(i)} = \log W_{k-1}^{(i)} - \alpha_k^{(i)} + \alpha_{k-1}^{(i)}$

    **end for**

    $\log W_k = \log W_k - \max_i \log W_k^{(i)}.$     *// recenter weights*

    Resample $\{X_k^{(i)}\}_{i=1}^{N}$ w.r.t $w_k^{(i)}$ and set $W_k^{(i)} = 1$ if ESS $< \tau_{\mathrm{res}} N$.

**end for**

**Return** tuned parameters $\theta_K$, and optionally history of other quantities.

---

---

**Algorithm 3** IMPDIFF for reward tuning a pretrained EBM (forward or reverse KL)

---

**Require:** Frozen pre-trained $E_{\theta_0}$, reward $R(\cdot)$, number of particles $N$, outer iterations $K$, step size $\gamma$, noise scale $\sigma$, regularisation coefficient $\beta_{\mathrm{KL}}$, optimiser OPT.

Initialise trainable model $E_\theta$ at $\theta_0$, particles $\{X_0^{(i)}\}_{i=1}^N \sim \pi_0$.

**for** $k = 1, \ldots, K-1$ **do**

    $w_{k-1}^{(i)} = 1/N$

    Compute $g_{k-1}(w_{k-1})$ as in (24) for forward-KL, or as in (25) for reverse-KL.

    $\theta_k = \mathrm{OPT}(\theta_{k-1}, g_{k-1}(w_{k-1}))$

    **for** $i = 1, \ldots, N$ **do**

        $X_k^{(i)} = X_{k-1}^{(i)} - \gamma \nabla_x E_{\theta_{k-1}}(X_{k-1}^{(i)}) + \sqrt{2\gamma}\,\sigma\,\varepsilon_{k-1}^{(i)}$,      where $\varepsilon_{k-1}^{(i)} \sim \mathcal{N}(0, I)$.

    **end for**

**end for**

**Return** tuned parameters $\theta_K$, and optionally history of other quantities.

---

# E. Experimental Details

### E.1. Reward tuning of Langevin processes

Here, we consider the reward tuning of a *stationary Langevin sampler*, whose invariant distribution is a Gaussian mixture with learnable mixture weights. This setup mirrors that of Section 5.1 in Marion et al. (2025), which we utilise to compare their proposed method, termed throughout as IMPDIFF, with our SOSMC method, for a variety of kernel choices, as well as a nested inner-loop baseline method, specifically that of SOUL.

**Setup.** In accordance with Marion et al. (2025), we parameterise a family of potentials $V(\cdot, \theta) : \mathbb{R}^2 \to \mathbb{R}$, with parameters $\theta \in \mathbb{R}^m$, of the form

$$V(x, \theta) = -\log\left(\sum_{i=1}^m \sigma(\theta)_i \exp\left(-\frac{\|x - \mu_i\|_2^2}{2\sigma^2}\right)\right), \tag{26}$$

where $\sigma(\theta)$ denotes the softmax mapping $\mathbb{R}^m$ to the probability simplex, while $\mu_i \in \mathbb{R}^2$ for $i = 1, \ldots, m$ denote fixed component means, and $\sigma^2 > 0$ is a fixed variance parameter.

The corresponding stationary distribution is the Gibbs measure,

$$\pi_\theta(x) \propto \exp(-V(x, \theta)), \tag{27}$$

which we note is a mixture of isotropic Gaussians, with means $\mu_i$ and common variance proportional to $\sigma^2$. As noted in Marion et al. (2025), the normalising constant does not depend on $\theta$ for (26).

Given a reward function $R : \mathbb{R}^2 \to \mathbb{R}$, and a fixed reference distribution $\pi_{\mathrm{ref}}$, we optimise the KL-regularised objective, as in Gutmann & Hyvärinen (2012) and Marion et al. (2025), where

$$\ell(\theta) = -\mathbb{E}_{X \sim \pi_\theta}[R(X)] + \beta_{\mathrm{KL}}\, \mathrm{KL}(\pi_{\mathrm{ref}} \,\|\, \pi_\theta), \qquad \beta_{\mathrm{KL}} > 0, \tag{28}$$

with $\pi_\theta$ denoting the Gibbs distribution induced by the parameterised potential $V(\cdot, \theta)$. The reference distribution $\pi_{\mathrm{ref}}$ is taken to be a fixed Gibbs distribution, of the same functional form, induced by the frozen parameter $\theta_{\mathrm{ref}}$, from which independent samples are readily available.

**Reference Distribution.** Essentially, $\pi_{\mathrm{ref}}$ acts as a fixed baseline, against which deviations induced by reward tuning are penalised. Since the potential in (26) corresponds to a Gaussian mixture model, sampling from $\pi_{\mathrm{ref}}(x) \propto \exp(-V(x, \theta_{\mathrm{ref}}))$ is both exact and computationally inexpensive. In fact, a sample $X_{\mathrm{ref}} \sim \pi_{\mathrm{ref}}$ is generated by simply drawing a component index, followed by sampling from the corresponding isotropic Gaussian component,

$$I \sim \sigma(\theta_{\mathrm{ref}}), \qquad X_{\mathrm{ref}} = \mu_I + \sqrt{\sigma^2}\,\varepsilon, \qquad \varepsilon \sim \mathcal{N}(0, I_2). \tag{29}$$

Indeed, $\mathrm{KL}(\pi_{\mathrm{ref}}\|\pi_\theta)$ penalises deviations from $\pi_{\mathrm{ref}}$ such that regions which possess non-negligible probability under $\pi_{\mathrm{ref}}$ cannot be assigned insignificant probability under $\pi_\theta$, at least without incurring a large penalty.

**Reward Functions.** Regarding the reward functions considered, we consider a collection designed to examine the behaviour of reward tuning under non-smoothness, gating constraints, and even disconnected high-reward regions. For example, following Marion et al. (2025), we consider a gated reward, as well as its smoothed version, such as

$$R_{\text{hard}}(x) = \mathbf{1}_{\{x_1 \geq 0\}} \exp\left(-\|x - c\|_2^2 / \tau\right), \qquad R_{\text{smooth}}(x) = \sigma(x_1/\lambda) \exp\left(-\|x - c\|_2^2 / \tau\right), \tag{30}$$

where we note $R_{\text{smooth}}$ remains differentiable no matter the transition sharpness, as determined by $\lambda > 0$. Furthermore, a multi-modal reward, of the form

$$R_{\text{multi}}(x) = \max_{j=1,\dots,J} \exp\left(-\|x - c_j\|_2^2 / \tau\right), \tag{31}$$

is also considered. Since $\nabla_\theta \ell$ does not depend on derivatives of $R$ here, as outlined in Appendix F.1, the optimisation of (28) is well defined for all the reward functions outlined above.

**Tuning Procedure.** The tuning procedure begins by first initialising the persistent particle(s) $X_0^{(i)} \sim \pi_{\text{ref}}$, which are then propagated for a pre-specified number of outer iterations, $K$, through applying a chosen Markov kernel, $\mathsf{K}_{\theta_k}$. In this case we consider the ULA transition as a baseline, as in Marion et al. (2025),

$$X_{k+1}^{(i)} = X_k^{(i)} - \gamma_k \nabla_x V(X_k^{(i)}, \theta_k) + \sqrt{2\gamma_k}\, \xi_k^{(i)}, \qquad \xi_k^{(i)} \sim \mathcal{N}(0, I), \tag{32}$$

with $\gamma_k$ always fixed in the cases of IMPDIFF and SOUL. To highlight the flexibility of kernel choice within SOSMC, we consider not only SOSMC-ULA, but also SOSMC-MALA, SOSMC-RWM, and SOSMC-HMC, as discussed in Appendix A. Indeed, all methods considered utilise the resulting evolution of the persistent particle(s) to construct Monte Carlo estimates of the loss, whose gradient matches $\nabla\ell(\theta_k)$, as outlined in Appendix F.1, however it is worth noting again that the key difference across the method lies in how these estimates are formed. In the case of both IMPDIFF and SOSMC-ULA, a collection of particles $\{X_k^{(i)}\}_{i=1}^N$ are evolved by a single ULA step per outer iteration, where, in the latter method, weights are utilised within the respective Monte Carlo estimates, as detailed in Appendix C. For the remaining SOSMC variants, the particles are instead propagated by a single step under the respective kernel and weighted according to the corresponding incremental weights, as derived in Appendix A.2. In contrast, for the case of SOUL, a single ULA chain of length $N$ is run, with the last $N_{\text{eff}} = N - n_{\text{burn}}$ samples utilised in the respective Monte Carlo estimates. Nevertheless, in all methods, the respective particles are treated as *stop-gradient* quantities, that is we take care not to backpropagate through the Markov kernel $\mathsf{K}_{\theta_k}$, for which we refer to Appendix F.3 for a detailed discussion.

**Implementation Details.** A significant focus of our implementation is that of fairness and optimality from a compute perspective. To provide meaningful comparison, we ensure the number of computations, such as gradient evaluations, per outer step are minimised, whilst also measuring the wall-clock time of the methods in a device-synchronised manner. We implement the various methods using `JAX`, such that *just-in-time compiling* is leveraged, as are batched particle operations. Notably, `JAX` utilises asynchronous dispatch, meaning naive timing may significantly underestimate true runtimes by failing to measure true device execution times, as the dispatch overhead is instead measured. To avoid this, we purposely utilise a blocking call so that the actual compute time is indeed measured.

Regarding the wall-clock runtime to complete $K$ outer iterations, IMPDIFF is consistently the fastest method, as highlighted for a specific example in Figure A.2, which is to be expected in light of IMPDIFF requiring only a single (batched) evaluation of $\nabla_x V$ per outer iteration, along with minimal supplementary computation. Indeed, a potential drawback of the weights

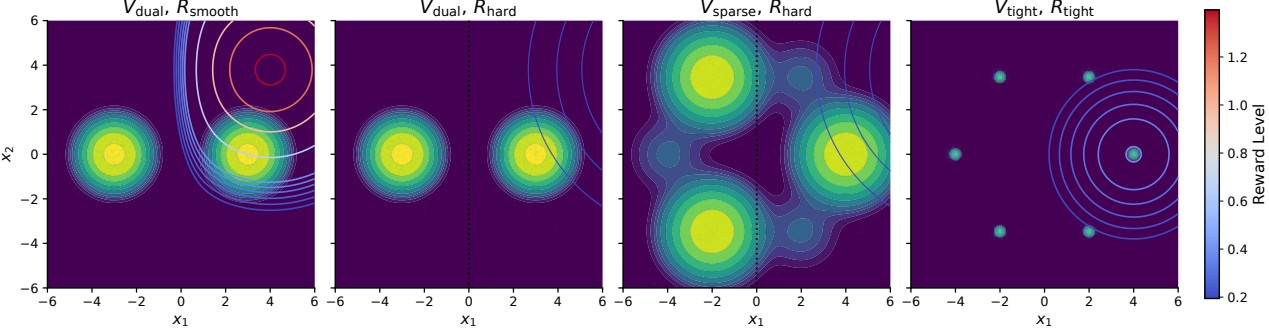

*Figure A.1.* Reference distributions and reward functions considered, where brighter regions indicate higher probability.

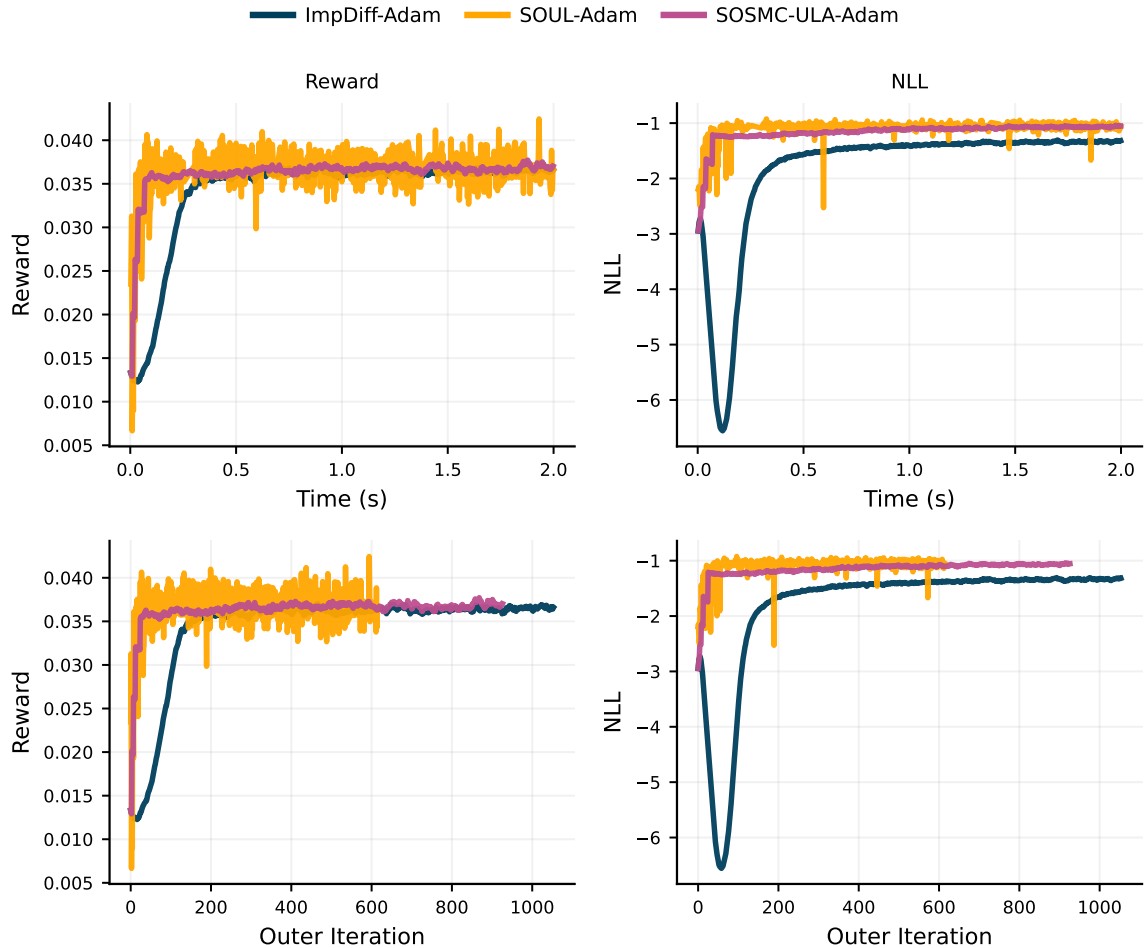

*Figure A.2.* Wall-clock (top) and outer iteration (bottom) convergence of reward (left) and NLL (right), for a single run, for IMPDIFF, SOUL, and SOSMC-ULA, under $V_{\text{sparse}}$ & $R_{\text{hard}}$, with $\beta_{\text{KL}} = 0$. A fixed wall-clock runtime of 2 seconds is specified in all cases.

utilised within SOSMC is the additional computation it can incur. For example, in the case of SOSMC-ULA, two (batched) evaluations of $\nabla_x V$ per outer iteration are required, along with an increased amount of supplementary computation due to weight normalisation and ESS bookkeping. In contrast, SOUL's inner ULA update operates on a single particle in low dimensions, and is thus computationally comparative in low $N$ regimes, particularly those in which the speed-ups obtained through parallelisation have not yet come into effect. Specifically, as $N$ increases, the sequential nature of SOUL leads to increased wall-clock runtimes, compared to IMPDIFF and SOSMC.

Notably, the $\nabla_x V$ computations are only part of the overall computation per outer iteration, since $\nabla_\theta V$ and supplementary computations are also required, and so the number of $\nabla_x V$ computations does not match in a straightforward manner with wall-clock runtimes, even if the process through which these are computed were the same. In light of this, for fair comparison, we choose to ensure instead that the number of times $\mathsf{K}_{\theta_k}$ is applied per outer iteration is equal across methods. Indeed, this leads us to utilise $N$ particles for IMPDIFF and SOSMC, whilst running the chain for $N$ steps in total for SOUL.

In particular, we utilise the SOSMC-ULA, SOSMC-MALA, SOSMC-RWM, SOSMC-HMC variants of our method here, with $\gamma_k$ initialised to $0.1$ across all methods. For each method, various choices of OPT are considered, albeit with a common learning rate of $\eta = 0.1$. Regarding the reference batch, $\{X_{\text{ref}}^{(j)}\}_{j=1}^{N_{\text{ref}}}$ is refreshed every outer iteration, with $N_{\text{ref}} = 5,000$, whilst $N = 10,000$ unless otherwise stated. Furthermore, we remark that for SOUL, we have that $n_{\text{burn}} = N/2$, so that also $N_{\text{eff}} = N/2$, whilst specific to SOSMC methods, we choose an ESS threshold of $\tau = 0.9$ and an optional adaptive threshold of $\tau = 0.95$. In order to keep per-iteration computational budget comparable, we set the number of HMC leapfrog steps to $L = 1$, and also note we set the mass to $m = 1$.

**Results.** In accordance with Marion et al. (2025), we report both the reward and trajectory of the negative log-likelihood (NLL), of the particle collection, at each outer iteration, averaged over the particles. In the interest of speed, and since convergence occurs relatively quickly, we limit wall-clock runtimes to 2.0 seconds. Qualitatively, as corroborated by Figure A.2, we observe the SOSMC variants to consistently converge to a high reward quickly, whilst IMPDIFF does so more slowly. Furthermore, in such scenarios, IMPDIFF often exhibits a more significant transient phase, and in fact stabilises at a lower NLL value. Indeed, re-weighting and long inner trajectories, present in SOSMC variants and SOUL respectively, provide gradient estimates that result in such quick convergence, in contrast to the unweighted estimates of IMPDIFF, that are driven by a single-step Markov kernel transition.

Additionally, we compare different choices of the first-order optimiser OPT, for each method, across multiple random seeds, for which the setting of $V_{\text{dual}}$ equipped with $R_{\text{smooth}}$, as displayed in Figure A.3, illustrates general behaviour. In

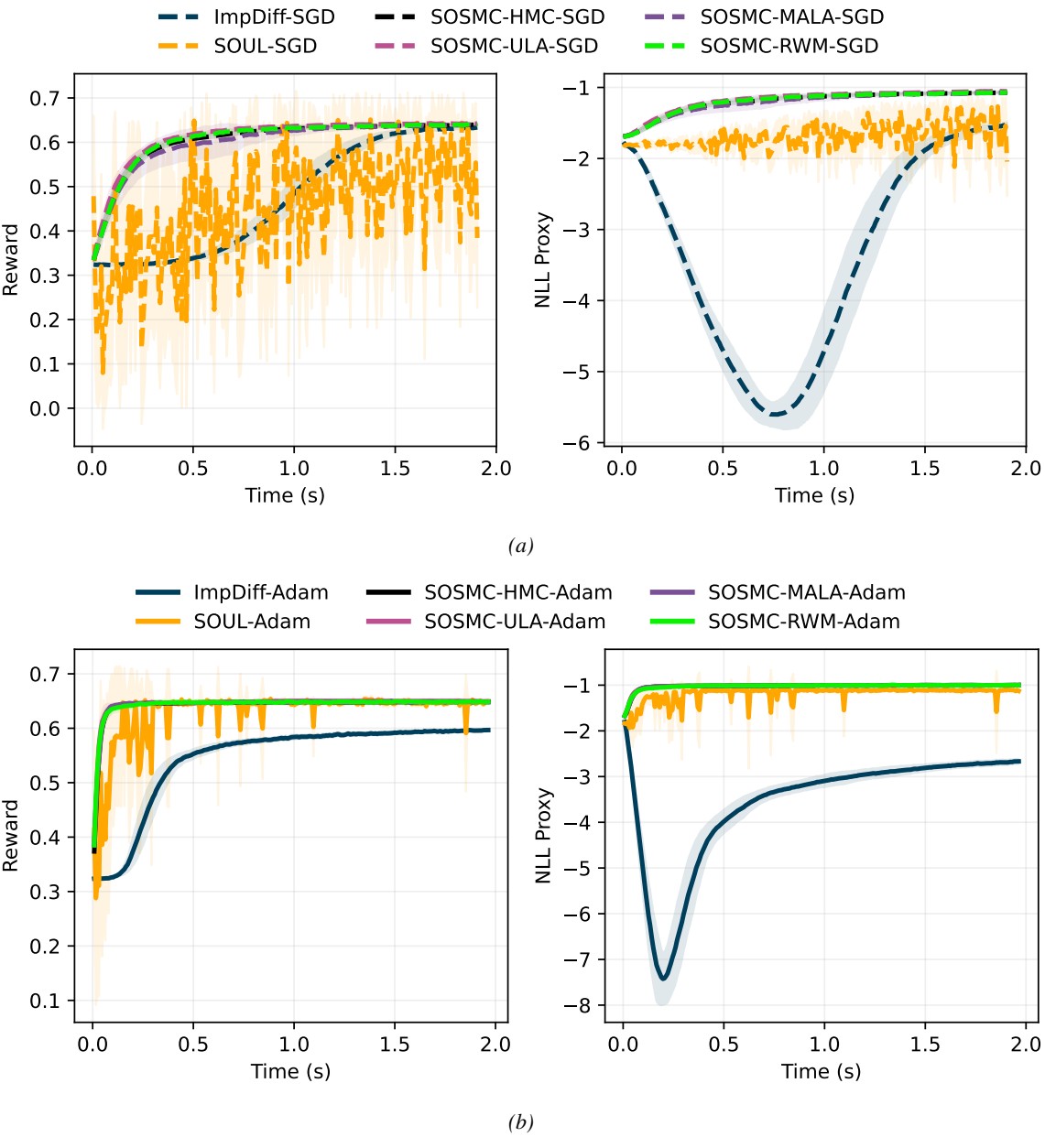

*Figure A.3.* Convergence of mean reward (left) and NLL (right), for IMPDIFF, SOUL, and SOSMC for $V_{\text{dual}}$ & $R_{\text{smooth}}$, 10 runs. In *(a)* OPT is SGD, whilst in *(b)* OPT is Adam.

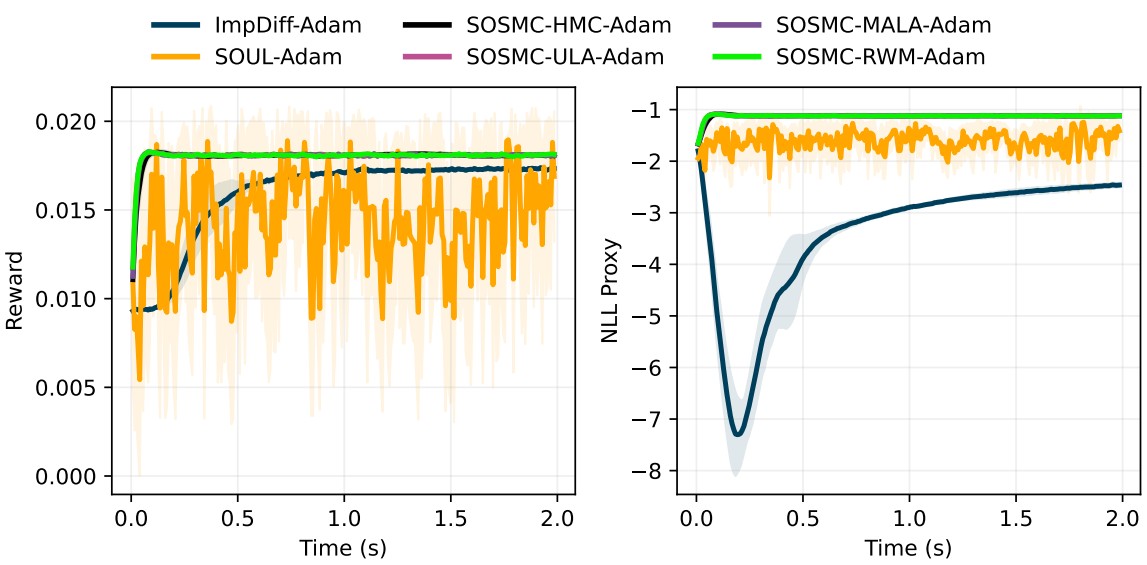

*Figure A.4.* Convergence of mean reward (left) and NLL (right), for IMPDIFF, SOUL, and SOSMC for $V_{\text{dual}}$ & $R_{\text{hard}}$, 10 runs.

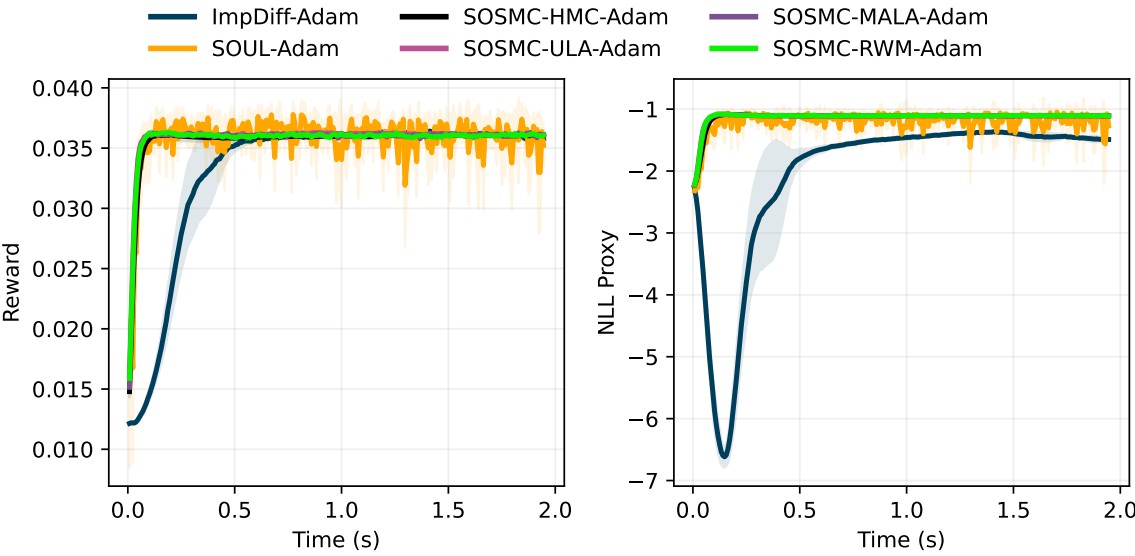

*Figure A.5.* Convergence of mean reward (left) and NLL (right), for IMPDIFF, SOUL, and SOSMC for $V_{\text{sparse}}$ & $R_{\text{hard}}$, 10 runs.

order to compute the mean and respective confidence intervals for each curve, for each experimental trial, the time axis is divided into a number of equally spaced points, for which the most recently logged reward and NLL value is associated with said time point, for the respective trial, which ensures we do not peek into the future.

First, we observe ADAM to consistently accelerate early stage progress across methods, particularly with respect to the initial increase in the reward. This does not, however, necessarily result in improved final performance, as observed in Figure A.3, for IMPDIFF. We attribute this to the fact that potentially aggressive adaptive preconditioning of ADAM may increase transient gradient noise present as a result of the bias induced by finite step sampling, which can in turn result in premature stabilisation due to conservative effective step sizes later on in the tuning procedure. On the other hand, SGD does not retain a memory of the (early) gradient magnitudes and so does not face the same issue of early stabilisation.

Although the impact of OPT is important, the dominant effect is that of the use of weighted SOSMC particle approximations. Indeed, for SOSMC variants, particles that reach high-reward regions, even if initially rare, disproportionately affect the $\theta$

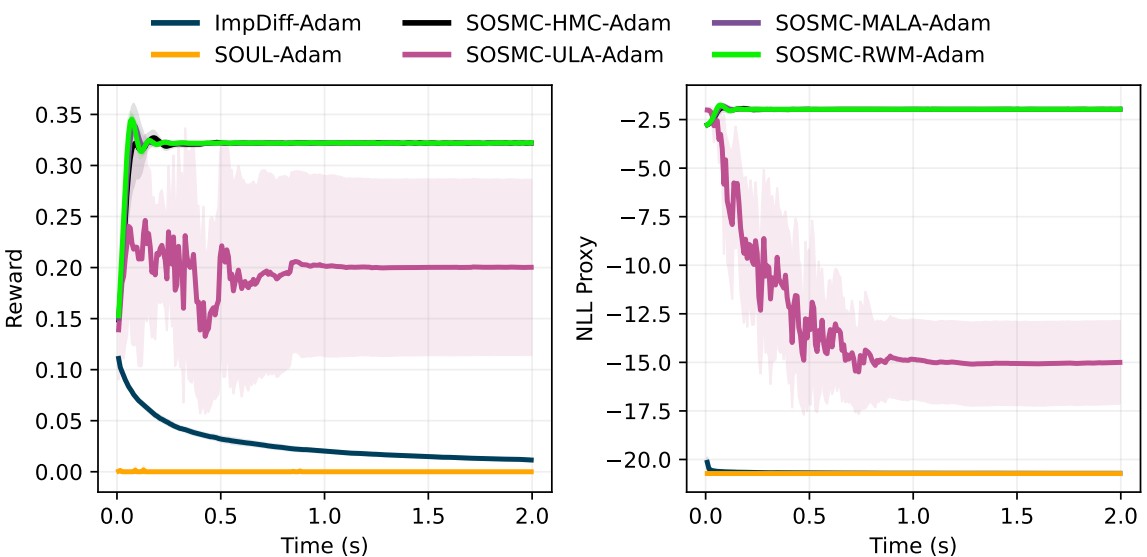

*Figure A.6.* Convergence of mean reward (left) and NLL (right), for IMPDIFF, SOUL, and SOSMC for $V_{\text{tight}}$ & $R_{\text{tight}}$, 10 runs.

updates, due to the respective re-weighting, whereas IMPDIFF relies on an unweighted single-step persistent-particle estimate. On the other hand, SOUL benefits from averaging over a long inner trajectory, provided sufficient mixing. The reliance on a single trajectory, however, explains the relatively large variability between runs, particularly for SGD, compared to say the SOSMC variants. In order to highlight the importance of kernel choice, and to outline a setting in which OPT alone cannot overcome limitations of the underlying method, we consider the setting of $V_{\text{tight}}$ equipped with $R_{tight}$, which note is displayed in Figure A.1. In this case, the reward-supporting region is narrow and constrained to a single mode, meaning that successful optimisation is dependent upon both sufficient exploration and accurate exploitation of the particles that reach said region. As illustrated in Figure A.6, SOSMC variants that possess a Metropolis-corrected kernel achieve high reward, whilst avoiding the instability observed in the case of SOSMC-ULA. Indeed, IMPDIFF struggles to reliably relocate mass towards the reward-relevant region, whereas SOUL relies on a single persistent trajectory, with inter mode transitions being rare in this case.

We conclude by noting that, from a practical perspective, kernel choice should be informed by problem geometry and computational budget. Local kernels (ULA, MALA) work well when successive targets are relatively close while momentum-based kernels (HMC) facilitate exploration. Furthermore, RWM, MALA, and HMC kernels include a Metropolis step, which reduces bias and can be helpful when the target has isolated modes, as seen for $V_{\text{tight}}$. We also note that, in low dimensions, the RWM variant is competitive despite not exploiting gradient information, however one would expect the respective performance to degrade in high dimensions.

### E.2. Reward training of Energy Based Models - 2D Datasets

Here, we consider synthetic two-dimensional datasets generated from standard benchmarking distributions, as visualised in Figure A.7. We refer to these datasets, after pre-processing, as the *two moons*, *circles*, and *blobs* datasets, denoted by $\mathcal{D}_{2M}$, $\mathcal{D}_{C}$, and $\mathcal{D}_{B}$ respectively.

### E.2.1 Pre-training details
First, we outline the details regarding the pre-training of the EBMs that are subsequently utilised within the reward tuning. Specifically, we utilise Persistent Contrastive Divergence (PCD) to approximate the gradient of the likelihood.

**Datasets.** For each distribution of interest, we generate $N = 20,000$ samples, forming the synthetic dataset $\tilde{\mathcal{D}} = \{\tilde{x}_i\}_{i=1}^{N}$, where $\tilde{x}_i \in \mathbb{R}^2$. Each dataset is then pre-processed via the transformation $x_i = s_{scale} \cdot (\tilde{x}_i - \mu_{\mathcal{D}})/\sigma_{\mathcal{D}}$, where $\mu_{\mathcal{D}}$ and $\sigma_{\mathcal{D}}$ are the empirical mean and standard deviations respectively, whilst $s_{scale} = 2.2$ is a global scaling factor that ensures the support, across datasets, is appropriately bounded in a desired region of $\mathbb{R}^2$. The resulting datasets, $\mathcal{D} \in \{\mathcal{D}_{2M}, \mathcal{D}_{C}, \mathcal{D}_{B}\}$, are displayed in Figure A.7.

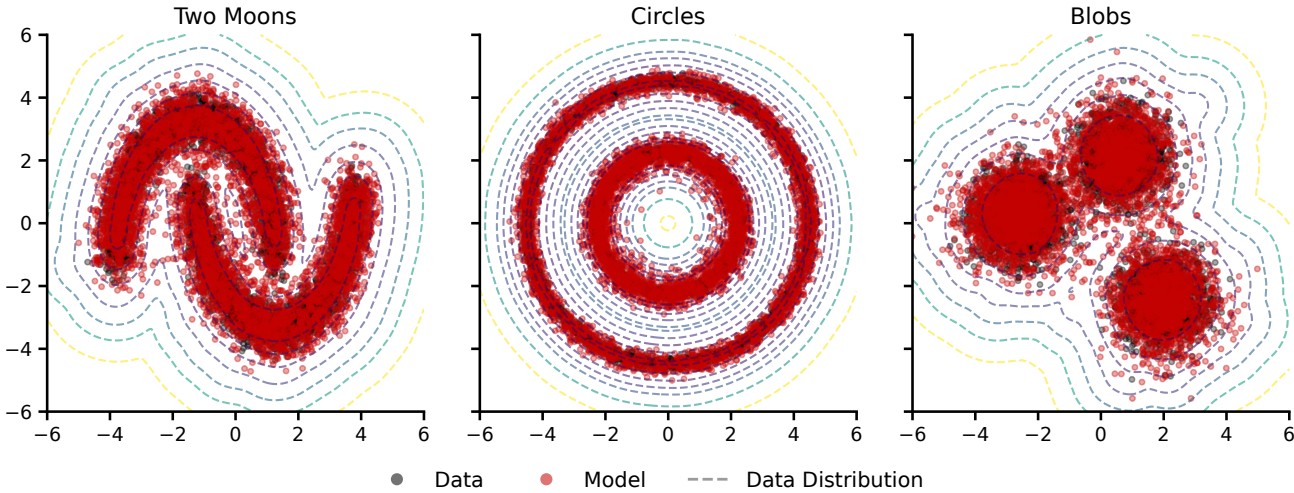

*Figure A.7.* **2D Datasets and EBMs:** The dataset samples (black), with corresponding contours, where darker lines indicating higher probability, and samples generated from the corresponding pretrained EBM (red), obtained after running 300 Langevin steps from random noise, for $\mathcal{D}_{2M}$, $\mathcal{D}_C$, and $\mathcal{D}_B$ (from left to right).

**Energy Model.** We parameterise the energy function, $E_\theta(x) : \mathbb{R}^2 \to \mathbb{R}$, using a Multi-Layer Perceptron (MLP). Specifically, our MLP consists of $h = 4$ hidden layers, with $D_h = 128$ hidden units each and SiLU activation functions (Ramachandran et al., 2017) after each hidden layer, defined by $\sigma(z) = z\ /\ (1 + \exp(-z))$, to help ensure $\nabla_x E_\theta(x)$ is differentiable everywhere, for the Langevin dynamics. Regarding initialisation, weights are drawn from $\mathcal{N}(0, \sigma_\mathcal{N}^2)$, with $\sigma_\mathcal{N} = 0.02$, and biases are initialised to zero.

**Training Objective.** We train the energy model by minimising the negative log-likelihood (NLL) of the data, where the gradient of the NLL is approximated via the contrastive divergence (CD) objective (Hinton, 2002).

More precisely, we have:

$$\mathcal{L}_{CD}(\theta) = \mathbb{E}_{x^+ \sim \mathcal{D}}[E_\theta(x^+)] - \mathbb{E}_{x^- \sim q_\theta}[E_\theta(x^-)],$$

where $x^+$ denotes *positive* samples from the data distribution $\mathcal{D}$, whilst $x^-$ denotes *negative* samples drawn from the model distribution $q_\theta(x) \propto \exp(-E_\theta(x))$.

To ensure the learned energy surface remains well-behaved and to prevent pathological gradients, we augment this standard CD loss with energy-magnitude and gradient-penalty regularisation terms, as inspired by Du & Mordatch (2020). The regularisation terms are defined as:

$$\mathcal{R}_E = \mathbb{E}_{x^+}[E_\theta(x^+)^2] + \mathbb{E}_{x^-}[E_\theta(x^-)^2],$$
$$\mathcal{R}_{GP} = \mathbb{E}_{x^+}[(\|\nabla_x E_\theta(x^+)\|_2 - 1)^2],$$

where the latter regularisation term serves to constrain the Lipschitz constant of the energy function, preventing pathological gradients, and thus stabilising the dynamics during the sampling phase (Du & Mordatch, 2020).

The total training loss $L(\theta)$ is thus given by:

$$\mathcal{L}(\theta) = \underbrace{\mathbb{E}_{x^+ \sim \mathcal{D}}[E_\theta(x^+)] - \mathbb{E}_{x^- \sim q_\theta}[E_\theta(x^-)]}_{\mathcal{L}_{CD}(\theta)} + \lambda_E \mathcal{R}_E + \lambda_{GP} \mathcal{R}_{GP}, \tag{33}$$

where we set $\lambda_E = 10^{-3}$ and $\lambda_{GP} = 0.2$.

**Sampling.** To approximate the negative phase of the gradient, $\nabla_\theta \mathbb{E}_{x^- \sim q_\theta}[E_\theta(x^-)]$, we require approximate samples from the current model distribution, $q_\theta$. To this end, we utilise ULA, where the update rule for particle state $X_k$ is

$$X_{k+1} = \text{Clamp}\left(X_k - \gamma \nabla_x E_\theta(X_k) + \sqrt{2\gamma}\,\xi_k, \quad \mathcal{C}_{min}, \mathcal{C}_{max}\right), \tag{34}$$

for $\gamma = 5 \times 10^{-3}$, $\xi_k \sim \mathcal{N}(0, I)$ where for $a, b \in \mathbb{R}$ the Clamp function takes vectors $\mathbb{R}^{d_x} \to \mathbb{R}^{d_x}$ and is defined component wise as

$$[\mathrm{Clamp}(X, a, b)]_i = \begin{cases} a \text{ if } [X]_i < a, \\ [X]_i \text{ if } a \leq [X]_i \leq b \\ b \text{ if } [X]_i > b \end{cases} ,$$

for $i = 1, \ldots, d_x$. With the choice of $\mathcal{C}_{min} = -6$ and $\mathcal{C}_{max} = 6$ the clamping restricts the particle evolution to the hypercube $\Omega = [\mathcal{C}_{min}, \mathcal{C}_{max}]^2 = [-6, 6]^2$, and this geometric constraint is utilised to prevent the divergence of the particle evolution to arbitrarily low energy pockets that may exist far from the data manifold, particularly due to the unbounded nature of the potential.

**Persistent Contrastive Divergence (PCD).** In order to reduce the computational cost associated with long mixing times often required to reach the stationary model distribution, $q_\theta$, we employ PCD augmented with a *replay buffer*. In particular, we maintain a persistent collection, $\mathcal{B} = \{X^{(m)}\}_{m=1}^{M}$, of $M = 20,000$ particles, that stores the final states of previous Markov chains. This buffer acts as a non-parametric approximation of $q_\theta$, so that, at each training step $t$, we construct a batch, of size $B$, of initial particles by sampling uniformly from said buffer, at indices $\{i_1, \ldots i_B\}$, resulting in significantly fewer steps under the transition kernel described above. Despite reducing the *burn-in* cost associated with long MCMC chains, persistent replay buffers risk *mode collapse*, where chains effectively become trapped in deep local minima, thus failing to effectively explore the support of the target distribution (Tieleman, 2008). To address this, and ensure ergodicity, we utilise the reinjection heuristic outlined in Du & Mordatch (2020), in which a fraction $\rho = 0.05$ of initialised particles are rejected and replaced with fresh uniform noise within the hypercube $\Omega$. The resulting initialised particles serve as the initial states, $X_0^{(i_b)}$, where $b \in [B]$, which are then evolved, in parallel, for K=80 steps, using the transition kernel outlined in (34). Indeed, $X_K^{(i_b)}$ are then saved to the buffer at the respective index, whilst simultaneously used to compute the negative phase of the gradient.

**Implementation Details.** For each $\mathcal{D} \in \{\mathcal{D}_{2\mathrm{M}}, \mathcal{D}_\mathrm{C}, \mathcal{D}_\mathrm{B}\}$ we leverage a number of further common components. First, for parameter optimisation, we utilise ADAM (Kingma & Ba, 2015), with a learning rate of $\eta = 2 \times 10^{-4}$ and $\beta_1 = 0.9$, and we further note that parameter gradients are clipped such that their resulting global norm is 10.0. Notably, gradients w.r.t. model parameters $\theta$ are detached during the sampling phase, that is the $X_K^{(i_b)}$ are considered as fixed negative samples for the loss computation, so that backpropagation does not occur through the sampling chain. Although a batch size of $B = 512$ is commonly utilised, the energy model is trained for 500 epochs in the cases of $\mathcal{D}_{2\mathrm{M}}$ and $\mathcal{D}_\mathrm{C}$, whilst for $\mathcal{D}_\mathrm{B}$ we only train for 200 epochs. The resulting energy model is subsequently frozen and denoted as $E_{\theta_0}$, in light of the fact that we subsequently utilise it for reward tuning, which takes place over a number of iterations.

### E.2.2 Reward tuning details

For a pre-trained energy model $E_{\theta_0}$, we perform reward tuning by optimising the reverse-KL regularised objective,

$$\ell(\theta) := -\mathbb{E}_{x \sim \pi_\theta}[R(x)] + \beta_{\mathrm{KL}} \, \mathrm{KL}(\pi_\theta \| \pi_0), \qquad \beta_{\mathrm{KL}} > 0, \tag{35}$$

where the reference distribution $\pi_0(x) \propto \exp(-E_{\theta_0}(x))$ is fixed throughout the reward tuning. We instantiate a trainable energy, $E_\theta$, with the same architecture outlined in Appendix E.2.1 , and with weights equal to that of the pre-trained weights $\theta_0$, whilst also maintaining a frozen reference copy, $E_0 \equiv E_{\theta_0}$, with all parameters detached. Indeed, the hyperparameter $\beta_{\mathrm{KL}} > 0$ explicitly governs the relative importance between reward maximisation and divergence from $\pi_0$, as illustrated in Figure A.8 and Figure A.13.

**Reward Functions.** Here, we consider indicator rewards, for half-planes, of the form

$$R_{\mathrm{left}} = \mathbf{1}_{\{x_1 < 0\}}, \quad R_{\mathrm{right}} = \mathbf{1}_{\{x_1 > 0\}}, \quad R_{\mathrm{lower}} = \mathbf{1}_{\{x_2 < 0\}}, \quad R_{\mathrm{upper}} = \mathbf{1}_{\{x_2 > 0\}}, \tag{36}$$

since not only do these rewards highlight that $R(x)$ may be non-differentiable, but these rewards also admit a closed form for the unique minimiser of (35). A standard variational argument shows that this objective is uniquely minimised by an exponential tilting of the reference distribution (Wainwright & Jordan, 2008),

$$\pi^\star(x) = \frac{1}{Z} \pi_0(x) \exp\left(\frac{1}{\beta_{\mathrm{KL}}} R(x)\right), \tag{37}$$

where $Z := \mathbb{E}_{\pi_0}[\exp(R(x)/\beta_{\mathrm{KL}})]$ is the partition function. In cases that $R(x) = \mathbf{1}_{\{x \in H\}}$, where $H \subset \mathbb{R}^2$ denotes a measurable half-space corresponding to one of the regions in (36), then (37) can be simply expressed as a piecewise reweighting of $\pi_0$,

$$\pi^\star(x) = \begin{cases} \dfrac{e^{1/\beta_{\mathrm{KL}}}}{Z}\,\pi_0(x), & x \in H, \\[2mm] \dfrac{1}{Z}\,\pi_0(x), & x \notin H, \end{cases} \tag{38}$$

where $Z = e^{1/\beta_{\mathrm{KL}}}\pi_0(H) + (1 - \pi_0(H))$. Equivalently, the induced mass on the reward half-plane, that is, the expected reward under $\pi^\star$, is

$$\pi^\star(H) = \frac{e^{1/\beta_{\mathrm{KL}}}\,\pi_0(H)}{e^{1/\beta_{\mathrm{KL}}}\pi_0(H) + (1 - \pi_0(H))}, \tag{39}$$

which monotonically increases as $\beta_{\mathrm{KL}}$ decreases. We note, provided $\pi_0(H) > 0$, that $\pi^\star$ approaches $\pi_0$ as $\beta_{\mathrm{KL}} \to \infty$, whereas $\pi^\star$ concentrates on $H$ as $\beta_{\mathrm{KL}} \to 0$.

**Tuning Procedure.** Reward tuning occurs through maintaining a collection of particles $\{X_k^{(i)}\}_{i=1}^N$, intended to approximate the evolving model distribution $\pi_{\theta_k}$. Specifically, we perform an outer-loop optimisation of $\theta$, utilising these persistent particles to construct Monte Carlo estimates of the surrogate loss, as outlined in Appendix F.4, whose gradient matches $\nabla_\theta \ell(\theta_k)$, to update the parameters $\theta_k$ by a single gradient descent step. At each outer iteration, the particles are, in general, propagated by applying a Markov kernel, $\mathsf{K}_{\theta_k}$, targeting $\pi_{\theta_k}$, for a fixed number of steps per outer iteration, denoted $K_{\mathrm{inner}}$. In this case we choose the ULA Gaussian transition,

$$X_{k+1}^{(i)} = X_k^{(i)} - \gamma_k \nabla_x E_{\theta_k}(X_k^{(i)}) + \sqrt{2\gamma_k}\,\sigma_{\mathrm{noise}}\,\xi_k^{(i)}, \qquad \xi_k^{(i)} \sim \mathcal{N}(0, I), \tag{40}$$

with $\gamma_k$ fixed in the case of IMPDIFF and $\sigma_{\mathrm{noise}} = 1.0$. Since we consider both IMPDIFF and SOSMC, we have $K_{\mathrm{inner}} = 1$ here, where we note the algorithm-specific details of IMPDIFF and SOSMC governing particle evolution are outlined in Appendix C and Appendix D.

During reward tuning, we take care to distinguish between two notions of the reward. The first notion of reward refers to the empirical mean computed on the particles utilised for gradient estimation, $\widehat{R}_{\mathrm{particle}}(k) = \frac{1}{N}\sum_{i=1}^N R(X_k^{(i)})$, which provides a low-variance statistic aligned with the particles that drive parameter updates. Crucially, this should not be interpreted as an unbiased estimate of $\mathbb{E}_{\pi_{\theta_k}}[R(X)]$, as particles are advanced only a single step between successive updates of $\theta_k$ and so their *unweighted* empirical distribution typically corresponds to a non-equilibrium distribution $q_k$ that may

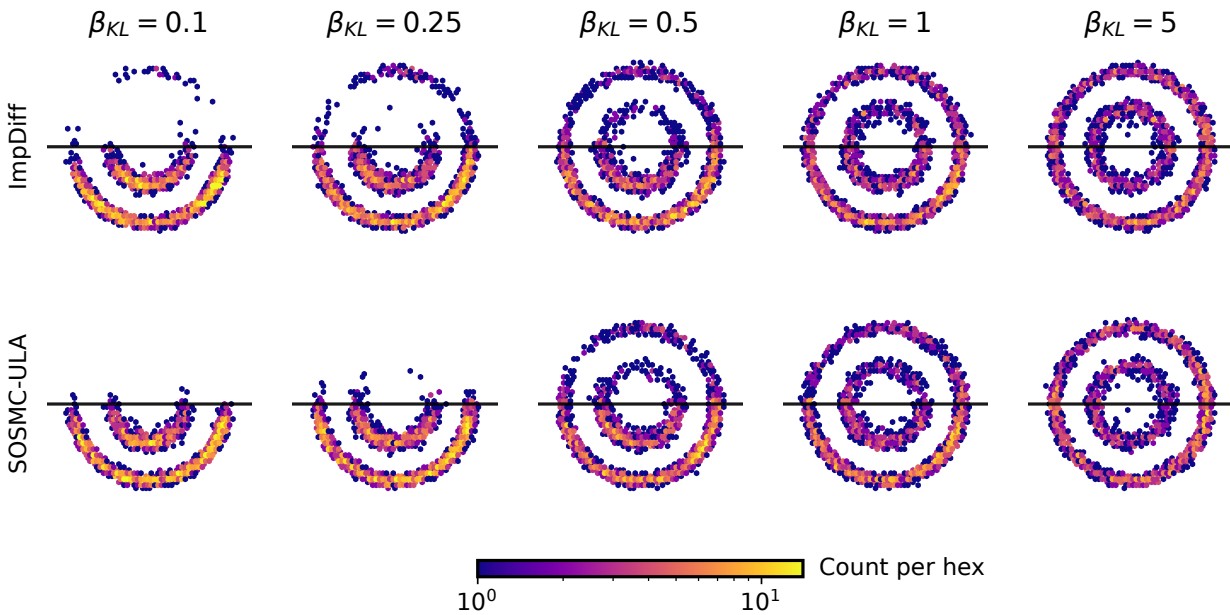

*Figure A.8.* Terminal density snapshots of $\pi_{\theta_K}$, using identical initialisation and shared noise, across increasing $\beta_{\mathrm{KL}}$, for $R_{\mathrm{lower}}$.

lag the instantaneous target $\pi_{\theta_k}$, particularly when $\theta_k$ changes quickly. Notably, for SOSMC, expectations under $\pi_{\theta_k}$ are instead approximated by a *weighted* empirical measure, as described in Appendix C, which for the ULA variant provides an asymptotically consistent estimator of expectations, of bounded test functions, under $\pi_{\theta_k}$ (Carbone et al., 2023; Cuin et al., 2025). As a result, estimates of the expected reward under the current model exhibit reduced bias relative to unweighted particle averages, for effectively the same propagation budget, as illustrated in Figure A.11. On the other hand, in the case of IMPDIFF, it is necessary to compute a *fresh reward* to accurately reflect the expected reward under the current model. Specifically, at pre-determined evaluation checkpoints we run a number of independent Langevin chains, targeting the *fixed* energy $E_{\theta_k}$, for a significantly greater number of steps than used in reward tuning, starting from a diffuse initialisation, such as uniform noise on our hypercube. After running and discarding an initial burn-in of $B_{\text{eval}}$ steps, the expected reward is estimated by time-averaging along each chain, that is, with $M_{\text{eval}}$ evaluation chains and $T_{\text{eval}}$ post-burn-in steps, we report

$$\widehat{R}_{\text{fresh}}(k) = \frac{1}{M_{\text{eval}} T_{\text{eval}}} \sum_{j=1}^{M_{\text{eval}}} \sum_{t=1}^{T_{\text{eval}}} R\left(\widetilde{X}_{k,t}^{(j)}\right), \tag{41}$$

where $\widetilde{X}_{k,t}^{(j)}$ are generated by the aforementioned chains. To be clear, $\widehat{R}_{\text{fresh}}(k)$ does not enter the optimisation procedure, but rather ensures that we report the performance of $E_{\theta_k}$, rather than transient properties of the persistent particles. Indeed, the effectiveness of the *weighted particle reward* is highlighted in Figure A.11.

Notably, it is possible to estimate $\text{KL}(p_\theta \| p_0)$ directly using numerical quadrature on the bounded hypercube $\Omega \in \mathbb{R}^2$ in a relatively efficient manner. In particular, we let $\{x^{(g)}\}_{g=1}^G$ denote a uniform grid of $G$ points, with associated cell area $\Delta^2$, so that

$$\log Z_\theta \approx \log \sum_{g=1}^G \Delta^2 \exp\left(-E_\theta(x_g)\right), \quad \log Z_0 \approx \log \sum_{g=1}^G \Delta^2 \exp\left(-E_0(x_g)\right),$$

$$\Rightarrow \text{KL} \approx \sum_{g=1}^G p_\theta(x_g) \left(\log p_\theta(x_g) - \log p_0(x_g)\right) \Delta^2,$$

where $\log p_\theta(x_g) \approx -E_\theta(x_g) - \log Z_\theta$, and similarly $\log p_0(x_g) \approx -E_0(x_g) - \log Z_0$.

Although explicitly detailed in Appendix F.4, we again highlight that throughout reward tuning, all particles and weights that form the (surrogate) losses are treated as stop-gradient quantities. To be clear, we take care not to backpropagate through the Markov kernel $\mathsf{K}_{\theta_k}$ and the dependence of both the reward terms in (61), and the energy difference terms in (63), on $\theta$. Indeed, both IMPDIFF and SOSMC ensure that optimisation does not depend on a path-wise gradient of the finite-step sampling distribution, that would notably depend on the sampler hyperparameters, in this manner.

**Implementation Details.** As alluded to, we specifically consider IMPDIFF and SOSMC-ULA, across all experimental setups. To begin, we outline implementation details common to both methods, and then subsequently detail method specific details. Any setup specific deviations are outlined in the relevant section.

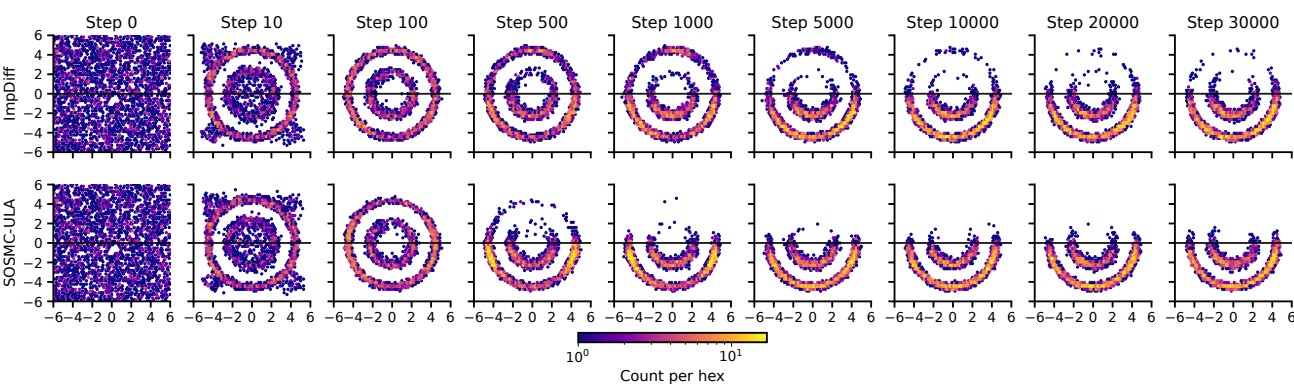

*Figure A.9.* Density snapshots along the sampling evolution of $\pi_{\theta_K}$ in the *illustrated example*, for both IMPDIFF (top) and SOSMC-ULA (bottom), with identical initialisation and shared noise.

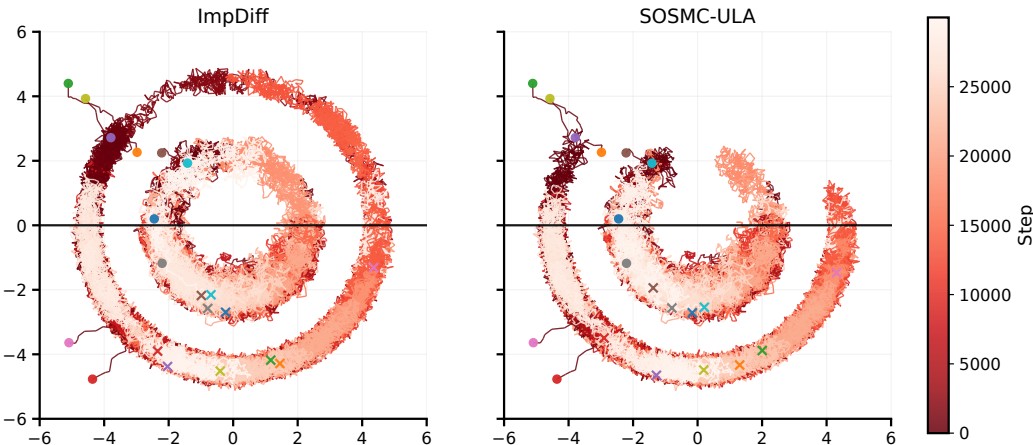

*Figure A.10.* Representative sampling trajectories for $\pi_{\theta_K}$ in the *illustrated example*, for both IMPDIFF (left) and SOSMC-ULA (right). Identical initialisation and shared noise are utilised for each trajectory across the methods.

For the sake of clarity, we first emphasise again that $K_{\text{inner}} = 1$ in both cases. Unless stated otherwise, reward tuning is conducted using $N = 10,000$ persistent particles over $K = 1,000$ outer iterations. Parameter updates are carried out using Adam, with a learning rate of $\eta = 2 \times 10^{-4}$ and $\beta_1 = 0.9$. Notably, the Langevin step size is initialised to $\gamma_k = 5 \times 10^{-3}$, coinciding with the sampling dynamics utilised in pre-training. In both cases, to evaluate the expected reward under the current model, $\widehat{R}_{\text{fresh}}(k)$ is periodically computed every $f_{\text{eval}}$ iterations, for which $M_{\text{eval}} = 1,000$ independent chains are utilised, each with $B_{\text{eval}} = 15,000$ burn-in steps followed by $T_{\text{eval}} = 5,000$ post burn-in steps. In fact, these values of $B_{\text{eval}}$ and $T_{\text{eval}}$ were chosen to ensure reliable mixing for IMPDIFF specifically, since this method exhibits significantly longer mixing times than SOSMC-ULA as evident in Figure A.9. Indeed, under a more restrictive computational budget, estimating the expected reward accurately for IMPDIFF thus becomes challenging.

Notably, in both IMPDIFF and SOSMC-ULA methods, particles are propagated using the unconstrained ULA Gaussian transition outlined in (40), with a fixed noise scale $\sigma_{\text{noise}} = 1.0$. For SOSMC-ULA, in order to preserve the validity of the closed-form kernel ratio underlying the weight correction, no clamping is permitted, and we further note such clamping is additionally absent from our implementation of IMPDIFF. Specific to SOSMC-ULA is the ESS threshold, set to $\tau = 0.9$, under which resampling occurs, w.r.t. normalised weights. In fact, to combat ESS instability, whilst maintaining exploration, we choose to adapt $\gamma_k$ based on the observed ESS, as described in Remark 4, using an adaptive threshold of $\tau_\gamma = 0.95$. In light of requiring particularly long MCMC chains to accurately estimate $\mathbb{E}_{p_{\theta_k}}[R(X)]$ in the case of IMPDIFF, we focus on evaluating relative performance in terms of iteration steps, rather than wall-clock runtimes, as was investigated in Appendix E.1.

**Results.** To begin, we first provide an illustrative example, highlighting key aspects of the reward tuning process for a single representative experimental trial. To this end, we specifically consider the *two moons* dataset $\mathcal{D}_{2M}$, equipped with the indicator reward $R_{\text{lower}}(x) = \mathbf{1}_{\{x_2 < 0\}}$, and regularisation hyperparameter $\beta_{\text{KL}} = 0.25$, to facilitate a setting that we expect to induce a meaningful shift of the model distribution during reward tuning.

Both $\widehat{R}_{\text{particle}}$ and $\widehat{R}_{\text{fresh}}$ are reported in Figure A.11, with the latter at a frequency of $f_{eval} = 100$. Under IMPDIFF, the (unweighted) particle reward increases slowly and is crucially a poor proxy for $\widehat{R}_{\text{fresh}}$, at least compared to under SOSMC-ULA, as expected from our discussion above. Despite $\widehat{R}_{\text{fresh}}$ stabilising at a greater value for SOSMC-ULA, which is notably closer to $p^*(H)$, we remark that this does not guarantee a lower value of $\ell$ is achieved. In light of this, we plot the optimisation trajectories in the $(\widehat{R}_{\text{fresh}}, \widehat{\text{KL}})$ plane, overlayed with objective contours, which provide geometric references against which the evolution of each method can be compared against. Notably, we observe, in Figure 3, SOSMC-ULA to result in a smoother trajectory that not only achieves a lower value of $\ell$, but does so in a lower number of iterations. Indeed the trade-off between the reward and deviation from the original distribution, as determined by $\beta_{\text{KL}}$, is also observed here.

As alluded to, the resulting energy landscape, of each tuned model, dictates the speed at which accurate estimates of $\mathbb{E}_{p_{\theta_k}}[R(X)]$, and in turn $\ell(\theta_k)$, can be obtained, since this landscape explicitly affects the dynamics of particles we choose to evolve, as demonstrated in Figure A.10. Indeed, the required mixing time for IMPDIFF is empirically observed to be significantly greater than that of SOSMC-ULA, as further highlighted in both Figure A.9 and A.12. We conclude this

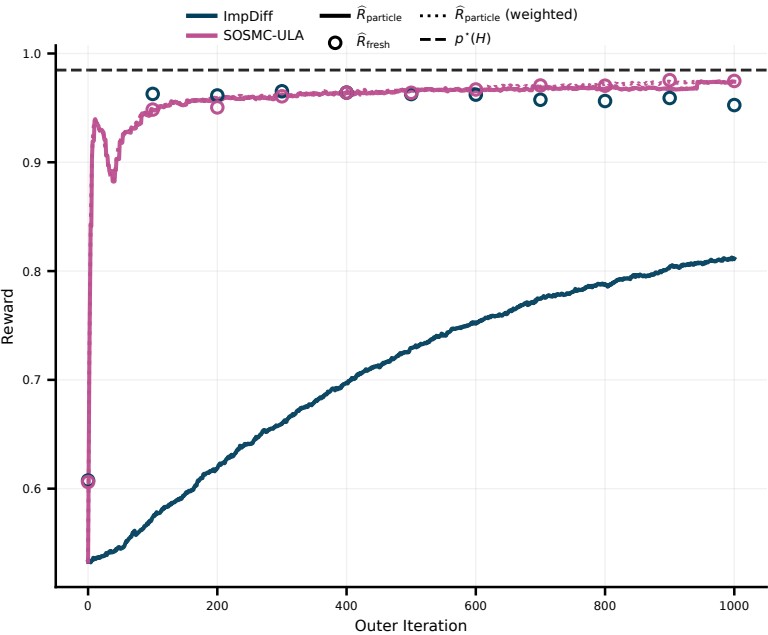

*Figure A.11.* Trajectories of $\widehat{R}_{\text{particle}}$ and $\widehat{R}_{\text{fresh}}$ for the *illustrated example*. The (weighted) $\widehat{R}_{\text{particle}}$ serves as a far better proxy for $\widehat{R}_{\text{fresh}}$ in the case of SOSMC-ULA, whilst $\widehat{R}_{\text{fresh}}$ also converges closer to $\pi^*(H)$ for this method.

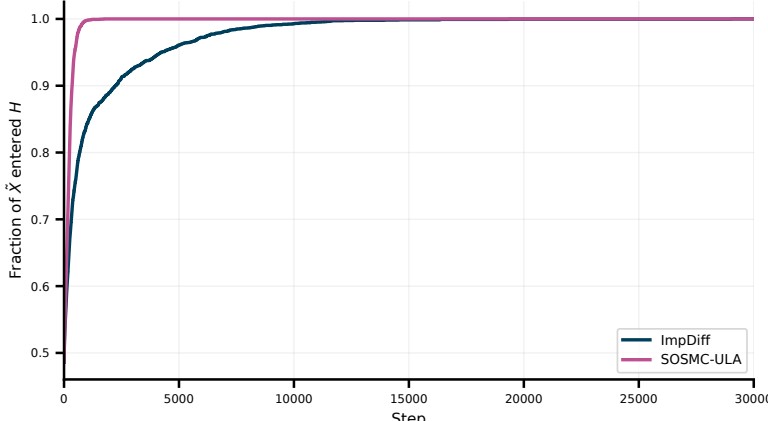

*Figure A.12.* Proportion of sampling trajectory, for $\pi_{\theta_K}$, in the reward region as in the *illustrated example*, for both IMPDIFF and SOSMC-ULA. Identical initialisation and shared noise are utilised for each trajectory across the methods.

illustrative example by remarking that more stable tuning dynamics are observed in the case of SOSMC-ULA, as indicated by logged tuning gradient norms, whilst the samples generated from noise, for the final tuned models, are visualised in Figure A.8, under the $\beta_{\text{KL}} = 0.25$ panel.

The relative performance of IMPDIFF and SOSMC-ULA, across a range of regularisation strengths $\beta_{\text{KL}}$, is illustrated in Figure A.13, where again the evolution is presented in the $(\widehat{R}_{\text{fresh}}, \widehat{\text{KL}})$ plane, as was the case in Figure 3. The best-performing IMPDIFF iterate is circled and indicated in red, as is the corresponding objective contour. In particular, for each $\beta_{\text{KL}}$ considered, SOSMC-ULA reaches a better, or at least comparable, contour, typically in a lower number of iterations. Indeed, by correcting for non-equilibrium bias through importance weighting, SOSMC-ULA results in improved (surrogate) gradient estimates, which not only result in models corresponding to improved objective values, in fewer iterations, but also demonstrates reduced reliance on potentially costly fresh evaluations to accurately understand performance, as demonstrated in Figure A.14.

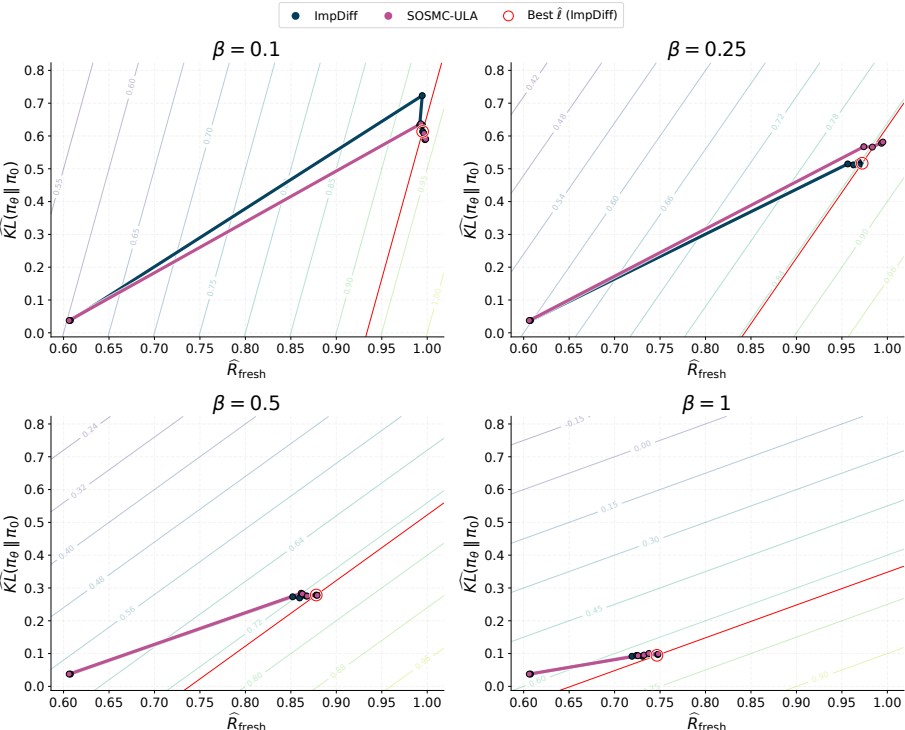

*Figure A.13.* Tuning trajectories for IMPDIFF and SOSMC-ULA in the $(\widehat{R}_{\text{fresh}}, \widehat{\text{KL}})$ plane, across values of $\beta_{\text{KL}}$. Contours denote level sets of $\ell$, with the best IMPDIFF solution highlighted in red. Note this corresponds to a $\widehat{R}_{\text{fresh}}$ budget of $20,000$ Langevin steps, with $15,000$ of those being burn-in steps.

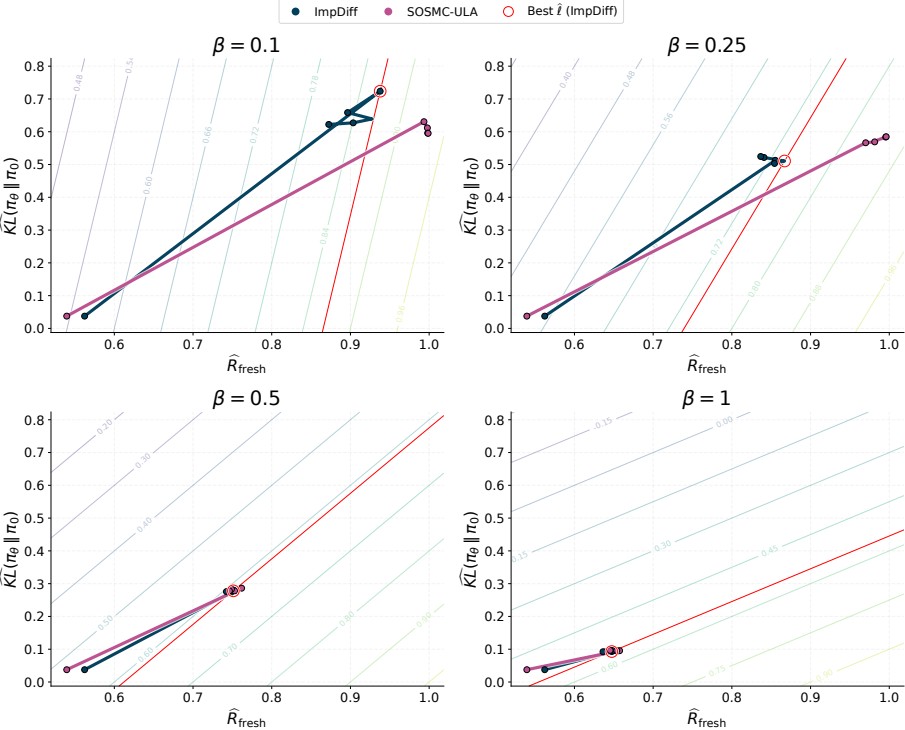

*Figure A.14.* Tuning trajectories for IMPDIFF and SOSMC-ULA in the $(\widehat{R}_{\text{fresh}}, \widehat{\text{KL}})$ plane, across values of $\beta_{\text{KL}}$. Contours denote level sets of $\ell$, with the best IMPDIFF solution highlighted in red. Note this corresponds to a $\widehat{R}_{\text{fresh}}$ budget of $5,000$ Langevin steps, with $500$ of those being burn-in steps.

## E.3. Reward tuning of Energy Based Models - MNIST

Here, we consider the reward tuning of a convolutional EBM pre-trained on MNIST, in the setting where the sampler utilised during pre-training differs from that of the Markov kernel utilised during reward tuning. Indeed, this experiment serves as not only a higher-dimensional setting, but one that is practically relevant, since pre-trained models, such as an EBM, are often only available together with a specific sampling procedure used to visualise samples, whilst reward tuning proceeds under a simpler, analytically tractable ULA transition kernel.

**Setup.** Here, we utilise a publicly available checkpoint of an EBM pre-trained on MNIST, where the underlying network is a convolutional architecture equipped with Swish activations. To be explicit, this network consists of four strided convolution blocks subsequently followed by two fully connected layers, with a hidden dimension of $D_h = 32$. To emphasise the dimensionality of the problem, recall inputs to this network are single-channel images, $x \in \mathbb{R}^{28 \times 28 \times 1}$.

**Pre-training details.** As was the case for in the setting of 2D datasets, pre-training of the EBM proceeds via PCD, albeit with a sampler that differs from our tuning kernel. Since many details are the same, we refer to Appendix E.2.1 for the features characteristic of PCD, and instead focus on the aforementioned sampler. Define the clip function as $\text{clip}_a(X) := \text{clamp}(X, -a, a)$ for a positive real number $a > 0$. Specifically, the following update is performed,

$$x_t = \text{clip}_1 \left\{ \text{clip}_1 \left( x_{t-1} + \sigma \varepsilon_{t-1} \right) - \alpha \, \text{clip}_c \left( \nabla_x E_\theta \, \text{clip}_1 \left( x_{t-1} + \sigma \varepsilon_{t-1} \right) \right) \right\},$$

where $\varepsilon_{t-1} \sim \mathcal{N}(0, I)$, $c = 0.03$ is a gradient clipping threshold, and $a = 1$ clamps resulting transitions to $[-1, 1]$, that is the expected pixel space for how the MNIST data is normalised. Here, $\sigma = 0.005$ denotes the scale of the additive *jitter* noise, and $\alpha = 10$ is a step-size parameter controlling the magnitude of the gradient descent step. Note, $\alpha$ should not be interpreted as a step size in the normal sense here, since element-wise gradient clipping occurs, the per-pixel update magnitude, as a result of the gradient term, is bounded by $\alpha c = 0.3$. To be clear, the noise injection occurs before the gradient step, and so the resulting Markov kernel does not admit a simple Gaussian transition density, nor does it satisfy detailed balance with respect to $\pi_\theta(x) \propto \exp(-E_\theta(x))$.

Therefore, although the sampler outlined above is utilised for warm-start initialisation, as well as evaluation of any tuned models, reward tuning itself is carried out using a *pure* (Gaussian) ULA kernel. Indeed, this ensures that the kernel-density ratios required by SOSMC-ULA are well-defined during optimisation, whilst our evaluation remains well aligned with the established sampling behaviour of the pre-trained EBM.

**Reward tuning details.**

Starting from the pre-trained energy model $E_{\theta_0}$, we perform reward tuning by optimising the reverse-KL regularised objective, as in (35), where again the reference distribution $\pi_0(x) \propto \exp(-E_{\theta_0}(x))$ is fixed throughout reward tuning.

Here, we consider three bounded rewards, evaluated after clamping pixel values to $[-1, 1]$, namely

$$R_{\text{bright}}(x) = \frac{1}{28 \cdot 28} \sum_{u,v} x_{u,v},$$

$$R_{\text{dark}}(x) = -\frac{1}{28 \cdot 28} \sum_{u,v} x_{u,v},$$

$$R_{\text{half}}(x) = \frac{1}{2} \left( \frac{1}{14 \cdot 28} \sum_{u=15}^{28} \sum_{v=1}^{28} x_{u,v} - \frac{1}{14 \cdot 28} \sum_{u=1}^{14} \sum_{v=1}^{28} x_{u,v} \right),$$

where we note $R_{\text{bright}}$ and $R_{\text{dark}}$ promote globally light and dark images respectively, whilst $R_{\text{half}}$ promotes mass to concentrate in the lower half of the image, since $u$ denotes pixel indexes in the vertical axis ranging from top to bottom.

Reward tuning proceeds in an identical manner to as detailed in Appendix E.2.2 , where we take care to emphasise that no clamping, gradient clipping, or additional jitter noise is applied within the proposal kernel for tuning. Indeed, the loss utilised for tuning is in fact the sum of the reward surrogate and reverse-KL surrogate (see Appendix F.4). Once again, we distinguish between $\widehat{R}_{\text{particle}}$ and $\widehat{R}_{\text{fresh}}$, however note the latter provides the reward estimate aligned with the established sampling procedure of the pre-trained model. Consequently, we opt to utilise $\widehat{R}_{\text{fresh}}$ for performance evaluation.

Unless stated otherwise, the reward tuning is run for $K = 1,000$ outer iterations, with $K_{\text{inner}} = 1$, for $N = 1,000$ particles. Parameter updates are carried out using ADAM, with a learning rate of $\eta = 1 \times 10^{-4}$ and $\beta_1 = 0.9$, which notably matches

the learning rate utilised in the pre-training procedure. Here, the step size is initialised to $\gamma_k = 5 \times 10^{-3}$, whilst for SOSMC-ULA an ESS threshold of $\tau = 0.9$ is leveraged, whilst $\tau_\gamma = 0.95$.

**Results.** Across all rewards and $\beta_{\text{KL}}$ values considered, both IMPDIFF and SOSMC-ULA consistently increase $\widehat{R}_{\text{fresh}}$ relative to the pre-trained baseline. Although we are unable to numerically estimate $\text{KL}(\pi_\theta \| \pi_0)$ in this high dimensional setting, we do crucially observe the digit-like structure to be preserved under the respective reward tuning, although some mode collapse is notably observed at low values of $\beta_{\text{KL}}$, particularly for $R_{\text{bright}}$ and $R_{\text{dark}}$, corroborating findings in Marion et al. (2025). Although performance is generally comparable for the $R_{\text{bright}}$ and $R_{\text{dark}}$ while SOSMC-ULA exhibits slightly stronger improvements for $R_{\text{half}}$, as evident in Figure 5 through more efficient sampling.

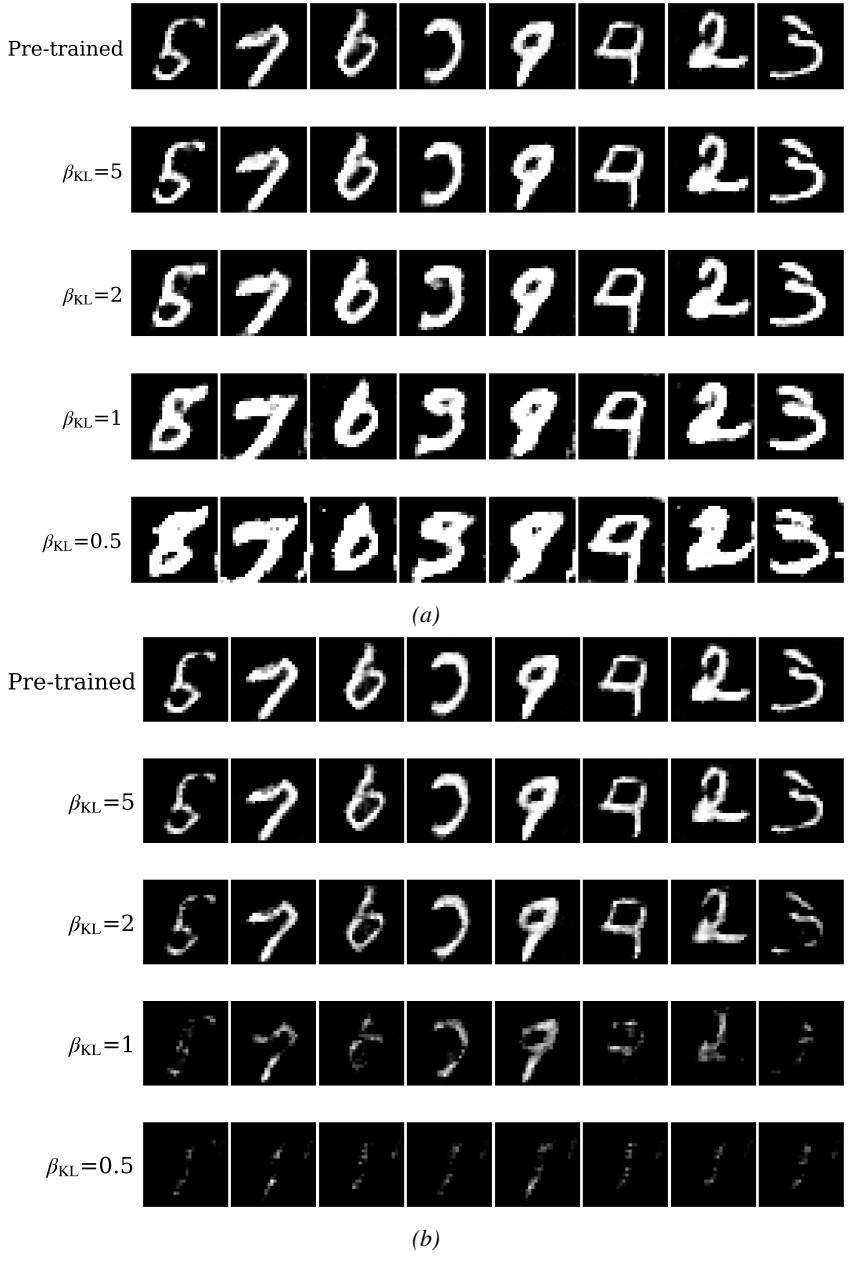

*Figure A.15.* Samples obtained from the frozen $\pi_{\theta_0}$ (top), and $\pi_{\theta_K}$ across various values of $\beta_{\text{KL}}$. In *(a)* $R_{\text{bright}}$ favours **brighter** images, whereas in *(b)* $R_{\text{dark}}$ favours **darker** images. All sampling trajectories are identically initialised and share noise.

### E.4. MMLE - Image Deblurring

Here, we consider a Bayesian image deblurring problem, in which the goal is to recover a latent clean image $x$ from a blurred and noisy observation $y$. In fact, this is an ill-conditioned inverse problem, and consequently the model is equipped with a total-variation prior, whose strength is governed by the unknown scalar parameter $\theta$, yielding a high-dimensional MMLE problem with a non-differentiable posterior over the latent image. This setup closely follows that of Section 5.3 in Encinar et al. (2025), and is similar to the image reconstruction problems studied in Durmus et al. (2018) and Goldman et al. (2022).

**Setup.** Let $x^\star \in \mathbb{R}^{d_x}$ denote the ground-truth clean image, with image dimension $d_x = n_1 \times n_2$, and the observed image by

$$y = Bx^\star + \varepsilon, \qquad \varepsilon \sim \mathcal{N}\left(0, \sigma^2 I_{d_x}\right),$$

where $B \in \mathbb{R}^{d_x \times d_x}$ is a known blurring operator. As in Encinar et al. (2025), $B$ is a sparse local averaging operator, that, for a specified patch size $P$, replaces interior pixels $x^\star_{i,j}$ by the respective uniform average over a $P \times P$ neighbourhood of pixels, whilst near image boundaries truncation and renormalisation is applied accordingly. Indeed, $B$ is information destroying and hence direct inversion is unstable, and the likelihood alone is insufficient to recover a visually meaningful reconstruction. Consequently, we regularise the latent image using an isotropic total-variation prior, which note not only favours piecewise-smooth images, but also helps to preserve sharp edges,

$$p_\theta(x) = C(\theta)\exp\left(-e^\theta \mathrm{TV}(x)\right),$$

where $C(\theta) \propto e^{d_x\theta}$ is the normalising constant and $e^\theta > 0$ is the TV precision, which, as noted in Encinar et al. (2025), often requires manual tuning, as is done in Durmus et al. (2018) and Goldman et al. (2022). Here, $\mathrm{TV}(x) = \|\nabla_d x\|_1$ denotes the total variation, with $\nabla_d$ denoting the two-dimensional discrete gradient operator and is crucially non-differentiable.

Combining the aforementioned TV prior with the Gaussian likelihood results in a posterior of the form

$$\pi_\theta(x) = p_\theta(x|y) \propto \exp\left(-\frac{1}{2\sigma^2}\|y - Bx\|_2^2 - e^\theta \mathrm{TV}(x) + \log C(\theta)\right).$$

Therefore, the posterior may be expressed as $\pi_\theta(x) \propto \exp\left(-U_\theta(x)\right)$, up to constants independent of $\theta$ and $x$, where

$$U_\theta(x) = \frac{1}{2\sigma^2}\|y - Bx\|_2^2 + e^\theta \mathrm{TV}(x) - d_x\theta,$$

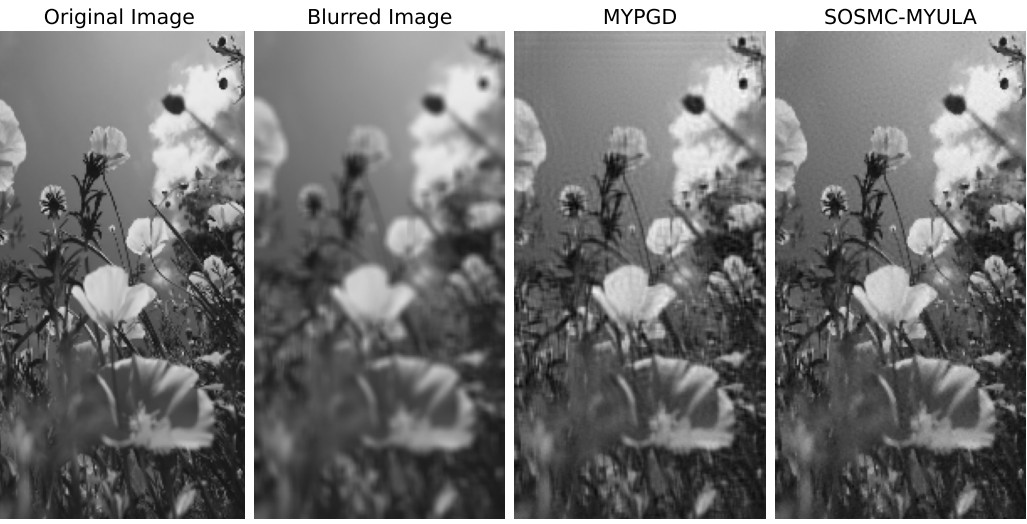

*Figure A.16.* The ground truth image (left), blurred image (middle-left) and posterior mean reconstructions for MYPGD (middle-right) and SOSMC-MYULA (right) particle clouds, for image deblurring Setup B.

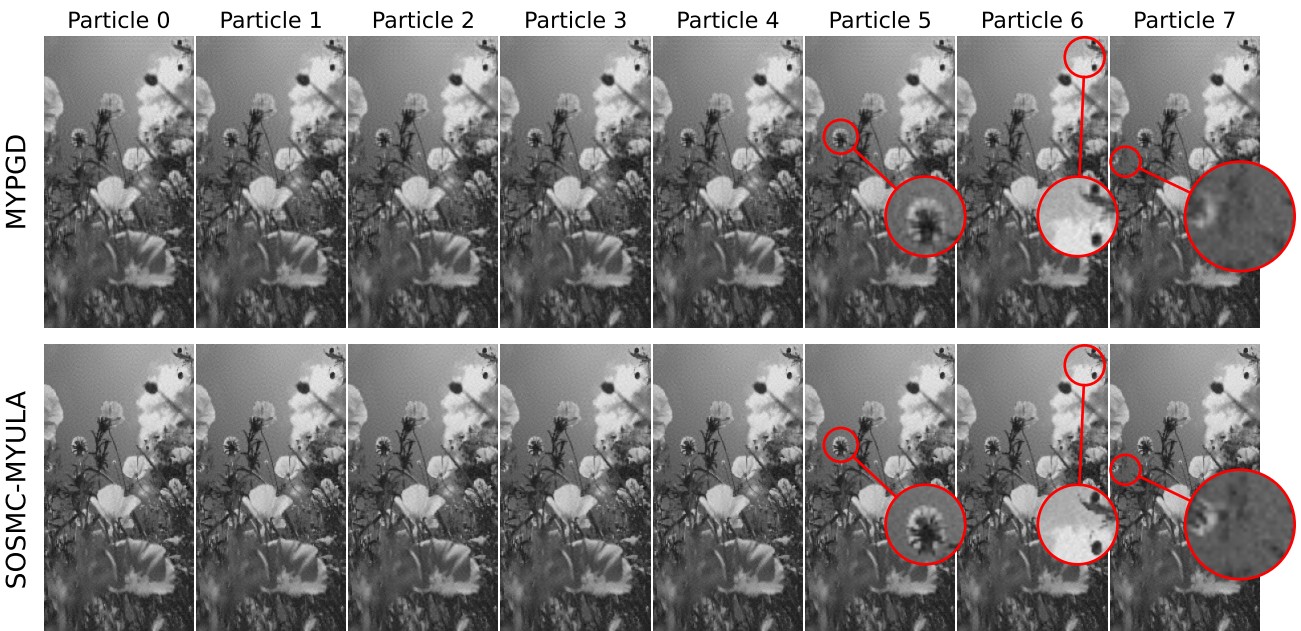

*Figure A.17.* Subset of the final particle clouds for MYPGD (top) and SOSMC-MYULA (bottom), in the image deblurring Setup B, where note $N = 20$. Selected areas are enlarged for detailed visual comparison between the two methods.

and so the gradient of the negative log-marginal likelihood can be expressed as

$$\nabla_\theta \ell(\theta) = -\mathbb{E}_{x \sim p_\theta(x|y)} \left[ \nabla_\theta \log p_\theta(x, y) \right]$$
$$= -\mathbb{E}_{x \sim p_\theta(x|y)} \left[ \nabla_\theta \left\{ \log p(y|x) + \log p_\theta(x) \right\} \right]$$
$$= -\mathbb{E}_{x \sim p_\theta(x|y)} \left[ \nabla_\theta \left\{ -\frac{1}{2\sigma^2} \|y - Bx\|_2^2 - e^\theta \mathrm{TV}(x) + d_x \theta \right\} \right]$$
$$= \mathbb{E}_{x \sim p_\theta(x|y)} \left[ e^\theta \mathrm{TV}(x) - d_x \right],$$

where in the second equality we have used the fact that $p_\theta(x, y) = p(y|x) p_\theta(x)$, whilst in the third equality we have used the fact that $p(y|x) = \left( 2\pi\sigma^2 \right)^{-d_x/2} \exp \left( -\frac{1}{2\sigma^2} \|y - Bx\|_2^2 \right)$. Therefore, the minimisation objective can indeed be expressed as an expectation over $\pi_\theta$, as in (2), with $H_\theta(\cdot) = e^\theta \mathrm{TV}(\cdot) - d_x$.

**Proximal Dynamics.** For fixed $\theta$, sampling from $\pi_\theta$ is challenging, since, as previously noted, TV is non-differentiable. Nevertheless, as outlined in Encinar et al. (2025), particle algorithms applicable to standard MMLE problems can be extended to cases in which $p_\theta(x, y)$ may be non-differentiable. In particular, such extensions are based on various discretisation schemes of

$$d\boldsymbol{\theta}_t^N = -\frac{1}{N} \sum_{i=1}^N \nabla_\theta U^\lambda(\boldsymbol{\theta}_t^N, \mathbf{X}_t^{i,N}) dt + \sqrt{\frac{2}{N}} d\mathbf{B}_t^{0,N}$$
$$d\mathbf{X}_t^{i,N} = -\nabla_x U^\lambda(\boldsymbol{\theta}_t^N, \mathbf{X}_t^{i,N}) dt + \sqrt{2} d\mathbf{B}_t^{i,N},$$

where $U^\lambda$ is the $\lambda$-Moreau-Yosida (MY) approximation of $U$. In fact, through removing the noise term from the $\theta$ dynamics, we obtain the following stochastic differential equations,

$$d\boldsymbol{\theta}_t^N = -\frac{1}{N} \sum_{i=1}^N \nabla_\theta U^\lambda(\boldsymbol{\theta}_t^N, \mathbf{X}_t^{i,N}) dt$$
$$d\mathbf{X}_t^{i,N} = -\nabla_x U^\lambda(\boldsymbol{\theta}_t^N, \mathbf{X}_t^{i,N}) dt + \sqrt{2} d\mathbf{B}_t^{i,N},$$

which we subsequently discretise to obtain the MYPGD baseline, as introduced in Encinar et al. (2025).

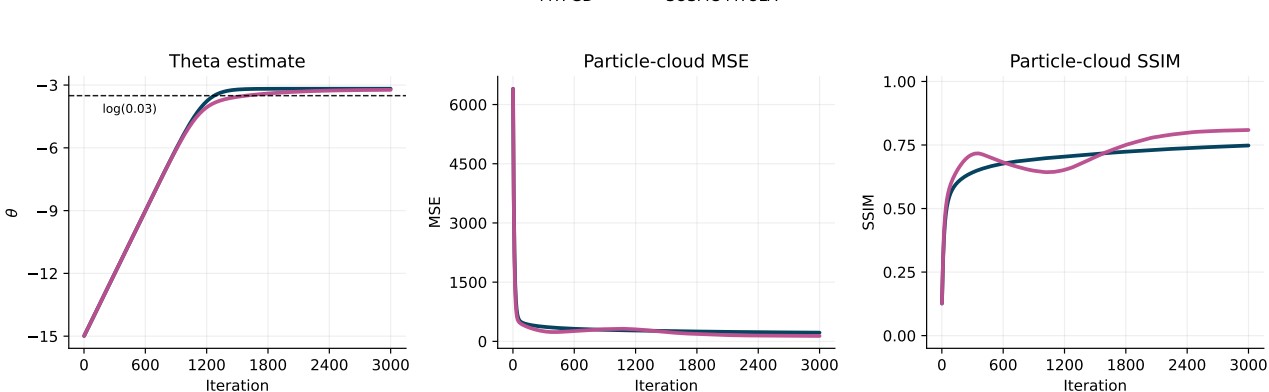

*Figure A.18.* Evolution of parameter estimates (left), MSE (middle) and structural similarity index measure (right) for MYPGD and SOSMC-MYULA particle clouds, for the image deblurring Setup B, with $N = 20$ particles and $\theta_0 = -15$. The horizontal dashed line indicates the strength of the total variation prior manually set in Durmus et al. (2018) and Goldman et al. (2022).

Here, the smooth component of $\nabla_x U_\theta(x)$ is given by $\nabla_x \Phi(x) = \frac{B^\top(Bx-y)}{\sigma^2}$, and so, for $\lambda > 0$ and image step-size $\gamma > 0$, the MYULA transition is defined as

$$X_{k+1}^{(i)} = \left(1 - \frac{\gamma}{\lambda}\right) X_k^{(i)} - \gamma \nabla_x \Phi(X_k^{(i)}) + \frac{\gamma}{\lambda} \text{prox}_{\lambda e^\theta \text{TV}}(X_k^{(i)}) + \sqrt{2\gamma}\xi_k^{(i)}, \qquad \xi_k^{(i)} \sim \mathcal{N}(0, I_{d_x}),$$

where

$$\text{prox}_{\lambda e^\theta \text{TV}}(z) = \arg\min_u \left\{\lambda e^\theta \text{TV}(u) + \frac{1}{2}\|u - z\|_2^2\right\}.$$

Indeed, both MYPGD and SOSMC-MYULA evolve the particle cloud $\{X_k^{(i)}\}_{i=1}^N$ by a single MYULA step per outer iteration, where, in the former method an equally weighted persistent particle cloud is evolved, whilst the parameter is updated using an unweighted empirical particle average estimate. To be clear, following Encinar et al. (2025), we avoid computing the joint proximal operator over $\theta$ and $x$ in the $\theta$-update, and instead utilise a hybrid scheme, so that we use proximal updates for the particles and standard gradient based updates for the parameter,

$$\theta_{k+1} = \theta_k - \gamma \left[\frac{1}{N} \sum_{i=1}^N \left(e^{\theta_k} \text{TV}(X_k^{(i)}) - d_x\right)\right],$$

or, equivalently using the dimension-normalised gradient estimator $H_\theta(\cdot)$, this may be written as

$$\theta_{k+1} = \theta_k - \gamma \left[\frac{1}{N} \sum_{i=1}^N \left(\frac{e^{\theta_k} \text{TV}(X_k^{(i)})}{d_x} - 1\right)\right].$$

In contrast, and as seen previously, in the case of SOSMC-MYULA a weighted estimate is utilised,

$$g_k = \sum_{i=1}^N w_k^{(i)} \left(\frac{e^{\theta_k} \text{TV}(X_k^{(i)})}{d_x} - 1\right).$$

**Implementation Details.** We evaluate MYPGD and SOSMC-MYULA on three distinct deblurring setups, designed to separate the effect of image content from the effect of blur severity. Setups A and B use different black and white ground-truth images $x^\star$ under the same blurring operator $B$, while Setup C increases the blur severity for the same image used in Setup A. Specifically, Setup A considers the acoustic-guitar image outlined in the left panel of Figure 6 as $x^\star$, where note $d_x = n_1 \times n_2 = 292 \times 119$, and applies $B$ with patch size $P = 5$, while we set $\sigma = 0.47$. Setup B considers the flower image outlined in the left panel of Figure A.16 as $x^\star$, where now $d_x = n_1 \times n_2 = 292 \times 150$, and applies the same $B$ as

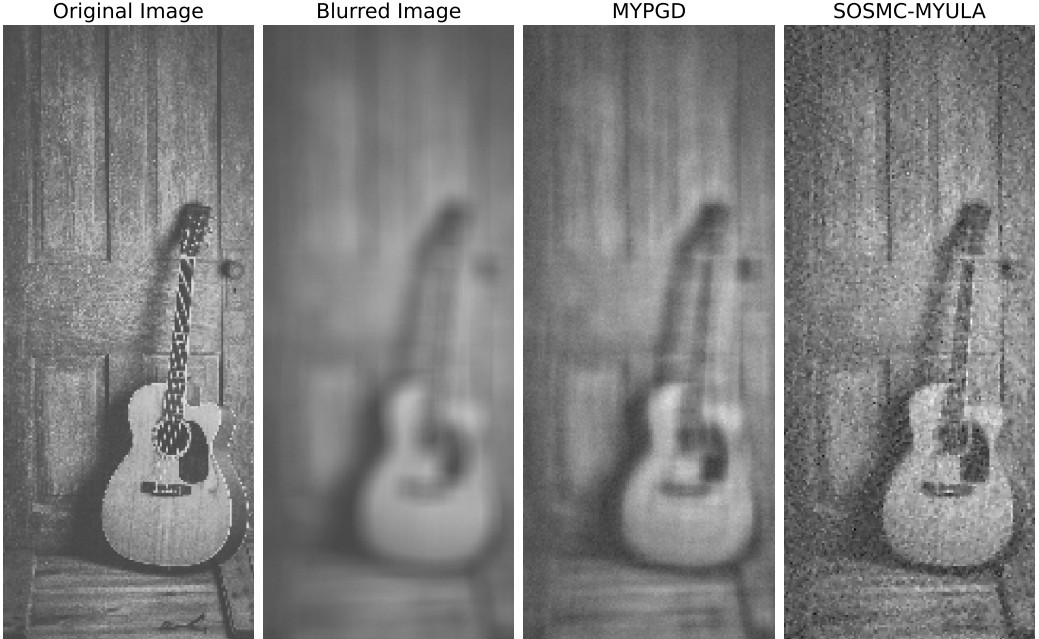

*Figure A.19.* The ground truth image (left), blurred image (middle-left) and posterior mean reconstructions for MYPGD (middle-right) and SOSMC-MYULA (right) particle clouds, for image deblurring Setup C.

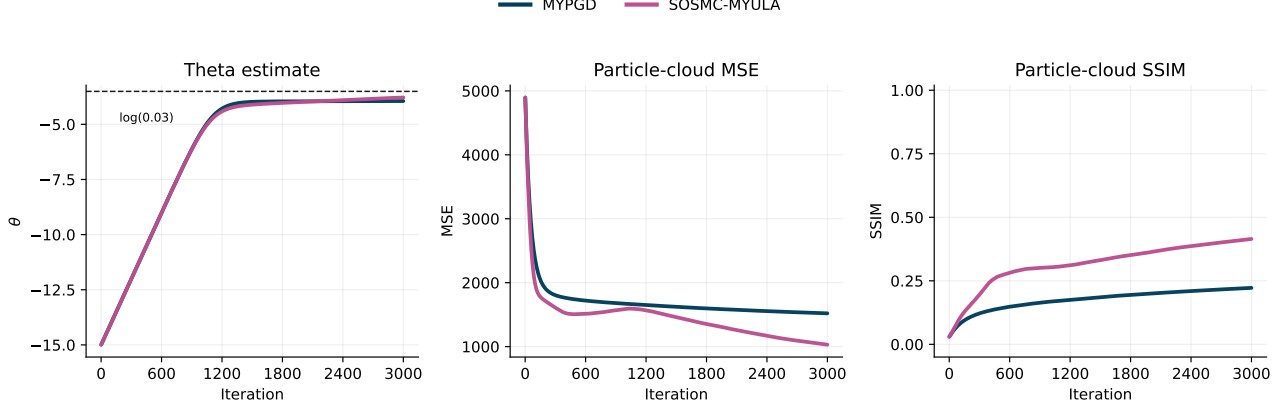

*Figure A.20.* Evolution of parameter estimates (left), MSE (middle) and structural similarity index measure (right) for MYPGD and SOSMC-MYULA particle clouds, for the image deblurring Setup C, with $N = 20$ particles and $\theta_0 = -15$. The horizontal dashed line indicates the strength of the total variation prior manually set in Durmus et al. (2018) and Goldman et al. (2022).

in Setup A. In Setup C, we again utilise the acoustic-guitar image from Setup A, however, to create a more challenging inverse problem, increase the patch size to $P = 10$ and the noise level to $\sigma = 1.0$. We note this effect can be understood by comparing the middle-left panels of Figures 6 and A.19. Furthermore, in Setups A and B particles are initialised according to $\{X_0^{(i)}\}_{i=1}^N \sim \mathcal{N}(50, 10^2)$, whereas in Setup C we initialise according to $\{X_0^{(i)}\}_{i=1}^N \sim \mathcal{N}(50, 30^2)$. Across all setups, we set $N = 20$, $K = 3000$, $\theta_0 = -15$, and $\gamma_k$ is initialised to $0.01$ for both methods, as is a common learning rate of $\eta = 0.01$ utilised. Following Encinar et al. (2025), we set $\lambda = 0.4$ and leverage a Douglas-Rachford TV method (Douglas & Rachford, 1956) throughout, via the `proxTV` Python package, for the computation of $\text{prox}_{\lambda e^\theta \text{TV}}(z)$.

**Results.** In accordance with Encinar et al. (2025), we report the evolution of the $\theta$ estimate, mean squared error (MSE) and structural similarity index (SSIM) between the respective particle clouds and ground-truth images across setups. The latter metric can be understood to capture similarity in local intensity, contrast and structural agreement between the reconstruction and $x^\star$, and is notably sensitive to the preservation of edges, making it particularly natural for deblurring

problems. In Figures 6, A.16 and A.19 we illustrate $x^\star$ and $y$, along with the posterior mean reconstructions for MYPGD and SOSMC-MYULA, while the corresponding evolutions of the aforementioned metrics can be seen in Figures 7, A.18 and A.20, for Setups A, B and C respectively. Across all setups, we observe superior performance for SOSMC-MYULA in terms of image reconstruction quality, through both the MSE and SSIM. Furthermore, in Figures A.21, A.17 and A.22 we visualise a subset of the final particle clouds for Setups A, B and C respectively, and notably observe visibly sharper particle clouds for SOSMC-MYULA, as highlighted in various locations of the corresponding image. We also note that the final value of $\theta$ estimates, for both MYPGD and SOSMC-MYULA, across all setups, is remarkably close to the value set manually in Durmus et al. (2018) and Goldman et al. (2022), as indicated in the left panel Figures 7, A.18 and A.20.

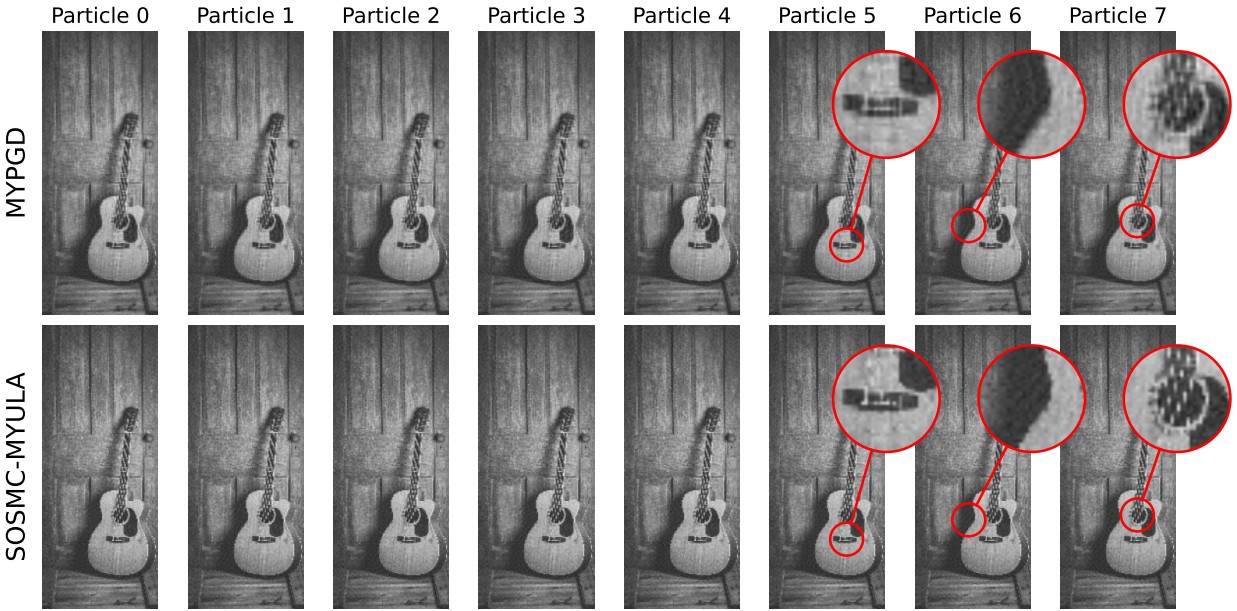

*Figure A.21.* Subset of the final particle clouds for MYPGD (top) and SOSMC-MYULA (bottom), in the image deblurring Setup A, where note $N = 20$. Selected areas are enlarged for detailed visual comparison between the two methods.

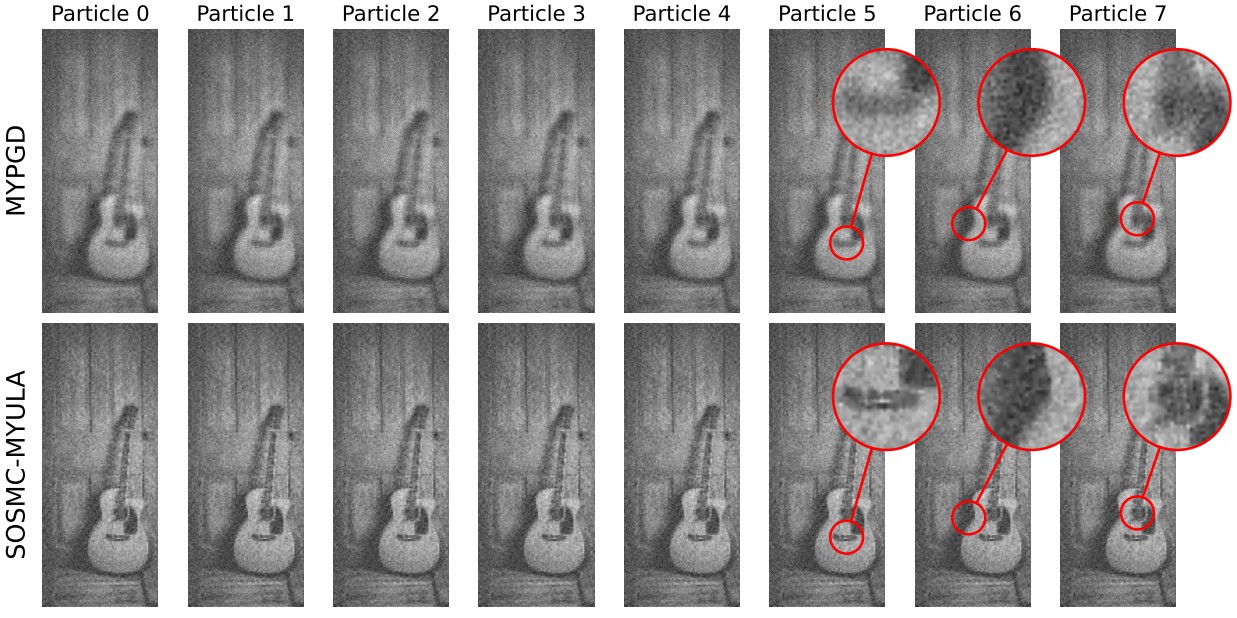

*Figure A.22.* Subset of the final particle clouds for MYPGD (top) and SOSMC-MYULA (bottom), in the image deblurring Setup C, where note $N = 20$. Selected areas are enlarged for detailed visual comparison between the two methods.

## F. Experimental Details - Theoretical Results

### F.1. Derivation of $\nabla_\theta \ell$ for reward tuning of Langevin Processes

We begin by letting $V(\cdot, \theta) : \mathbb{R}^{d_x} \to \mathbb{R}$ denote a smooth potential with parameters $\theta \in \mathbb{R}^{d_\theta}$, and thus define the corresponding Gibbs distribution

$$\pi_\theta(x) = \frac{\exp(-V(x, \theta))}{Z(\theta)}, \qquad Z(\theta) := \int_{\mathbb{R}^{d_x}} \exp(-V(x, \theta)) \, dx. \tag{42}$$

We use $\pi_\theta$ to denote the Gibbs distribution, and in our setup the Gaussian-mixture distribution defined by (26). In particular, for (26) the normalising constant is notably independent of $\theta$. In the interest of completeness, and as done in Marion et al. (2025), we in fact present the general case and remark on simplifications that occur for our specific case.

Given a potentially non-differentiable reward $R : \mathbb{R}^{d_x} \to \mathbb{R}$ and a fixed reference distribution $\pi_{\mathrm{ref}}$, which is taken to be a frozen Gibbs distribution of the same form in our case, induced by $\theta_{\mathrm{ref}}$, we consider the following KL-regularised objective

$$\ell(\theta) = -\mathbb{E}_{\pi_\theta}[R(X)] + \beta_{\mathrm{KL}} \, \mathrm{KL}\left(\pi_{\mathrm{ref}} \| \pi_\theta\right), \qquad \beta_{\mathrm{KL}} > 0. \tag{43}$$

We expand the forward KL term,

$$\mathrm{KL}(\pi_{\mathrm{ref}} \| \pi_\theta) = \mathbb{E}_{\pi_{\mathrm{ref}}}[\log \pi_{\mathrm{ref}}(X) - \log \pi_\theta(X)], \tag{44}$$

so that $\theta$ appears only through $\log \pi_\theta$.

**Lemma 2** (Gibbs score identity). *For $\pi_\theta$ as outlined in* (42),

$$\nabla_\theta \log \pi_\theta(X) = -\nabla_\theta V(X, \theta) + \mathbb{E}_{\pi_\theta}[\nabla_\theta V(X, \theta)]. \tag{45}$$

*Proof.* We have that $\log \pi_\theta(X) = -V(X, \theta) - \log Z(\theta)$, and so

$$\nabla_\theta \log \pi_\theta(X) = -\nabla_\theta V(X, \theta) - \nabla_\theta \log Z(\theta).$$

Furthermore,

$$\nabla_\theta \log Z(\theta) = \frac{1}{Z(\theta)} \nabla_\theta Z(\theta) = \frac{1}{Z(\theta)} \int -\exp(-V(x, \theta)) \, \nabla_\theta V(x, \theta) \, dx = -\mathbb{E}_{\pi_\theta}[\nabla_\theta V(X, \theta)],$$

where in the second equality we utilise the chain rule. Substituting the above into the previous equation gives the desired result. $\square$

Indeed, the mixture potential (26) gives a $Z(\theta)$ independent of $\theta$, and so $\nabla_\theta \log Z(\theta) = 0$, which by Lemma 2 means that $\mathbb{E}_{\pi_\theta}[\nabla_\theta V(X, \theta)] = 0$.

Now, we note that

$$\begin{aligned}
\nabla_\theta \mathbb{E}_{\pi_\theta}[R(X)] &= \mathbb{E}_{\pi_\theta}[R(X) \, \nabla_\theta \log \pi_\theta(X)] \\
&= -\mathbb{E}_{\pi_\theta}[R(X) \, \nabla_\theta V(X, \theta)] + \mathbb{E}_{\pi_\theta}[R(X)] \, \mathbb{E}_{\pi_\theta}[\nabla_\theta V(X, \theta)].
\end{aligned} \tag{46}$$

where, in the second equality, we have substituted in (45). Although the second term is indeed zero here, we leave this in the covariance form to match Marion et al. (2025) and the respective code implementation.

Now, since $\pi_{\mathrm{ref}}$ does not depend on $\theta$, we have that

$$\begin{aligned}
\nabla_\theta \mathrm{KL}(\pi_{\mathrm{ref}} \| \pi_\theta) &= \nabla_\theta \mathbb{E}_{\pi_{\mathrm{ref}}}[\log \pi_{\mathrm{ref}}(X) - \log \pi_\theta(X)] \\
&= -\mathbb{E}_{\pi_{\mathrm{ref}}}[\nabla_\theta \log \pi_\theta(X)] \\
&= \mathbb{E}_{\pi_{\mathrm{ref}}}[\nabla_\theta V(X, \theta)] - \mathbb{E}_{\pi_\theta}[\nabla_\theta V(X, \theta)],
\end{aligned} \tag{47}$$

where again in final equality we have substituted in (45).

Thus, from (46) and (47), we have that

$$\nabla_\theta \ell(\theta) = \left(\mathbb{E}_{\pi_\theta}[R \, \nabla_\theta V] - \mathbb{E}_{\pi_\theta}[R]\mathbb{E}_{\pi_\theta}[\nabla_\theta V]\right) + \beta_{\mathrm{KL}} \left(\mathbb{E}_{\pi_{\mathrm{ref}}}[\nabla_\theta V] - \mathbb{E}_{\pi_\theta}[\nabla_\theta V]\right). \tag{48}$$

As discussed in detail within Appendix F.3, for the closely related EBM setting, in practice we explicitly treat particles as *stop-gradient* quantities and in fact utilise scalar *surrogate* losses to realise (48).

## F.2. Derivation of $\nabla_\theta \ell(\theta)$ for reward tuning of EBMs

Let $\pi_\theta$ denote an energy-based model (EBM), where $x \in \mathbb{R}^{d_x}$ and $\theta \in \mathbb{R}^{d_\theta}$, so that

$$\pi_\theta(x) = \frac{\exp(-E_\theta(x))}{Z_\theta}, \qquad Z_\theta := \int_{\mathbb{R}^{d_x}} \exp(-E_\theta(x)) \, dx. \tag{49}$$

Given a reward function $R : \mathbb{R}^{d_x} \to \mathbb{R}$, and a fixed reference distribution $\pi_0$, we consider the reverse-KL regularised objective,

$$\ell(\theta) = -\mathbb{E}_{\pi_\theta}[R(X)] + \beta_{\mathrm{KL}} \, \mathrm{KL}(\pi_\theta \| \pi_0), \qquad \beta_{\mathrm{KL}} > 0, \tag{50}$$

where, to be clear, the parameter $\beta_{\mathrm{KL}}$ controls how strongly we penalise deviation from the reference distribution.

Notably, for EBM's, the score $\nabla_\theta \log \pi_\theta(x)$ takes on a straightforward form, as outlined in Lemma 3.

**Lemma 3** (EBM score identity)**.** *For $\pi_\theta$, as outlined in* (49)*,*

$$\nabla_\theta \log \pi_\theta(x) = -\nabla_\theta E_\theta(x) + \mathbb{E}_{x' \sim \pi_\theta} \left[ \nabla_\theta E_\theta(x') \right]. \tag{51}$$

*Proof.* From (49) we have that $\pi_\theta(x) = \frac{\exp(-E_\theta(x))}{Z_\theta} \Rightarrow \log \pi_\theta(x) = -E_\theta(x) - \log Z_\theta$, and so

$$\nabla_\theta \log \pi_\theta(x) = -\nabla_\theta E_\theta(x) - \nabla_\theta \log Z_\theta.$$

Furthermore,

$$\nabla_\theta \log Z_\theta = \frac{1}{Z_\theta} \nabla_\theta Z_\theta = \frac{1}{Z_\theta} \int -\exp(-E_\theta(x)) \, \nabla_\theta E_\theta(x) \, dx = -\mathbb{E}_{\pi_\theta}[\nabla_\theta E_\theta(x)].$$

Substituting the above two relations results in (51). $\qquad\square$

This allows us to express the gradients of an expectation as follows.

**Lemma 4** (Gradient of an expectation)**.** *If $f : \mathbb{R}^{d_x} \to \mathbb{R}$ is independent of $\theta$ and integrable under $\pi_\theta$, then we have*

$$\nabla_\theta \mathbb{E}_{\pi_\theta}[f(x)] = - \left( \mathbb{E}_{\pi_\theta}[f(x) \, \nabla_\theta E_\theta(x)] - \mathbb{E}_{\pi_\theta}[f(x)] \, \mathbb{E}_{\pi_\theta}[\nabla_\theta E_\theta(x)] \right). \tag{52}$$

*Proof.* We have that

$$\begin{aligned}
\nabla_\theta \mathbb{E}_{\pi_\theta}[f(x)] = \nabla_\theta \int_{\mathbb{R}^{d_x}} f(x) \pi_\theta(x) \, dx &= \int_{\mathbb{R}^{d_x}} f(x) \nabla_\theta \pi_\theta(x) \, dx \\
&= \int_{\mathbb{R}^{d_x}} f(x) \pi_\theta(x) \nabla_\theta \log \pi_\theta(x) \, dx \\
&= \mathbb{E}_{\pi_\theta} \left[ f(x) \nabla_\theta \log \pi_\theta(x) \right] \\
&= \mathbb{E}_{\pi_\theta} \left[ f(x) \left( -\nabla_\theta E_\theta(x) + \mathbb{E}_{\pi_\theta} \left[ \nabla_\theta E_\theta(x) \right] \right) \right] \\
&= \mathbb{E}_{\pi_\theta} \left[ -f(x) \nabla_\theta E_\theta(x) \right] + \mathbb{E}_{\pi_\theta} \left[ f(x) \right] \mathbb{E}_{\pi_\theta} \left[ \nabla_\theta E_\theta(x) \right] \\
&= - \left( \mathbb{E}_{\pi_\theta} \left[ f(x) \nabla_\theta E_\theta(x) \right] - \mathbb{E}_{\pi_\theta} \left[ f(x) \right] \mathbb{E}_{\pi_\theta} \left[ \nabla_\theta E_\theta(x) \right] \right),
\end{aligned}$$

where we have utilised Lemma 3 in the fourth equality, giving us the desired identity. $\qquad\square$

First we consider the gradient of the reward term, $\nabla_\theta \mathbb{E}_{\pi_\theta}[R(x)]$, which we obtain through leveraging Lemma 4, with $f(x) = R(x)$, resulting in

$$\nabla_\theta \mathbb{E}_{\pi_\theta}[R(x)] = - \left( \mathbb{E}_{\pi_\theta}[R(x) \, \nabla_\theta E_\theta(x)] - \mathbb{E}_{\pi_\theta}[R(x)] \, \mathbb{E}_{\pi_\theta}[\nabla_\theta E_\theta(x)] \right). \tag{53}$$

Then, to obtain the gradient of the KL term, $\nabla_\theta \text{KL}\left(\pi_\theta \| \pi_0\right)$, we first note that

$$\text{KL}\left(\pi_\theta \| \pi_0\right) = \mathbb{E}_{\pi_\theta}\left[\log \pi_\theta(x) - \log \pi_0(x)\right],$$

$$
\begin{aligned}
\Rightarrow \nabla_\theta \text{KL}\left(\pi_\theta \| \pi_0\right) &= \nabla_\theta \mathbb{E}_{\pi_\theta}\left[\log \pi_\theta(x) - \log \pi_0(x)\right] \\
&= \mathbb{E}_{\pi_\theta}\left[\nabla_\theta \log \pi_\theta(x)\right] + \mathbb{E}_{\pi_\theta}\left[\left(\log \pi_\theta(x) - \log \pi_0(x)\right)\nabla_\theta \log \pi_\theta(x)\right] \\
&= \mathbb{E}_{\pi_\theta}\left[\left(\log \pi_\theta(x) - \log \pi_0(x)\right)\nabla_\theta \log \pi_\theta(x)\right],
\end{aligned}
$$

where we have used the product rule for the gradient of the expectation in the third equality, whilst in the fourth equality we have used the fact that $\mathbb{E}_{\pi_\theta}\left[\nabla_\theta \log \pi_\theta(x)\right] = \int \pi_\theta(x)\frac{\nabla_\theta \pi_\theta(x)}{\pi_\theta(x)}\,dx = \nabla_\theta \int \pi_\theta(x)\,dx = \nabla_\theta(1) = 0$, that is, the score has zero mean.

Now, subbing in the logarithmic form of the first part of (49), into the above, yields

$$
\begin{aligned}
\nabla_\theta \text{KL}\left(\pi_\theta \| \pi_0\right) &= \mathbb{E}_{\pi_\theta}\left[\left\{\left(-E_\theta(x) - \log Z_\theta\right) - \left(-E_0(x) - \log Z_0\right)\right\}\nabla_\theta \log \pi_\theta(x)\right] \\
&= -\mathbb{E}_{\pi_\theta}\left[\left(E_\theta(x) - E_0(x)\right)\nabla_\theta \log \pi_\theta(x)\right] - \left(\log Z_\theta - \log Z_0\right)\mathbb{E}_{\pi_\theta}\left[\nabla_\theta \log \pi_\theta(x)\right] \\
&= -\mathbb{E}_{\pi_\theta}\left[\left(E_\theta(x) - E_0(x)\right)\nabla_\theta \log \pi_\theta(x)\right],
\end{aligned}
\tag{54}
$$

where we have again used the fact that the score has zero mean.

Subsequently, utilising Lemma 3 and expanding terms results in

$$\nabla_\theta \text{KL}\left(\pi_\theta \| \pi_0\right) = \mathbb{E}_{\pi_\theta}\left[\left(E_\theta(x) - E_0(x)\right)\nabla_\theta E_\theta(x)\right] - \mathbb{E}_{\pi_\theta}\left[E_\theta(x) - E_0(x)\right]\mathbb{E}_{\pi_\theta}\left[\nabla_\theta E_\theta(x)\right]. \tag{55}$$

Indeed, subbing in (53) and (55) into the gradient of (50) gives

$$
\begin{aligned}
\nabla_\theta \ell(\theta) &= -\nabla_\theta \mathbb{E}_{x \sim \pi_\theta}[R(x)] + \beta_{\text{KL}}\,\nabla_\theta \text{KL}(\pi_\theta \| \pi_0) \\
&= \left(\mathbb{E}_{\pi_\theta}[R(x)\nabla_\theta E_\theta(x)] - \mathbb{E}_{\pi_\theta}[R(x)]\mathbb{E}_{\pi_\theta}[\nabla_\theta E_\theta(x)]\right) \\
&\quad + \beta_{\text{KL}}\left(\mathbb{E}_{\pi_\theta}\left[\left(E_\theta(x) - E_0(x)\right)\nabla_\theta E_\theta(x)\right] - \mathbb{E}_{\pi_\theta}\left[E_\theta(x) - E_0(x)\right]\mathbb{E}_{\pi_\theta}\left[\nabla_\theta E_\theta(x)\right]\right),
\end{aligned}
\tag{56}
$$

which can also be concisely expressed using the quantity $A_\theta(x) = R(x) + \beta_{\text{KL}}\left(E_\theta(x) - E_0(x)\right)$,

$$\nabla_\theta \ell(\theta) = \mathbb{E}_{\pi_\theta}\left[A_\theta(x)\nabla_\theta E_\theta(x)\right] - \mathbb{E}_{\pi_\theta}\left[A_\theta(x)\right]\mathbb{E}_{\pi_\theta}\left[\nabla_\theta E_\theta(x)\right]. \tag{57}$$

### F.3. Motivation of surrogate losses for reward tuning of EBMs

Although conceptually straightforward, particularly when presented in the concise covariance form outlined in (57), directly optimising the objective $\ell(\theta)$ via automatic differentiation in an efficient manner is not trivial. Specifically, the difficulty is not in deriving $\nabla_\theta \ell(\theta)$, but rather in realising this gradient when expectations under $\pi_\theta$ are approximated from MCMC.

Indeed, $\nabla_\theta \ell(\theta)$ is a distributional derivative, that is it describes how the (stationary) model density $\pi_\theta$ changes w.r.t. $\theta$. In particular, not only are local changes in the energy $E_\theta(x)$, at fixed $x$, accounted for, but so is the global re-weighting of the probability mass induced by the $\theta$ dependence of the partition function $Z(\theta)$. In fact, this dependence exists implicitly through the score $\nabla_\theta \log \pi_\theta(x)$, where $\nabla_\theta \log \pi_\theta(x) = -\nabla_\theta E_\theta(x) - \nabla_\theta \log Z_\theta$, and since $\nabla_\theta \log Z_\theta = -\mathbb{E}_{\pi_\theta}[\nabla_\theta E_\theta(x)]$, the intractable partition function contributes implicitly as a mean-centering term in the score, and is consequently responsible for the covariance structure of the exact gradient, $\nabla_\theta \ell(\theta)$.

In contrast, samples from $\pi_\theta$ are typically generated by a finite number of steps of a Markov chain, where unrolling the respective sampler, and differentiating through the resulting computation graph, results in a path-wise gradient of the finite-step sampling distribution. Indeed, this depends upon the step size and number of steps specified. Crucially, such gradients do not, in general, correspond to the score-function gradient of the (stationary) distribution $\pi_\theta$, but instead optimise a different, sampler-dependent objective. It is for this reason that MCMC samples are treated as stop-gradient inputs and we instead rely on analytical score identities to obtain gradients w.r.t. $\theta$.

Given particles $\{X_i\}_{i=1}^N$, approximately distributed as $\pi_\theta$, a natural Monte Carlo estimator of $\nabla_\theta \ell(\theta)$ is thus

$$\widehat{\nabla_\theta \ell} = \frac{1}{N}\sum_{i=1}^N A_\theta(X_i)\nabla_\theta E_\theta(X_i) - \left(\frac{1}{N}\sum_{i=1}^N A_\theta(X_i)\right)\left(\frac{1}{N}\sum_{i=1}^N \nabla_\theta E_\theta(X_i)\right), \tag{58}$$

where we note that the same-sample plug-in estimator is consistent, given exact i.i.d. samples from $\pi_\theta$, up to usual finite-sample covariance bias. Implementing this estimator efficiently in current automatic differentiation frameworks is not straightforward. First, note that (58) requires per-sample gradients, $\nabla_\theta E_\theta(X_i) \in \mathbb{R}^{d_\theta}$, which can involve a significantly greater number of backward passes and is thus more computationally expensive. Furthermore, a more nuanced subtlety arises from the fact that the covariance multiplier, $A_\theta(x) = R(x) + \beta_{\mathrm{KL}}(E_\theta(x) - E_0(x))$ depends explicitly on $\theta$. Indeed, whilst in the analytical gradient $\nabla_\theta \ell(\theta)$, $A_\theta(x)$ is considered a scalar weight multiplying the score $\nabla_\theta \log \pi_\theta(x)$, a naive implementation of automatic differentiation that propagates through both factors would introduce additional terms from differentiating $A_\theta(x)$ itself. To be clear, for the reverse-KL term, differentiating through the factor $E_\theta(x) - E_0(x)$ results in additional contributions that are absent from the true score-function gradient and correspond to optimising a distinctly different objective. That is, the reverse-KL term is particularly prone to incorrect differentiation in practice.

### F.4. Derivation of surrogate losses for reward tuning of EBMs

Instead of optimising the objective outlined in (50), using the exact gradient derived in Appendix F.2, we instead derive scalar *surrogate* losses, whose gradients match that of the intended particle approximation of $\nabla_\theta \mathbb{E}_{\pi_\theta}[R(x)]$ and $\nabla_\theta \mathrm{KL}(\pi_\theta \| \pi_0)$ respectively, whilst permitting standard automatic differentiation. The key to deriving these surrogates is through recognising that the weighted sums, of per-sample energy gradients, can be equivalently expressed as the gradient of a scalar energy surrogate, provided that the respective multipliers are considered as constants during differentiation.

**Lemma 5** (Surrogate loss form). *Let $a_\theta(x)$ be a scalar multiplier function and $\{(X_i, w_i)\}_{i=1}^N$ denote particles, with respective normalised weights, approximating $\pi_\theta$. We define the surrogate loss as*

$$\widehat{\mathcal{L}}_a(\theta) = \sum_{i=1}^N w_i \, \mathrm{stopgrad}\left\{ a_\theta(X_i) - \sum_{j=1}^N w_j a_\theta(X_j) \right\} E_\theta(X_i), \tag{59}$$

*where $\mathrm{stopgrad}(\cdot)$ denotes the treatment of the argument as constant during differentiation.*

*Consequently,*

$$\nabla_\theta \widehat{\mathcal{L}}_a(\theta) = \sum_{i=1}^N w_i \, a_\theta(X_i) \, \nabla_\theta E_\theta(X_i) - \left( \sum_{j=1}^N w_j a_\theta(X_j) \right) \left( \sum_{i=1}^N w_i \, \nabla_\theta E_\theta(X_i) \right) \tag{60}$$

*Proof.* Since $\mathrm{stopgrad}\left\{ a_\theta(X_i) - \sum_{j=1}^N w_j a_\theta(X_j) \right\}$ is detached, the only remaining $\theta$-dependence exists in $E_\theta$,

$$\nabla_\theta \widehat{\mathcal{L}}_a(\theta) = \nabla_\theta \sum_{i=1}^N w_i \, \mathrm{stopgrad}\left\{ a_\theta(X_i) - \sum_{j=1}^N w_j a_\theta(X_j) \right\} E_\theta(X_i),$$

$$= \sum_{i=1}^N w_i \, \mathrm{stopgrad}\left\{ a_\theta(X_i) - \sum_{j=1}^N w_j a_\theta(X_j) \right\} \nabla_\theta E_\theta(X_i),$$

$$= \sum_{i=1}^N w_i \, a_\theta(X_i) \, \nabla_\theta E_\theta(X_i) - \left( \sum_{j=1}^N w_j a_\theta(X_j) \right) \left( \sum_{i=1}^N w_i \, \nabla_\theta E_\theta(X_i) \right),$$

as required. $\square$

Now, setting $a_\theta(x) = R(x)$ explicitly gives, by Lemma 5,

$$\widehat{\mathcal{L}}_{\mathrm{Rew}}(\theta) = \sum_{i=1}^N w_i \, \mathrm{stopgrad}\left\{ R(X_i) - \sum_{j=1}^N w_j R(X_j) \right\} E_\theta(X_i), \tag{61}$$

$$\Rightarrow \nabla_\theta \widehat{\mathcal{L}}_{\mathrm{Rew}}(\theta) = \sum_{i=1}^N w_i \, R(X_i) \, \nabla_\theta E_\theta(X_i) - \left( \sum_{j=1}^N w_j R(X_j) \right) \left( \sum_{i=1}^N w_i \, \nabla_\theta E_\theta(X_i) \right), \tag{62}$$

where $\nabla_\theta \widehat{\mathcal{L}}_{\text{Rew}}(\theta)$ is thus the particle approximation of $-\nabla_\theta \mathbb{E}_{\pi_\theta}[R(x)]$.

Next, setting $a_\theta(x) = E_\theta(x) - E_0(x)$ explicitly gives, again by Lemma 5,

$$\widehat{\mathcal{L}}_{\text{KL}}(\theta) = \sum_{i=1}^N w_i \, \text{stopgrad} \left\{ E_\theta(X_i) - E_0(X_i) - \left( \sum_{j=1}^N w_j \left( E_\theta(X_j) - E_0(X_j) \right) \right) \right\} E_\theta(X_i), \tag{63}$$

$$\Rightarrow \nabla_\theta \widehat{\mathcal{L}}_{\text{KL}}(\theta) = \sum_{i=1}^N w_i \left( E_\theta(X_i) - E_0(X_i) \right) \nabla_\theta E_\theta(X_i) - \left( \sum_{j=1}^N w_j \left( E_\theta(X_j) - E_0(X_j) \right) \right) \left( \sum_{i=1}^N w_i \, \nabla_\theta E_\theta(X_i) \right),$$
$$\tag{64}$$

where, to be clear, $\nabla_\theta \widehat{\mathcal{L}}_{\text{KL}}(\theta)$ is the particle approximation of $\nabla_\theta \text{KL}\left(\pi_\theta \| \pi_0\right)$.

Thus, if $\widehat{\mathcal{L}}_{\text{total}} = \widehat{\mathcal{L}}_{\text{Rew}}(\theta) + \beta_{\text{KL}} \widehat{\mathcal{L}}_{\text{KL}}(\theta)$, then we have, by (62) and (64), that $\nabla_\theta \widehat{\mathcal{L}}_{\text{total}}$ is the particle approximation of $\nabla_\theta \ell(\theta)$, which means that descent on $\widehat{\mathcal{L}}_{\text{total}}(\theta)$ implements descent using the intended particle approximation of $\nabla_\theta \ell(\theta)$.

### F.5. Derivation of $\nabla_\theta \ell$ as expectation over $\pi_\theta$ for reward tuning

For completeness, we outline how the gradient of the loss, for both the forward-KL and reverse-KL regularised objectives, can be expressed as an expectation over $\pi_\theta$, as in (2). For clarity, we follow the notation of Appendix F.1 and F.2 respectively.

In the case of the forward-KL regularised objective, recall we have that

$$\ell(\theta) = -\mathbb{E}_{\pi_\theta}[R(x)] + \beta_{\text{KL}} \, \text{KL}\left(\pi_{\text{ref}} \| \pi_\theta\right), \qquad \beta_{\text{KL}} > 0,$$

where, from the first equality of (46), we note

$$\nabla_\theta \mathbb{E}_{\pi_\theta}[R(x)] = \mathbb{E}_{\pi_\theta}[R(x) \nabla_\theta \log \pi_\theta(x)],$$

whilst, from the last equality of (47),

$$\nabla_\theta \text{KL}(\pi_{\text{ref}} \| \pi_\theta) = \mathbb{E}_{\pi_{\text{ref}}}[\nabla_\theta V(x, \theta)] - \mathbb{E}_{\pi_\theta}[\nabla_\theta V(x, \theta)],$$

and thus combine accordingly, to obtain

$$\nabla_\theta \ell(\theta) = -\mathbb{E}_{\pi_\theta}[R(x) \nabla_\theta \log \pi_\theta(x)] + \beta_{\text{KL}} \left( \mathbb{E}_{\pi_{\text{ref}}}[\nabla_\theta V(x, \theta)] - \mathbb{E}_{\pi_\theta}[\nabla_\theta V(x, \theta)] \right),$$

that is, we have

$$H_\theta(\cdot) = -R(\cdot) \nabla_\theta \log \pi_\theta(\cdot) + \beta_{\text{KL}} \left( \mathbb{E}_{y \sim \pi_{\text{ref}}}[\nabla_\theta V(y, \theta)] - \nabla_\theta V(\cdot, \theta) \right).$$

In the case of the reverse-KL regularised objective, recall we instead have

$$\ell(\theta) = -\mathbb{E}_{\pi_\theta}[R(X)] + \beta_{\text{KL}} \, \text{KL}(\pi_\theta \| \pi_0), \qquad \beta_{\text{KL}} > 0,$$

and again note

$$\nabla_\theta \mathbb{E}_{\pi_\theta}[R(x)] = \mathbb{E}_{\pi_\theta}[R(x) \nabla_\theta \log \pi_\theta(x)].$$

In contrast, we instead have, from (54), that

$$\nabla_\theta \text{KL}\left(\pi_\theta \| \pi_0\right) = -\mathbb{E}_{\pi_\theta}\left[ \left( E_\theta(x) - E_0(x) \right) \nabla_\theta \log \pi_\theta(x) \right],$$

and so obtain

$$\nabla_\theta \ell(\theta) = -\mathbb{E}_{\pi_\theta}[R(x) \nabla_\theta \log \pi_\theta(x)] - \beta_{\text{KL}} \mathbb{E}_{\pi_\theta}\left[ \left( E_\theta(x) - E_0(x) \right) \nabla_\theta \log \pi_\theta(x) \right],$$

that is, we have

$$H_\theta(\cdot) = -\left[ R(\cdot) + \beta_{\text{KL}} \left( E_\theta(\cdot) - E_0(\cdot) \right) \right] \nabla_\theta \log \pi_\theta(\cdot).$$

Therefore, both the forward-KL and reverse-KL regularised objectives fit the form of (2) as required.

## G. Experimental Details - Quantitative Results

Here, we collate quantitative results for the reward tuning of Langevin processes experiment (see Appendix E.1) and the Bayesian image deblurring experiment (see Appendix E.4). For full experimental details, refer to the respective appendices.

*Table 1.* Terminal reward values, across Langevin processes problem settings, with $\beta_{\mathrm{KL}} = 0$ in all cases.

| Method | OPT | Problem Setting | | | |
|---|---|---|---|---|---|
| | | $V_{\mathrm{dual}}$ & $R_{\mathrm{smooth}}$ | $V_{\mathrm{dual}}$ & $R_{\mathrm{hard}}$ | $V_{\mathrm{sparse}}$ & $R_{\mathrm{hard}}$ | $V_{\mathrm{tight}}$ & $R_{\mathrm{tight}}$ |
| IMPDIFF | SGD | $0.6364 \pm 0.0009$ | $0.0095 \pm 0.0001$ | $0.0173 \pm 0.0001$ | $0.0115 \pm 0.0008$ |
| | ADAM | $0.6026 \pm 0.0013$ | $0.0169 \pm 0.0002$ | $0.0363 \pm 0.0001$ | $0.0142 \pm 0.0012$ |
| SOUL | SGD | $0.5714 \pm 0.1270$ | $0.0059 \pm 0.0055$ | $0.0173 \pm 0.0044$ | $0.0000 \pm 0.0000$ |
| | ADAM | $0.6477 \pm 0.0037$ | $0.0187 \pm 0.0003$ | $0.0365 \pm 0.0008$ | $0.0000 \pm 0.0000$ |
| SOSMC-ULA | SGD | $0.6434 \pm 0.0010$ | $0.0133 \pm 0.0002$ | $0.0224 \pm 0.0003$ | $0.1327 \pm 0.0994$ |
| | ADAM | $0.6484 \pm 0.0007$ | $0.0186 \pm 0.0001$ | $0.0369 \pm 0.0001$ | $0.2497 \pm 0.0869$ |
| SOSMC-MALA | SGD | $0.6435 \pm 0.0013$ | $0.0128 \pm 0.0001$ | $0.0204 \pm 0.0002$ | $0.3788 \pm 0.0006$ |
| | ADAM | $0.6491 \pm 0.0006$ | $0.0185 \pm 0.0000$ | $0.0368 \pm 0.0002$ | $0.3994 \pm 0.0000$ |
| SOSMC-RWM | SGD | $0.6424 \pm 0.0015$ | $0.0129 \pm 0.0002$ | $0.0207 \pm 0.0003$ | $0.3803 \pm 0.0004$ |
| | ADAM | $0.6476 \pm 0.0010$ | $0.0186 \pm 0.0000$ | $0.0368 \pm 0.0002$ | $0.3994 \pm 0.0000$ |
| SOSMC-HMC | SGD | $0.6416 \pm 0.0008$ | $0.0127 \pm 0.0001$ | $0.0202 \pm 0.0003$ | $0.3775 \pm 0.0009$ |
| | ADAM | $0.6485 \pm 0.0011$ | $0.0183 \pm 0.0004$ | $0.0366 \pm 0.0001$ | $0.3994 \pm 0.0000$ |

*Table 2.* Terminal reward values, across Langevin processes problem settings, with $\beta_{\mathrm{KL}} = 0.1$ in the cases of $V_{\mathrm{dual}}$ & $R_{\mathrm{smooth}}$ and $V_{\mathrm{tight}}$ & $R_{\mathrm{tight}}$, whilst $\beta_{\mathrm{KL}} = 0.001$ in the cases of $V_{\mathrm{dual}}$ & $R_{\mathrm{hard}}$ and $V_{\mathrm{sparse}}$ & $R_{\mathrm{hard}}$.

| Method | OPT | Problem Setting | | | |
|---|---|---|---|---|---|
| | | $V_{\mathrm{dual}}$ & $R_{\mathrm{smooth}}$ | $V_{\mathrm{dual}}$ & $R_{\mathrm{hard}}$ | $V_{\mathrm{sparse}}$ & $R_{\mathrm{hard}}$ | $V_{\mathrm{tight}}$ & $R_{\mathrm{tight}}$ |
| IMPDIFF | ADAM | $0.5766 \pm 0.0013$ | $0.0173 \pm 0.0002$ | $0.0357 \pm 0.0002$ | $0.0102 \pm 0.0003$ |
| SOUL | ADAM | $0.3680 \pm 0.1997$ | $0.0132 \pm 0.0057$ | $0.0367 \pm 0.0010$ | $0.0000 \pm 0.0000$ |
| SOSMC-ULA | ADAM | $0.6020 \pm 0.0010$ | $0.0181 \pm 0.0000$ | $0.0361 \pm 0.0002$ | $0.2002 \pm 0.0854$ |
| SOSMC-MALA | ADAM | $0.6036 \pm 0.0012$ | $0.0181 \pm 0.0001$ | $0.0361 \pm 0.0001$ | $0.3220 \pm 0.0004$ |
| SOSMC-RWM | ADAM | $0.6016 \pm 0.0012$ | $0.0181 \pm 0.0001$ | $0.0362 \pm 0.0001$ | $0.3216 \pm 0.0004$ |
| SOSMC-HMC | ADAM | $0.6033 \pm 0.0009$ | $0.0181 \pm 0.0000$ | $0.0360 \pm 0.0002$ | $0.3218 \pm 0.0004$ |

*Table 3.* Terminal evaluation metrics, across Bayesian image deblurring setups, with $\lambda = 0.4$ in all cases.

| Method | Setup A | | | Setup B | | | Setup C | | |
|---|---|---|---|---|---|---|---|---|---|
| | $\hat{\theta}$ | MSE | SSIM | $\hat{\theta}$ | MSE | SSIM | $\hat{\theta}$ | MSE | SSIM |
| MYPGD | $-3.30$ | $312.71$ | $0.74$ | $-3.18$ | $220.42$ | $0.75$ | $-3.94$ | $1520.36$ | $0.22$ |
| SOSMC-MYULA | $-3.48$ | $215.47$ | $0.80$ | $-3.23$ | $136.63$ | $0.81$ | $-3.78$ | $1030.20$ | $0.41$ |

