# OpenReview forum: "Efficient Stochastic Optimisation via Sequential Monte Carlo"
_ICML.cc/2026/Conference — ICML 2026 regular_

### Official Review · Reviewer_kNru · 2026-02-20

**Soundness:** 4
**Presentation:** 4
**Significance:** 2
**Originality:** 3
**Overall Recommendation:** 5
**Confidence:** 3

**Summary:**

The authors introduce an algorithm---stochastic optimization via sequential Monte Carlo (SOSMC)---that leverages SMC to optimize functions with intractable gradients. As such the proposed method can be used to, for example, maximize the marginal likelihood of latent variable models or do reward tuning in energy-based models (EBMs). The proposed algorithm is presented as a general framework that unifies/generalizes methods previously proposed in the literature, including for example the JALA-EM algorithm (Cuin et al. 2025). The authors also provide some theoretical characterization of their proposed method, in particular a convergence result for an idealized version of SOSMC as well as an analysis of the dependence of the effective sample size (ESS) on the step size. The basic viability of SOSMC is demonstrated in a few small-to-medium scale experiments, where the method is shown to perform favorably against baseline methods from the literature.

**Compliance With Llm Reviewing Policy:**

Affirmed.

**Final Justification:**

The additional empirical results strengthen my conviction in the value of this submission. I recommend acceptance due to the conceptual and theoretical unification of existing algorithms provided, and the catalyst this could provide for future work in this area.

**Key Questions For Authors:**

My key questions can be found above in my discussion of "Soundness" in the "Strengths And Weaknesses" section.

Additional questions:
- Can you explain the erratic/oscillatory behavior of ImpDiff in Figure 1?
- Why not use systematic resampling?

**Limitations:**

yes

**Strengths And Weaknesses:**

**Soundness:**

As best I can tell---I haven't gone through the proofs in detail---the submission is technically sound, both with respect to theoretical claims and experimental setup. That said I would have liked to see more from the authors on both fronts.

In particular, with respect to the theoretical claims, the idealized convergence result is encouraging, but of course it is not directly relevant to the proposed algorithm. Moreover it seems to me that Remark 3 underplays/undercontextualizes the technical difficulty of going beyond the idealized theorem. The remark highlights bias/variance, but it seems to me like the hairiest theoretical obstacle would be the fact that the stochastic gradient estimates are *correlated* across steps. Can the authors please comment?

Similarly the discussion in Sec 4.2 and Remark 4 are helpful, and the proposed adaptive step size scheme is sensible, but the broader algorithmic questions are left largely undertheorized. In particular how are choices for forward/backward kernels linked to choices for the gradient update rule? Can we say for example, that kernels used with Adam should exhibit property X or that kernels used with SGD should have property Y? Can we say that a gradient update rule needs to (or should) satisfy some condition Z if it's to be plugged into SOSMC? Can the authors please comment on this broader set of theoretical questions?

Finally, while the experiments are carefully constructed and support the general viability of the method, they are not particular extensive. The number of baselines considered is minimal and the largest scale experiment uses MNIST. If we consider that CNNs were first trained on MNIST in 1998 and that nowadays LLMs are trained on large chunks of the internet, it would be hard to argue that the experiment in Sec 5.3 constitutes a large-scale experiment. Can the authors not think of additional experiments or baseline methods that would strengthen the empirical evaluation in the paper? For example, I was disappointed to find that Maximum marginal likelihood estimation (MMLE) was not included in the experiments. The authors claim generality of the framework but provide empirical evaluation in a rather limited problem space.

**Presentation:**

The presentation is generally clear and the discussion of related work seems appropriate. However, I do have a few concrete suggestions:

- Define gamma_k after eqn 3
- I'm missing conditions on forward/backward kernels in sec 3.1. Granted space in the main text is at a premium, but it seems to me that at a minimum you should cite something here or point to a section in the appendix where you give more details. Remark #2 isn't really sufficient.
- Dashed/solid lines are hard to distinguish in figure 1. It's also unclear how the reader is supposed to know how R/V_dual/sparse differ on the basis of sec 5.1. The reader could use some context/interpretation for how two problems differ.
- All experimental results are reported in an essentially visual way. Can't you include quantitative results? Tables with NLLs or rewards or KLs or the like?
- Bonus points for spelling Łojasiewicz correctly

**Significance:**

Given the somewhat limited punch of the theoretical results, to me the main significance of this submission is in generalizing/unifying other approaches in the literature. As such it suggests future areas of research (both theoretical and more narrowly algorithmic) and as such I think it makes a valuable contribution that could be of interest to many in the ICML community. In my opinion a significance score of "good" or "excellence" would require a more thoroughgoing theoretical analysis.

**Originality:**

As far as I can tell the main ideas presented in the submission, in particular its unification of some existing methods, is novel; however, I am not sufficiently familiar with all the relevant literature to assess the originality of this work in fine-grained detail.

---

> ### Author Rebuttal · Authors · 2026-03-31
>
> We thank the reviewer for their thoughtful review.
>
> **1. Discussion on kernels**
>
> > [...] missing conditions on forward/backward kernels in sec 3.1. [...] Remark #2 isn't really sufficient.
>
> Backward kernels can be chosen flexibly whereas forward kernels are MCMC kernels. The forward kernels $K_k$ should admit densities w.r.t. a reference measure, and that the backward kernels $L_{k-1}$ must be chosen such that the importance weights are well-defined.
>
> To expand Rem. 2, we have now added three more choices of kernels, namely, MALA, RWM, and HMC kernels, into one of our experiments. In order to save space, we refer the reviewer to our **Response to Reviewer Lrtn, #1, Discussion on kernels**. We will incorporate all of this into our manuscript.
>
> **2. Theoretical aspects**
>
> > [...] the idealized convergence result is encouraging, [...] obstacle would be the fact that the stochastic gradient estimates are correlated across steps.
>
> We appreciate that our analysis is idealised. For finite-$N$ analysis, we point to an analysis done by Cuin et al. (2025), for MMLE in a simplified setting of no-resampling. Similar to Cuin et al. (2025), one can exploit biased optimization results, e.g., in Demidovich et al. (2025) which do not require gradient independence.
>
> An analysis in full generality is difficult due to several factors. For example, we note that Cuin et al. (2025) treats the path as pre-specified, whereas in reality the targets are chosen on-the-fly. A fully general finite $N$ analysis requires a mean-field particle analysis, akin to Del Moral and Rio (2011). This requires a significant effort, which we hope to study in a separate work.
>
> > [...] How are choices for forward/backward kernels linked to choices for the gradient update rule? [...]
>
> As we noted above a dedicated effort is required to clarify these aspects due to complicated structure in our SMC method. However, as we noted we clarify some of these concerns empirically, please see **Response to Reviewer Lrtn, #1, Discussion on kernels**.
>
> **3. Additional experiment**
>
> > [...] additional experiments or baseline methods that would strengthen the empirical evaluation in the paper? [...]
>
> We fully agree with the reviewer, and have thus added an image deblurring experiment in a nonsmooth latent-variable model with TV regularisation, following Encinar et al. (2025). This is an ill-conditioned inverse problem, the latent variable being the clean image $x$, and $y$ the blurred and noisy observation, while the parameter $\theta$ controls the TV-prior strength:
> $$
> p_\theta(x\mid y)\propto\exp[-\Vert y-Hx\Vert^2/(2\sigma^2)-e^\theta TV(x)],
> $$
> so that both the latent image and regularisation level must be inferred from blurred observations. In this setup, we compare Moreau–Yosida PGD (MYPGD) to a (new) MYULA variant of SOSMC, which is natural for the nonsmooth TV-prior. The considered experiments, where both the true image and blurring applied is altered, are visualised at https://github.com/icmlsosmc/icml_sosmc, and highlight consistent superior performance for SOSMC-MYULA, for not only image reconstruction quality, via the MSE and the structural similarity index  measure (Figs. ID3, ID7, ID11), but also uncertainty localisation around meaningful image structure (Figs. ID4, ID8). Furthermore, final particle clouds are visibly sharper for SOSMC-MYULA (Figs. ID2, ID6, ID10), whilst posterior mean reconstruction is closer to the ground truth.
>
> **4. Other suggestions and fixes**
>
> > Dashed/solid lines are hard to distinguish in figure 1. [...] how R/V_dual/sparse differ on the basis of sec 5.1.
>
> With the new kernel variants of SOSMC (see **Response #1 to Reviewer Lrtn**), we separate results by optimiser into distinct panels. Second, we have prepared Fig. LP5 (see https://github.com/icmlsosmc/icml_sosmc), which we will add to Appendix E.1 of our paper, to visualise how problems differ.
>
> > All experimental results are reported in an essentially visual way. Can't you include quantitative results?
>
> Thanks, we will include this.
>
> > Why not use systematic resampling?
>
> Thank you for this suggestion. This would indeed reduce the variance, we have simply gone by the simplest choice.
>
> > [...] erratic/oscillatory behavior of ImpDiff in Figure 1?
>
> We think ImpDiff’s oscillatory behaviour is caused by the fact that parameter updates lack a "correction" mechanism after a single Langevin step, so particles lag behind the current target, creating a delayed gradient and oscillatory optimisation dynamics. However, this also depends on the loss, so a definitive answer is difficult.
>
> We also thank for pointing out various other issues, we will fix all of them.
>
> Encinar et al. (2025). Proximal Interacting Particle Langevin Algorithms.
>
> Cuin et al. (2025). Learning Latent Variable Models via Jarzynski-adjusted Langevin Algorithm.
>
> Demidovich et al. (2023). A Guide Through the Zoo of Biased SGD.
>
> Del Moral & Rio (2011). Concentration inequalities for mean field particle models.

---

> > ### Author Rebuttal · Reviewer_kNru · 2026-03-31
> >
> > I thank the authors for their detailed response and am encouraged by the additional experimental results and exploration of different kernels; I have raised my score accordingly.

---

### Official Review · Reviewer_Lrtn · 2026-03-05

**Soundness:** 2
**Presentation:** 3
**Significance:** 3
**Originality:** 2
**Overall Recommendation:** 3
**Confidence:** 3

**Summary:**

The paper develops a general SMC based method for estimating intractable gradients of loss-functions assuming energy-based parametrizations alongside a gradient-based optimization method, eliminating the need for computationally expensive inner-loop estimators. Theoretical discussions of algorithm convergence assuming access to the exact expectations in the Feynman-Kac identity and effective sample size behaviour assuming gaussian parametrizations are provided. Additionally, the algorithm is specified using ULA kernels and applied to the fine tuning of three EBMs, demonstrating its superiority over previous algorithms.

**Compliance With Llm Reviewing Policy:**

Affirmed.

**Final Justification:**

The authors provided sufficient additional experiments and explanations on experiment design choices, so as promised I am happy to adjust my overall recommendation by a mark. However, the issue of lack of theoretical depth/new insights driven from theory persists so that I would not go higher.

**Key Questions For Authors:**

Q1: Have the authors tried out different choices of kernels? If so, what are good choices and are there kernels that are better adapted to certain kind of problems? If not, is there a specific reason as to why they only tried ULA and how does this significantly differ from previous literature?

Q2: Why did the authors choose to compare their algorithm to SOUL and ImpDiff? Why not other algorithms and why did they omit SOUL from the second and third experiment?

Q3: Did the authors apply their algorithm to different problems than the fine tuning of EBMs? If not, how would they expect their algorithm to perform on different problems?

Should the authors provide revisions to their paper, possibly by providing additional experimental material, that clarify these key questions, I would be happy to adjust my overall recommendation by a mark. Additionally, although I am unsure if this can be done without a major revision, I would be happy to further adjust my overall recommendation if the authors provide significantly deeper theoretical insights into limiting behavior.

**Limitations:**

Yes

**Strengths And Weaknesses:**

The paper presents the necessary accompanying literature in order to follow the main line of thought and all arguments are generally presented in a sound and elaborated way. Moreover, the provided proofs for the theoretical results as well as the derivations of concrete formulae seem to be correct. However, the theoretical contributions lack significant depth, since the presented results in Section 4.1, given the well-established Feynman-Kac identity and access to exact expected values, reduce to convergence of gradient descent. Furthermore, it is unclear what kind of infinite limit for ESS the authors intend to analyze since the cited result refers to the ESS as an approximation of the presented quantity and the number of particles thus still appears in the formula. The practical claims made throughout the paper are well-evidentiated through the experimental results. However, the soundness of the paper could be improved by elaborating on why not all of the baseline algorithms used for the first experiment are used troughout the other ones and why they are used in the first place. Moreover, the specific choice of kernels that are used could be motivated better. Finally, and specially because one of the attributed strengths of the developed algorithm is its generality, an analysis of different choices of kernels and different classes of experiments should be included.

The paper follows a clear outline and ideas and concepts are described cleanly and in an organized fashion. The string of thought is generally coherent, altough the cohesion of the paper would benefit from choosing experiments from more than one of the introduced key applications or at least discussing how to apply the method in the other cases.
The introduced method is presented in a very general way and thus has a broad scope of impact. In principle it is easy to apply and has many potential variants that could be experimented on including some that are already established to be useful. In order to be more easily accessible, without defaulting to already known algorithms, the authors could include an additional discussion on which types of kernels should be used when building upon their general framework.
Reading the paper it sounds as though the general SMC framework that is presented in a unified scope was already widely used in different stochastic optimization applications with intractable gradients. However, I believe most of the previous literature does not clearly refer to or develops theory through the lens presented in the paper and that there is a clear novelty and merit in the presented framework. But this does not seem to translate into new insights into the theory or practice of the application of SMC methods to this kind of setting.

---

> ### Author Rebuttal · Authors · 2026-03-31
>
> We thank the reviewer for their thoughtful review. We organise our response around main concerns below.
>
> **1. Discussion on kernels**
>
> > Have the authors tried out different choices of kernels? [...] are there kernels that are better adapted to certain kind of problems?
>
> We reply below this question from two perspectives to clarify the role of kernels.
>
> **(i) Empirical considerations:** To respond to reviewer's question, we have run new experiments to evaluate SOSMC with new kernels, namely MALA, RWM and HMC, for the experiment in Section 5.1, with new results at https://github.com/icmlsosmc/icml_sosmc/, in Figs LP1-4. We have found that in certain settings HMC and MALA can bring significant advantages. In particular, to highlight the criticality of the kernel choice, we have constructed a new challenging multimodal example (Fig. LP5 (right panel)). In this case, the Metropolis-corrected kernels (MALA, HMC) significantly outperform SOSMC-ULA, emphasising SOSMC is highly adaptable.
>
> From a practical perspective, kernel choice should be informed by problem geometry and computational budget. Local kernels (ULA, MALA) work well when successive targets are relatively close while momentum-based kernels (HMC) facilitate exploration. Furthermore, RWM, MALA, and HMC include a Metropolis step, which reduces bias and can be helpful when the target has isolated modes (e.g. Fig. LP4). Note in low dimensions, RWM is competitive despite not exploiting gradient information -- but one would expect the performance to degrade in high dimensions for RWM.
>
> **(ii) Theoretical motivation:** The choice of the kernel is problem-dependent, and depends on the geometry of the target sequence. Some prior work, e.g., Nilmeier et al. (2011), Chen & Roux (2015), Schönle et al. (2025), analyse deterministic (Hamiltonian), underdamped, and overdamped (ULA) dynamics within the MCMC framework. The central goal in these settings is to minimise the variance of the importance weights computed from forward and backward kernels. The results of Schönle et al. (2025) suggest that a deterministic kernel augmented with auxiliary momenta is advantageous; however, a definitive mathematical conclusion cannot yet be drawn in general contexts, and a thorough comparative analysis of different kernels is left for future work. We will include these discussions in our updated manuscript.
>
> **2. Benchmark choices**
>
> > Why did the authors choose to compare their algorithm to SOUL and ImpDiff?
>
> SOUL and Implicit Diffusion are two clear choices in the literature for stochastic optimisation with gradients of the form $\nabla_\theta \ell(\theta) = E_{\pi_\theta}[H_\theta(X)]$. In particular, SOUL estimates the gradients using MCMC on $\pi_\theta$ whereas Implicit Diffusion retains $N$ particles (targeting $\pi_\theta$ for every $k$) to approximate the gradient at step $k$. Our work is related to these two works, as we target the exact same problem using an SMC samplers framework.
>
> > why did they omit SOUL from the second and third experiment?
>
> As SOUL is MCMC based (every optimiser update requires a new MCMC run), it scales very poorly in dimensionality. Specifically in the examples considered in the paper the runtimes of SOUL were very uncompetitive, therefore we have left it out of the experiments.
>
> **3. Additional experiment**
>
> > Did the authors apply their algorithm to different problems than the fine tuning of EBMs?
>
> We added an image deblurring experiment in a nonsmooth latent-variable model with TV regularisation, following the setup in Encinar et al. (2025). This is an ill-conditioned MMLE problem. The considered experiments (with benchmarks), where both the true image and blurring applied is altered, are visualised at https://github.com/icmlsosmc/icml_sosmc/, and highlight consistent superior performance for SOSMC. Due to character limit, we refer the reviewer to our **Response to Reviewer kNru, #3 Additional experiment** for full details.
>
> **4. Other comments:**
>
> > [...] infinite limit for ESS [...]
>
> Apologies for this confusion. We will clarify the approximation of ESS in this setting.
>
> > Should the authors provide revisions to their paper, possibly by providing additional experimental material, that clarify these key questions, I would be happy to adjust my overall recommendation by a mark.
>
> We added significant new experimental material for (i) new kernels and (ii) new experiments; and did our best to reply all concerns about our framework. We hope that the reviewer finds these helpful -- we are happy to clarify any remaining questions.
>
> Nilmeier et al. (2011). Nonequilibrium candidate Monte Carlo: A new tool for efficient equilibrium simulation.
>
> Chen & Roux (2015). Constant-pH Hybrid Nonequilibrium Molecular Dynamics–Monte Carlo Simulation Method.
>
> Schönle et al. (2025). Efficient Monte-Carlo sampling of metastable systems using non-local collective variable updates.
>
> Encinar et al. (2025). Proximal Interacting Particle Langevin Algorithms.

---

> > ### Author Rebuttal · Reviewer_Lrtn · 2026-04-01
> >
> > The authors provided sufficient additional experiments and explanations on experiment design choices, so as promised I am happy to adjust my overall recommendation by a mark. However, the issue of lack of theoretical depth/new insights driven from theory persists so that I would not go higher.

---

### Official Review · Reviewer_mNuj · 2026-03-10

**Soundness:** 3
**Presentation:** 4
**Significance:** 4
**Originality:** 4
**Overall Recommendation:** 5
**Confidence:** 4

**Summary:**

The authors propose an SMC based stochastic optimization procedure for optimization problems with intractable objective gradient. They generalize a number of previous works where sampling algorithms were used to perform the first order optimization with the proposed framework and provide theoretical analysis and convergence results for the limit behavior.

**Compliance With Llm Reviewing Policy:**

Affirmed.

**Key Questions For Authors:**

In the paper [1] authors suggest that for a Bayesian posterior, stochastic approximation of the annealed posterior in AIS (a simplified  version of SMC) would lead to growing bias in expectation estimation with sequential progress which cannot be corrected with infinitesimal step sizes. This bias is mainly due to the Girsanov's equality and the mismatch between the stochastic paths ($\nabla U_k in this case) and the ideal values. I suspect a similar effect will arise in the particle-based analysis of the proposed optimization algorithm bounding the converged distribution away from the optimal.

1 - Can the authors provide an intuition about this problem whether it applies to the considered setup in the limit step size $\gamma\to 0$?

2 - If yes how it can be corrected? (there are relevant approaches to correct estimation bias in sequential sampling methods with Monte Carlo approximation e.g. [2]. Is this a relevant solution?)

I suggest the authors provide a section with the intuition of why this is a problem or not a problem and help clarify the limitations of the particle method in comparison to the ideal version of the algorithm. and provide general possible directions to address this problem.

[1] Differentiable Annealed Importance Sampling and the Perils of Gradient Noise, G. Zhang et al, 2021

[2] Theoretical guarantees for sampling and inference in generative models with latent diffusions, B. Tzen et al, 2019

**Limitations:**

Partially. the main issue is lack of theoretical analysis in the particle estimation of gradients, which are biased and convergence is not guaranteed (see Questions)

**Strengths And Weaknesses:**

Strength: This paper studies the first order optimization problem where the objective gradient is defined in terms of the parameterized distribution and its gradient is intractable. The proposed method is backed by the provided theoretical analysis in the limit and generalizes several other works under a unifying framework.

Presentation: The paper is well-written and structured. The motivation, problem formulation and method are carefully explained and accompanied by detailed background, and reasoned design and implementation procedure. Experiments include challenging problems that showcase the advantages of the proposed method.

Significance: The authors address the problem of optimization of expectations with intractable gradient information, a relevant problem to generative and amortized probabilistic modeling in machine learning. The proposed method is well-reasoned and back with theoretical results and is a valuable contribution to the field. Opening the avenue for new research in stochastic optimization and possible application of modern sampling approaches to the stochastic optimization problem.

Novelty: This work is a generalization of some groundbreaking previous work where sampling approaches are used for stochastic optimization with theoretical guarantees. Application of SMC to optimization in time is a very interesting and well-implemented idea with gradual progress toward the real expectation which may be difficult to approximate due to initialization and optimization landscape.

---

> ### Author Rebuttal · Authors · 2026-03-31
>
> We thank the reviewer for their thoughtful review. We respond to their main concerns below.
>
> **1. Bias accumulated in the optimisation across time**
>
> > In the paper [1] authors suggest that for a Bayesian posterior, stochastic approximation of the annealed posterior in AIS (a simplified version of SMC) would lead to growing bias in expectation estimation with sequential progress which cannot be corrected with infinitesimal step sizes. This bias is mainly due to the Girsanov's equality and the mismatch between the stochastic paths ($\nabla U_k in this case) and the ideal values. I suspect a similar effect will arise in the particle-based analysis of the proposed optimization algorithm bounding the converged distribution away from the optimal.
>
> Thank you for the reference, we will include this in our discussion. The bias coming from Monte Carlo estimates can be controlled with a rate $\mathcal{O}(C_t / N)$ under some regularity assumptions (see Cuin et al. (2025) for a related example). However, to make the time-dependent constant $C_t$ uniform-in-time (i.e. $C_t = C$ for all $t$), one requires typically stringent assumptions, e.g. bounded gradients. Our setting can be analysed in this setting using results from Del Moral (2004, 2013), which we leave for future work.
>
> > 1.  Can the authors provide an intuition about this problem whether it applies to the considered setup in the limit step size $\gamma \rightarrow 0$.
>
> We thank for this interesting question! Indeed, in the limit $\gamma \to 0$, one can obtain a continuous-time equivalent of our Feynman-Kac flow, also studied in Del Moral (2004, 2013). The general story we outlined above still holds in this case, that is, over fixed time horizons, the particle approximation error decreases with $N$, whilst obtaining time-uniform control typically requires stringent conditions. Under general conditions, the best known upper bounds on the bias grows over time. However, while time-uniform theoretical guarantees require stringent assumptions, our algorithms perform well in practice, showing the need of new approaches to prove these results.
>
> > 2. If yes how it can be corrected? (there are relevant approaches to correct estimation bias in sequential sampling methods with Monte Carlo approximation e.g. [2]. Is this a relevant solution?)
>
> In general, bias introduced by SMC cannot be removed for finite $N$ without additional assumptions. The most direct way to reduce the bias is by increasing $N$. To the best of our knowledge, SMC bias cannot be generically corrected, but its variance can be reduced via proper choices of forward and backward kernels. Indeed, this flexibility is an advantage of the SOSMC framework.
>
> > I suggest the authors provide a section with the intuition of why this is a problem or not a problem and help clarify the limitations of the particle method in comparison to the ideal version of the algorithm. and provide general possible directions to address this problem.
>
> Thanks for this suggestion! We will definitely add a discussion to provide clear intuition how the idealised version of the method relate to the practical implementation, what error bounds are within reach, and what cannot be shown in general.
>
> Many thanks again for the interesting questions and suggestions. We would also like to make the reviewer aware of a new experiment (see **Response to Reviewer kNru, #3 Additional experiment**) and additional kernels variants experimented with (see **Response #1 to Reviewer Lrtn**), with relevant figures found at: https://github.com/icmlsosmc/icml_sosmc. All code for these new examples will be made available in the future in the public repository.
>
> **References**
>
> Cuin et al. (2025), Learning Latent Variable Models via Jarzynski-adjusted Langevin Algorithm
>
> Del Moral (2004), Feynman-Kac Formulae
>
> Del Moral (2013), Mean Field Simulation for Monte Carlo Integration

---

> > ### Author Rebuttal · Reviewer_mNuj · 2026-04-02
> >
> > I thank the authors for their detailed response on the questions. The additional experimental results are great contribution that show how different design choices impact the performance of the proposed particle method. I believe with the new updates this paper has merits on par with the ICML expectations.

---

### Official Review · Reviewer_jgkx · 2026-03-13

**Soundness:** 3
**Presentation:** 4
**Significance:** 4
**Originality:** 4
**Overall Recommendation:** 5
**Confidence:** 4

**Summary:**

The article develops a stochastic optimization method for problems with latent observations. In the problems considered, the conditional expectation or exact samples from the conditional distribution of the latent observations given the data are not available and MCMC-based sampling is expensive. The article proposes the use of sequential Monte Carlo algorithms for approximating conditional expectations in each EM iteration. The theoretical convergence properties of the proposed algorithm are studied in several fine-tuning energy-based models.

**Compliance With Llm Reviewing Policy:**

Affirmed.

**Key Questions For Authors:**

With regards to uncertainty quantification, could you use the SMC algorithm for estimating the information matrix and demonstrate if the estimates are efficient (i.e., they attain the Cramer-Rao type lower bound) using theoretical or numerical investigation?

**Limitations:**

The limitations of the framework have not been described clearly; however, several areas for improvement have been discussed in the conclusions.

**Strengths And Weaknesses:**

Strengths: The key idea of the paper is to replace fresh MCMC samples at each iteration of the EM optimization by sequential Monte Carlo samples that reuse samples across iterations. The scope of the method includes a broad class of problems in machine learning and statistics. Several stochastic gradient algorithms are shown to be special cases of the proposed algorithm. In addition, the theoretical behavior of the effective sample size has been derived for the sequential Monte Carlo algorithm. The practical utility of the proposed framework has been demonstrated through a numerical study of Langevin process reward tuning and the analysis of several benchmark datasets as well as an MNIST handwritten digits dataset. Overall, the framework is shown to improve both optimization performance and computational costs.

Weaknesses: The framework and experiments mostly focus on reward tuning of Langevin processes and energy-based models. However, other inferential aspects, including quantifying uncertainty in estimation and predictions, have not been demonstrated.

---

> ### Author Rebuttal · Authors · 2026-03-31
>
> We thank the reviewer for thoughtful review. We organise our response around main concerns below.
>
> **1. Estimator efficiency**
>
> > With regards to uncertainty quantification, could you use the SMC algorithm for estimating the information matrix and demonstrate if the estimates are efficient (i.e., they attain the Cramer-Rao type lower bound) using theoretical or numerical investigation?
>
> Thank you for this question. To address this concern, we have set up a new experiment to demonstrate the efficiency of our SMC estimators in an empirical setting as requested. We aim at demonstrating the efficiency of our estimators by empirically testing the Cramer-Rao lower bound (CRLB) of the form:
> $$
> \mathsf{var}(\hat{\theta}) \geq \frac{1}{I_n(\theta)},
> $$
> where $\mathsf{var}(\hat{\theta})$ denotes the variance of our estimator and $I_n(\theta)$ is the Fisher-information matrix (note that this formula is for unbiased estimators -- but we use it for testing purposes).
>
> For this, we consider an MMLE-type problem and work with the latent variable model over $\mathbb{R}$
> $$p(y | x) = \mathcal{N}(y; x, \sigma^2_y),$$ and
> $$p_\theta(x) = w \mathcal{N}(x;\theta, \sigma_x^2) + (1-w) \mathcal{N}(x;a, \sigma_x^2),$$
> where $a=-3$ $w=0.7$, $\sigma_x=1$ and $\sigma_y=0.5$.
>
> This results in an analytically tractable marginal likelihood (hence good for testing purposes):
> $$p_\theta(y) = w \mathcal{N}(y;\theta, \sigma_x^2+\sigma_y^2) + (1-w) \mathcal{N}(y;a, \sigma_x^2+\sigma_y^2).
> $$
> Given the explicit form of $p_\theta(y)$, one can explicitly compute $I_n(\theta)$ as a function of $\theta$. This problem fits to our framework, see, Section 2.2 of our paper. Running $M$ SOSMC-ULA runs on the model will produce parameter estimates from which we can empirically estimate $\mathsf{var}(\hat{\theta})$.
>
> Specifically, we set $\theta^\star=2$ and first compute $I_n(\theta^*)$, with $n$ denoting the number of datapoints, and subsequently the CRLB. Then, for each $m \in [M]$, we generate a fresh dataset $y_{1:n}^{(m)}  \sim p_{\theta^\star}$, compute the exact MLE to obtain $\hat \theta_{\text{MLE}}$, and run SOSMC-ULA to obtain $\hat \theta_{SOSMC}$. Across the $M=500$ datasets, we then compute the empirical variance of $\hat \theta_{MLE}$ and of $\hat{\theta}_{SOSMC}$, which we compare to the CRLB in Fig. UQ1, at https://github.com/icmlsosmc/icml_sosmc/blob/main/uncertainty_quantification/crlb_mmle_example.pdf. The empirical variance of both estimators decrease with sample size $n$ and closely follow the CRLB, with SOSMC-ULA becoming nearly indistinguishable from MLE as $n$ increases.
>
> **2. Uncertainy quantification and a new experiment**
>
> > The framework and experiments mostly focus on reward tuning of Langevin processes and energy-based models. However, other inferential aspects, including quantifying uncertainty in estimation and predictions, have not been demonstrated.
>
> We thank the reviewer for raising this concern. To address this, we have added an image debluring problem as an example where we demonstrate uncertainty quantification and is a primary inferential target. The discussion about this experiment can be found in **Response to Reviewer kNru, #3 Additional experiment** and can be seen from Figures ID 1--12 at https://github.com/icmlsosmc/icml_sosmc. In this setting, we maintain a particle approximation to the posterior distribution over the latent clean image $x$, allowing computation of both posterior mean reconstructions and pixel-wise posterior variance maps, which provide a direct measure of local uncertainty in the recovered image.
>
> In particular, Figs. ID4, ID8, and ID12 report, for three different deblurring setups, such variance maps alongside the corresponding posterior mean reconstructions for both MYPGD and SOSMC-MYULA. Figs. ID4 and ID8 illustrate the uncertainty in the SOSMC-MYULA particle cloud is more spatially structured and interpretable than that of MYPGD. Specifically, for SOSMC-MYULA, higher variance is concentrated around ambiguous regions, such as edges and  fine-scale details. For example, in Fig. ID4 uncertainty is elevated in regions corresponding to the guitar body and strings, whilst in Fig. ID8 in regions corresponding to flower contours. Fig. ID12 complements this by illustrating a more challenging regime (see Fig. ID9 vs ID1), in which variance maps are seen to reflect an increase in posterior uncertainty as the inverse problem becomes more difficult.
>
>
> **3. Other comments**
>
> > The limitations of the framework have not been described clearly; however, several areas for improvement have been discussed in the conclusions.
>
> We will clarify these limitations in our update, many thanks for this comment.

---

> > ### Author Rebuttal · Reviewer_jgkx · 2026-04-03
> >
> > The authors have adequately responded to my comments.

---

### Decision · Program_Chairs · 2026-04-30

**Decision:**

Accept (regular)

**Comment:**

The paper considers the problem of optimizing functions with intractable gradients. The paper mentions that prior stochastic approximation methods require expensive inner MCMC sampling loops at each iteration. The paper proposes a SOSMC framework where the key idea is to replace these inner loops with sequential Monte Carlo samplers that reuse particles across optimization steps. The proposed approach provides a unifying perspective that generalizes several existing methods (JALA-EM, SOUL, Implicit Diffusion). All four reviewers recognized the paper's contribution in generalizing and unifying existing methods under a common SMC framework.  Reviewers jgkx, mNuj, and kNru also mentioned the work as technically sound. Author rebuttal addressed empirical concerns of different reviewers.  One concern about theoretical depth for finite-particle analysis remains but it does not outweigh the paper's current contribution. Therefore,  I recommend accepting the paper while requesting the authors to incorporate comments from the reviewers.